# Interstitial-fluid shear stresses induced by vertically oscillating head motion lower blood pressure in hypertensive rats and humans

Shuhei Murase[1,2,22], Naoyoshi Sakitani[1,3,22], Takahiro Maekawa[1], Daisuke Yoshino[4], Kouji Takano[5], Ayumu Konno[6], Hirokazu Hirai[6], Taku Saito[2], Sakae Tanaka[2], Keisuke Shinohara[7], Takuya Kishi[8], Yuki Yoshikawa[9], Takamasa Sakai[9], Makoto Ayaori[10], Hirohiko Inanami[11], Koji Tomiyasu[12], Atsushi Takashima[13], Toru Ogata[1,14], Hirotsugu Tsuchimochi[15], Shinya Sato[16], Shigeyoshi Saito[17], Kohzoh Yoshino[18], Yuiko Matsuura[19], Kenichi Funamoto[20], Hiroki Ochi[1], Masahiro Shinohara[1], Motoshi Nagao[1] & Yasuhiro Sawada[1,2,3,4,21] ✉

The mechanisms by which physical exercise benefits brain functions are not fully understood. Here, we show that vertically oscillating head motions mimicking mechanical accelerations experienced during fast walking, light jogging or treadmill running at a moderate velocity reduce the blood pressure of rats and human adults with hypertension. In hypertensive rats, shear stresses of less than 1 Pa resulting from interstitial-fluid flow induced by such passive head motions reduced the expression of the angiotensin II type-1 receptor in astrocytes in the rostral ventrolateral medulla, and the resulting antihypertensive effects were abrogated by hydrogel introduction that inhibited interstitial-fluid movement in the medulla. Our findings suggest that oscillatory mechanical interventions could be used to elicit antihypertensive effects.

Exercise is effective as a therapeutic and preventative measure for numerous physical disorders and diseases, including hypertension[1,2]—a major cause of stroke and cardiovascular disease and the biggest risk factor for death worldwide[3]. However, the mechanisms underlying the antihypertensive effect of exercise are unclear.

While the majority (more than 90%) of human hypertension comprises essential hypertension, the cause of which is unidentifiable[4], long-term regulation of blood pressure has been recognized to be largely dependent on sodium-excretion-adjusting systems mainly involving kidney functions[5]. Elevated activity of the sympathetic nervous system also contributes to the development of hypertension[6–8]. The rostral ventrolateral medulla (RVLM), which is located in the brainstem,

has a critical role in determining the basal activity of the sympathetic nervous system, and its functional integrity is essential for the maintenance of basal vasomotor tone and regulation of blood pressure[6,9]. Angiotensin II is the major bioactive peptide of the renin–angiotensin system (RAS), and is known to regulate blood pressure as well as other biological processes, such as cell growth, apoptosis and migration, and inflammation and fibrosis[10]. The biological effects of angiotensin II are mediated by its interaction with two distinct high-affinity G-protein-coupled receptors, and the angiotensin II type 1 (AT1R) and type 2 (AT2R) receptor. Of these receptors, AT1R is responsible for most of the known physiological and pathophysiological processes related to angiotensin II. Whereas RAS is involved in the functional regulation of

various peripheral organs and tissues such as the kidney and vessels, it also regulates brain functions within the blood–brain barrier, including the control and maintenance of sympathetic nerve activity and cognitive ability[11]. In particular, the role of AT1R signalling in the RVLM in cardiovascular regulation has been extensively studied. For example, the pressor and depressor responses to angiotensin II and angiotensin II antagonists, respectively, injected into the RVLM have been reported to be enhanced in spontaneously hypertensive rats (SHRs)[12,13]. We have previously shown that treadmill running at moderate velocities alleviates sympathetic nerve activity, and that this involves the attenuation of AT1R signalling in the RVLM of stroke-prone SHRs (SHRSPs)[14], a substrain of SHRs that exhibit more severe hypertension compared with SHRs[15]. However, the details about the changes in AT1R signalling in the RVLM of these hypertensive rats have yet to be elucidated. It remains unclear what type(s) of cells (for example, neurons or astrocytes) are primarily responsible for the altered AT1R signalling in the RVLM of SHRs or SHRSPs. Furthermore, the causal relationship between the increased AT1R signal activity in the RVLM and high blood pressure in SHRs or SHRSPs in their steady state (that is, apart from their responses to pharmacological interventions) is unclear.

AT1R has also been shown to have a vital role in the regulation of a variety of physiological or pathological processes, including cellular responses to mechanical perturbations[16,17]. For example, mechanical stretching of cardiac myocytes activates AT1R signalling[18], and fluid shear stress of an average of 1.5 Pa lowers AT1R expression in human vein endothelial cells[19]. Although intervention of the angiotensin-II–AT1R system using pharmacological approaches, such as the administration of angiotensin-converting enzyme inhibitor or a selective AT1R blocker, has been established as an effective therapeutic strategy for hypertension[20], the mechanoresponsive attenuation of AT1R signalling has not been clinically used as an antihypertensive measure.

Many physical workouts, particularly aerobic exercise, involve vertical body motions that generate mechanical accelerations in the head at the time of foot contact with the ground (that is, when landing). The importance of mechanical loads is well established in the physiological regulation of bones, which allows for only tiny deformations[21]. Osteocytes, the mechanosensory cells embedded in bones[22], are assumed to undergo minimal deformations under physiological conditions. We have reported that fluid shear stress on osteocytes arising from physical-activity-induced interstitial-fluid flow has an important role in maintaining bone homeostasis[23]. Given that the brain is not a rigid organ, minimally deforming forces or stress-distribution changes in the brain during exercise or even during activities of daily living (such as walking) may produce beneficial effects. We have shown that, in the prefrontal cortex (PFC) of rodents, moderate mechanical intervention-induced fluid shear stress modulates serotonin signalling in neurons in situ[24]. On the basis of these previous findings,

and considering the distribution of interstitial fluid throughout the whole brain, here we hypothesized that moderate mechanical intervention might have antihypertensive effects involving the fluid-shear-stress-mediated modulation of AT1R signalling in the RVLM.

## Results

### The application of cyclical mechanical forces to the head through passive motion lowers blood pressure in SHRSPs

To determine the effects of a mechanical intervention of moderate intensity on blood pressure, we first sought to develop an experimental system that reproduces the acceleration generated in a rat's head during treadmill running at a modest velocity (20 m min$^{-1}$)—a typical experimental intervention to test the effects of physical exercise on rats[25,26]. In a recent study, we observed that treadmill running of rats (20 m min$^{-1}$) generated a 5 mm vertical oscillation of their heads with around 1.0$g$ peak accelerations at 2 Hz. We therefore developed a passive head motion (PHM) system to produce 2 Hz 5 mm vertical oscillations exerting 1.0$g$ acceleration peaks in the heads of rodents[24,27] (Supplementary Video 1). Here we examined the effects of such mechanical intervention on blood pressure in SHRSPs, using the PHM system. Similar to the antihypertensive effect of treadmill running on SHRs or SHRSPs that we and others reported previously[25,28–31], the application of PHM (30 min per day, 28 consecutive days; Fig. 1a) significantly lowered the blood pressure of the rats (Fig. 1b,c), whereas the heart rate was not significantly affected by PHM (Fig. 1d). Anaesthesia alone (daily 30 min) did not significantly alter the blood pressure in SHRSPs (Extended Data Fig. 1a), indicating that the antihypertensive effect resulted specifically from PHM. The anticardiac hypertrophy effect of PHM on SHRSPs (Fig. 1e) as well as the lack of these PHM effects on control normotensive rats (Wistar Kyoto (WKY)) (Fig. 1b,c,e) were also consistent with previous reports describing treadmill running as an antihypertensive intervention for SHRs[29]. As was observed in our treadmill-running experiments[14], PHM decreased 24 h urinary noradrenaline excretion in SHRSPs (Fig. 1f). This suggests that PHM mitigates sympathetic hyperactivity[32]. Collectively, these results support our hypothesis that the cyclical application of a moderate mechanical intervention to the head has an antihypertensive effect. Notably, PHM for at least 4 weeks significantly decreased or delayed the incidence of stroke in SHRSPs (Extended Data Fig. 1b–d).

Next, we characterized PHM as an antihypertensive intervention by testing for various directions, frequencies and amplitudes (peak magnitudes of acceleration). PHM that generated acceleration peaks of 1.0$g$ in the rostral–caudal direction, but not in the left–right direction, had antihypertensive effects on SHRSPs (Extended Data Fig. 2a–c), suggestive of directional selectivity. Regarding the frequency, vertical PHM (peak magnitude of 1.0$g$) of 0.5 Hz, but not of 0.2 Hz, lowered the blood pressure of SHRSPs by approximately the

**Fig. 1 | Application of cyclical mechanical intervention to the head by passive motion lowers blood pressure in SHRSPs and AT1R expression in SHRSP RVLM astrocytes. a**, Schematic of the experimental protocol to analyse the effects of PHM on blood pressure in rats. HR, heart rate. **b,c**, Time courses (**b**) and values on day 29 (**c**) of the MAP in WKY rats and SHRSPs that were treated either with daily PHM or anaesthesia only (SHRSP without versus with PHM: $P = 0.1344$ (day 15), $P = 0.0110$ (day 22), $P = 0.0463$ (day 29); WKY without versus with PHM: $P > 0.9999$ (day 15, day 22 and day 29) (**b**); $P = 0.9739$ (column 1 versus 2), $P = 0.0046$ (column 3 versus 4) (**c**)). $n = 7$ (each group of WKY) and $n = 8$ (each group of SHRSP) rats. **d**, Heart rate values on day 29 ($P = 0.9650$ (column 1 versus 2), $P = 0.2362$ (column 3 versus 4)). $n = 7$ (each group of WKY) and $n = 8$ (each group of SHRSP) rats. **e**, The relative heart weight (heart weight/whole body weight) measured on day 30 ($P = 0.9866$ (column 1 versus 2), $P = 0.0152$ (column 3 versus 4)). $n = 10$ (WKY, −PHM), $n = 13$ (WKY, +PHM), $n = 10$ (SHRSP, −PHM) and $n = 14$ (SHRSP, +PHM) rats. **f**, The 24 h (day 29 to day 30) urinary noradrenaline excretion ($P = 0.9854$ (column 1 versus 2), $P = 0.0085$ (column 3 versus 4)). $n = 8$ (each group of WKY), $n = 16$ (SHRSP, −PHM) and $n = 13$ (SHRSP, +PHM) rats. **g,h**, Micrographic images of anti-NeuN (blue), anti-GFAP (green) and anti-AT1R

(red) immunostaining of the RVLM of WKY rats (**g**) and SHRSPs (**h**) that were either left sedentary (top) or treated with PHM (bottom) under anaesthesia (30 min per day, 28 days). The higher-magnification images (centre and right) show the areas indicated by dotted rectangles in the low-magnification images (left). The arrows point to anti-AT1R immunosignals that overlap with anti-GFAP, but not anti-NeuN, immunosignals in the merged images. Scale bars, 50 µm. Images are representative of three rats. **i,j**, Quantification of AT1R-positive neurons (**i**) and astrocytes (**j**) in the RVLM of WKY rats and SHRSPs that were either left sedentary or treated with PHM. A total of 50 NeuN-positive (NeuN$^+$) cells and 100 GFAP-positive (GFAP$^+$) cells was analysed for each rat ($P = 0.9602$ (column 1 versus 2), $P = 0.9215$ (column 1 versus 3), $P = 0.9313$ (column 3 versus 4) (**i**); $P = 0.9455$ (column 1 versus 2), $P = 0.0004$ (column 1 versus 3), $P = 0.0002$ (column 3 versus 4) (**j**)). $n = 3$ rats for each group. Data are mean ± s.e.m. Statistical analysis was performed using two-way repeated-measures analysis of variance (ANOVA) with Bonferroni's post hoc multiple-comparison test (**b**) or one-way ANOVA with Tukey's post hoc multiple-comparison test (**c**–**f**, **i** and **j**); *$P < 0.05$, **$P < 0.01$, ***$P < 0.001$; NS, not significant.

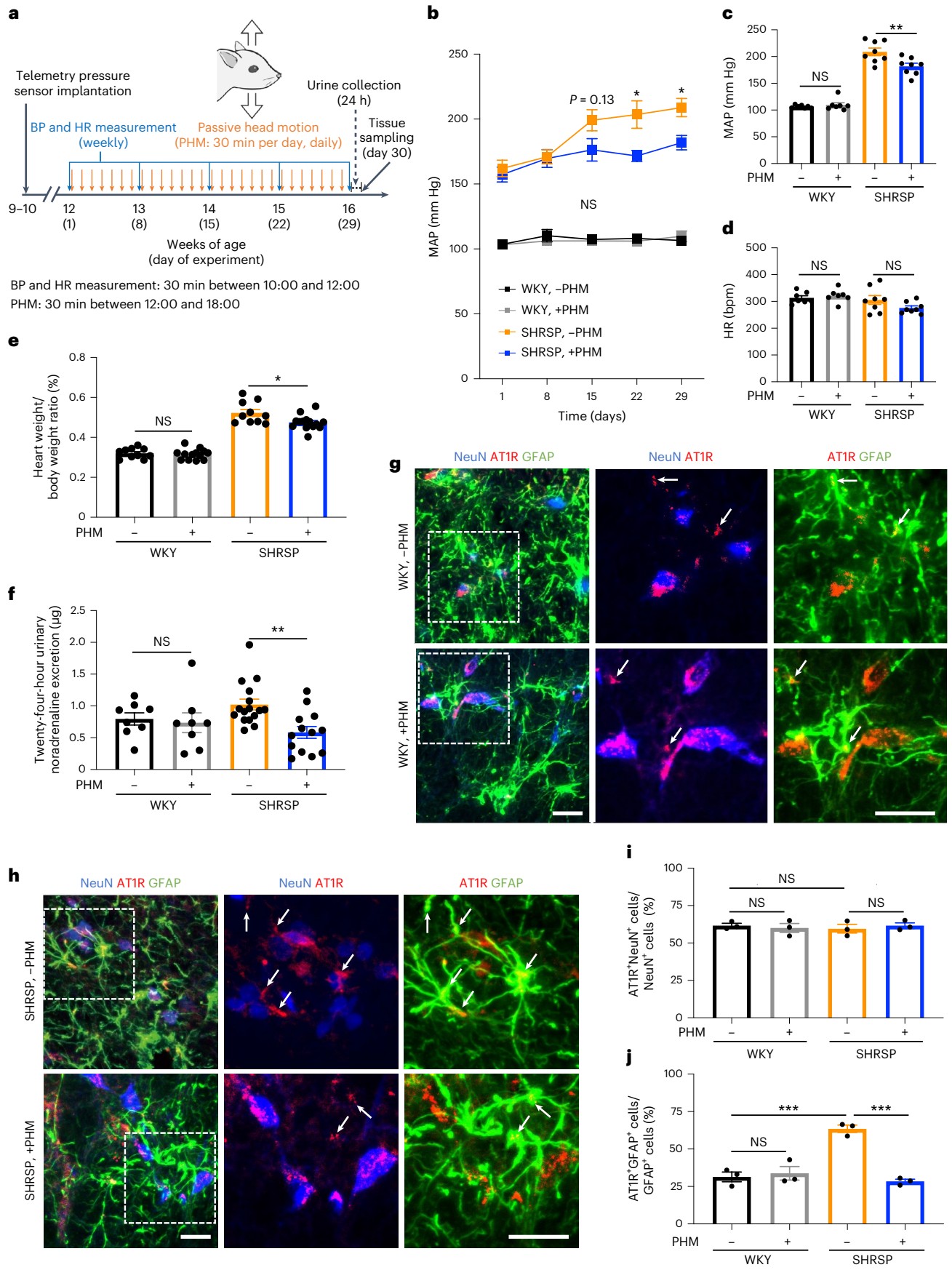

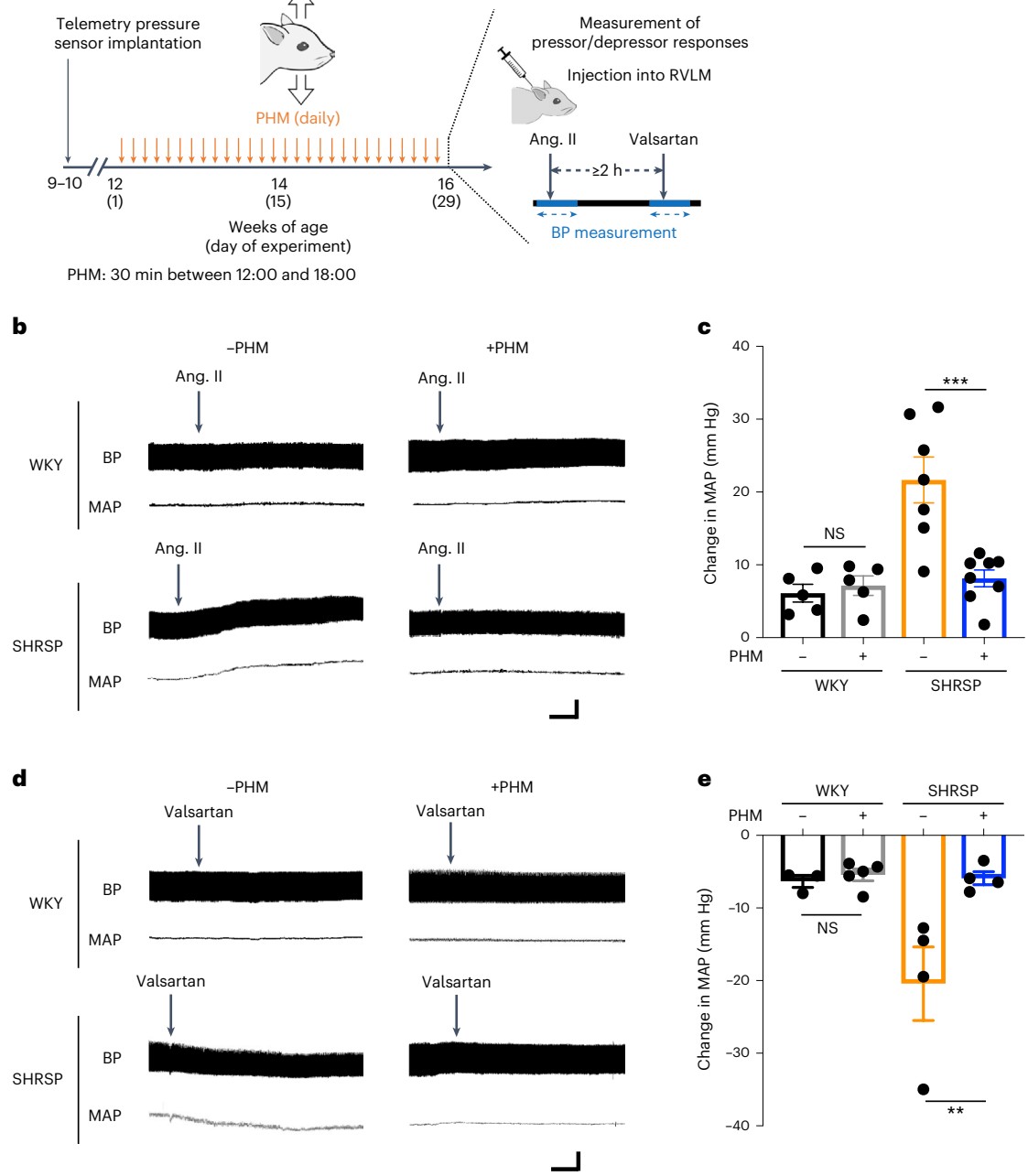

**Fig. 2 | PHM alleviates the sensitivity of the RVLM of SHRSPs to angiotensin II or valsartan. a**, Schematic of the experimental protocol to analyse the effects of PHM on the sensitivity to angiotensin II (ang. II) or valsartan injected into the unilateral RVLM. Angiotensin II (100 pmol) was injected into the unilateral RVLM of WKY rats and SHRSPs that were either left sedentary (daily anaesthesia) or treated with PHM (30 min per day, 28 days), with their blood pressure monitored under urethane anaesthesia. Injection of valsartan (100 pmol) into the RVLM was conducted at least 2 h after the injection of angiotensin II. **b**, Representative trajectories of blood pressure (top) and MAP (bottom). The arrows point to the time of the initiation of RVLM injection of angiotensin II. **c**, Quantification of the change in MAP caused by injection of angiotensin II ($P = 0.9876$ (column 1 versus 2), $P = 0.0003$ (column 3 versus 4)). $n = 5$ (each group of WKY), $n = 7$ (SHRSP, −PHM) and $n = 8$ (SHRSP, +PHM) rats. **d,e**, The effects of injection of valsartan (100 pmol) in the RVLM were examined as described in **b** and **c** (the change in blood pressure and MAP (**d**) and the MAP trajectory (**e**)) ($P = 0.9953$ (column 1 versus 2), $P = 0.0099$ (column 3 versus 4) (**e**)). $n = 3$ (WKY, −PHM), $n = 5$ (WKY, +PHM) and $n = 4$ (each group of SHRSPs) rats. Data are mean ± s.e.m. Statistical analysis was performed using one-way ANOVA with Tukey's post hoc multiple-comparison test; $**P < 0.01$, $***P < 0.001$. For **b** and **d**, scale bars, 1 min (horizontal) and 50 mm Hg (vertical).

same extent as 2 Hz PHM (Extended Data Fig. 3a–c). Furthermore, PHM generating a peak magnitude of 0.5*g*, but not 0.2*g*, was approximately as antihypertensive as 1.0*g* PHM (Extended Data Fig. 3a,d,e). These results suggest the existence of a threshold and a plateau phase of frequency and amplitude (magnitude) of PHM in terms of its antihypertensive effects.

## PHM downregulates AT1R expression in RVLM astrocytes in SHRSPs

We next examined the mechanism of how PHM alleviates the development of hypertension in SHRSPs. We previously reported that downregulation of AT1R signalling in the RVLM is responsible for the treadmill-running-induced sympathoinhibition in SHRSPs[14]. Given

the mechanical regulation of AT1R expression in endothelial cells[19], we examined whether PHM modulated AT1R expression in RVLM neurons and astrocytes in SHRSPs. In our histochemical analysis, we defined neuronal nuclei (NeuN)-positive cells as neurons[33] and glial fibrillary acidic protein (GFAP)-positive cells as astrocytes[34]. PHM (30 min per day, 28 days) did not significantly change the relative population of AT1R-expressing neurons and astrocytes in the RVLM of WKY rats (Fig. 1g). By contrast, PHM for 4 weeks significantly decreased the expression of AT1R in the astrocytes, but not in the neurons, in the RVLM of SHRSPs (Fig. 1h). Notably, AT1R expression in the RVLM neurons was comparable between WKY rats and SHRSPs, either with or without PHM (Fig. 1i). By contrast, AT1R expression was significantly higher in the RVLM astrocytes in SHRSPs without PHM (Fig. 1j (columns 1 and 3)). PHM lowered AT1R expression in RVLM astrocytes of SHRSPs to a level that was equivalent to that of WKY rats (Fig. 1j (columns 1, 2 and 4)). Taken together, AT1R expression in RVLM astrocytes appeared to be correlated with the antihypertensive effect of PHM on SHRSPs. Consistent with this observation, treadmill running for 4 weeks in SHRSPs also decreased AT1R expression in the RVLM astrocytes, but not in the neurons (Extended Data Fig. 4a–c).

### PHM alleviates the sensitivity of the RVLM in SHRSPs to angiotensin II or angiotensin II antagonist

We next sought to examine whether the PHM-induced decrease in AT1R expression in RVLM astrocytes in SHRSPs (Fig. 1j (columns 3 and 4)) was functionally relevant to the suppression of AT1R signalling. To this end, we analysed the pressor responses to angiotensin II injected into the unilateral RVLM of WKY rats and SHRSPs that were either treated with PHM for 4 weeks or left sedentary under anaesthesia (30 min per day, 28 days) (Fig. 2a). As we previously reported[14], SHRSPs without PHM exhibited a significantly greater pressor response to angiotensin II administered to the RVLM compared with WKY rats (Fig. 2b (compare top left and bottom left) and 2c (compare columns 1 and 3)). PHM for 4 weeks alleviated the pressor response to angiotensin II injected into the RVLM of SHRSPs, but not of WKY rats (Fig. 2b (compare left and right) and 2c (compare columns 1 versus 2 and 3 versus 4)). Furthermore, the depressor response to angiotensin II antagonist injected into the unilateral RVLM[13] was also mitigated by 4-week PHM in SHRSPs, but not in WKY rats (Fig. 2d,e). These results support the functional relevance of the PHM-induced decrease in AT1R expression in RVLM astrocytes of SHRSPs (Fig. 1j).

To examine whether the increased AT1R expression in RVLM astrocytes of SHRSPs was associated with their development of hypertension, we manipulated AT1R signalling by introducing exogenous expression of AT1R-associated protein (AGTRAP), which interacts with AT1R and tempers the angiotensin-II-mediated signals by promoting AT1R internalization[35]. To this end, we used an adeno-associated virus (AAV)-mediated gene delivery system[36]. AAV serotype 9 (AAV9) vectors were injected locally to transduce the RVLM cells (Fig. 3a and Extended Data Fig. 5a). To achieve astrocyte- and neuron-specific gene expression, we used the AAV9 vectors containing the mouse *Gfap* promoter (AAV-GFAP) and rat neuron-specific enolase (*Eno2* (also known as NSE)) promoter (AAV-NSE), respectively (Fig. 3a). As these vectors contained a region encoding GFP and the 2A sequence of

porcine teschovirus-1 (P2A; self-cleaving peptides)[37] (Fig. 3a), observation of the green fluorescence enabled us to identify the cells in which the transgene was expressed (Fig. 3b,c and Extended Data Fig. 5a–e). AAV-mediated expression of AGTRAP in astrocytes (Fig. 3b) but not in neurons (Fig. 3c) of the bilateral RVLMs in SHRSPs significantly lowered blood pressure compared with in the control SHRSPs in which only GFP was virally expressed in the RVLM astrocytes or neurons (Fig. 3d,e). Furthermore, AAV-mediated expression of AGTRAP in astrocytes, but not neurons, in the bilateral RVLMs of SHRSPs decreased the 24 h urinary noradrenaline excretion (Fig. 3f). Injection of the control AAV vector (GFAP-control or NSE-control) did not significantly affect the blood pressure in SHRSPs (Extended Data Fig. 5f). These results support the importance of AT1R signal intensity in the RVLM astrocytes for the development of hypertension and sympathetic hyperactivity in SHRSPs. However, the blood-pressure-lowering effect of the exogenous expression of AGTRAP in RVLM astrocytes was not long lasting and became non-significant 3 weeks after the AAV injection (Extended Data Fig. 5g) perhaps due to a compensatory or neutralizing mechanism for the steady-state AT1R signalling[38–40] that is yet to be defined.

### PHM generates low-amplitude pressure waves and induces interstitial-fluid movement in the rat RVLM

We next sought to determine the physical effects that PHM produced in the rat RVLM. To do so, we analysed local pressure changes using a telemetry pressure sensor (Fig. 4a) as we described previously[24]. PHM generated pressure waves (changes) with a peak amplitude of around 1.2 mm Hg (Fig. 4b–d). As the frequency of these PHM-induced pressure changes was the equivalent to that of PHM (2 Hz), they were probably due to the local cyclical microdeformation generated during PHM. We then postulated an analogy to bone—an organ that yields to only minimal deformation. As the function of bone is known to be modulated by interstitial-fluid-flow-derived shear stress on osteocytes[23], we speculated that the interstitial-fluid movement generated by microdeformation-induced stress-distribution changes in the brain might result in the shear-stress-mediated regulation of nervous cell functions[24].

To analyse the PHM-induced interstitial-fluid movement in the RVLM, we injected an iodine-based contrast agent (Isovist) into the RVLM of anaesthetized rats, and tracked its distribution using sequential micro-computed tomography (μCT) (Fig. 4e) as we previously did to analyse the movement of intramuscular interstitial fluid[41]. We found that PHM significantly promoted Isovist spreading in the rostral–caudal and dorsal–ventral (Fig. 4f) directions (Fig. 4g,h). By contrast, PHM did not significantly affect the left–right spreading (Fig. 4f) of Isovist (Fig. 4g,h). From the extent of PHM-induced increase in Isovist spread, we estimated that the velocity of interstitial-fluid movement in the rat RVLM was increased by approximately two to three times during PHM; however, precise evaluation was difficult owing to differences in the size and time scales between our μCT analysis (about 100 μm, more than 30 min interval) and PHM effects at the cellular level (of the order of 1 μm, about 0.5 s interval).

Our analyses using multiphoton microscopy (Extended Data Fig. 6a–c) and magnetic resonance imaging (MRI) (Extended Data Fig. 6d) indicated that the interstitial space of the rat RVLM is not

**Fig. 3 | AAV-mediated expression of AGTRAP in RVLM astrocytes, but not neurons, lowers blood pressure in SHRSPs. a**, Schematic of the experimental protocol to analyse the effects of AAV-mediated transduction of RVLM astrocytes or neurons with the *Agtrap* gene. ITR, inverted terminal repeat. **b,c**, Astrocyte-specific (**b**) and neuron-specific (**c**) transgene expression by RVLM injection of AAV9 vectors. Micrographic images of GFP (green) and anti-GFAP (**b**) or anti-NeuN (**c**) immunostaining (blue) of RVLM in SHRSPs 15 days after the injection of the AAV9 vectors indicated at the top. Scale bars, 50 μm. Images are representative of three rats. **d–f**, MAP values just before (day 1) (**d**) and 2 weeks after (day 15) (**e**) AAV injection into the RVLM ($P = 0.0222$

(GFAP-control versus GFAP-AGTRAP) and $P > 0.9999$ (NSE-control versus NSE-AGTRAP) (**d**); $P = 0.0229$ (column 1 versus 2), $P = 0.6864$ (column 3 versus 4) (**e**); $n = 6$ (GFAP-control), $n = 7$ (GFAP-AGTRAP), $n = 7$ (NSE-control) and $n = 6$ (NSE-AGTRAP) rats) and 24 h urinary noradrenaline excretion (**f**) ($P = 0.0497$ (column 1 versus 2), $P = 0.7455$ (column 3 versus 4); $n = 4$ rats for each group) in SHRSPs after RVLM injection of AAV9 vectors. Data are mean ± s.e.m. Statistical analysis was performed using two-way repeated-measures ANOVA with Bonferroni's post hoc multiple-comparison test (**d**) or one-way ANOVA with Tukey's post hoc multiple-comparison test (**e** and **f**); *$P < 0.05$.

randomly structured but oriented approximately along the centroidal line of this part of the brain, as shown in Extended Data Fig. 6e. Furthermore, the cross-sectional area of the interstitial space was estimated to be 0.0083–0.18 μm² (Extended Data Fig. 7a–d). We assume that PHM

generates cyclical microdeformation in the rat RVLM (Extended Data Fig. 7e), thereby promoting interstitial-fluid movement.

Integrating these findings with previous reports on the property[42], flow velocity[43–45] and occupancy[46] of interstitial-fluid in the brain, we

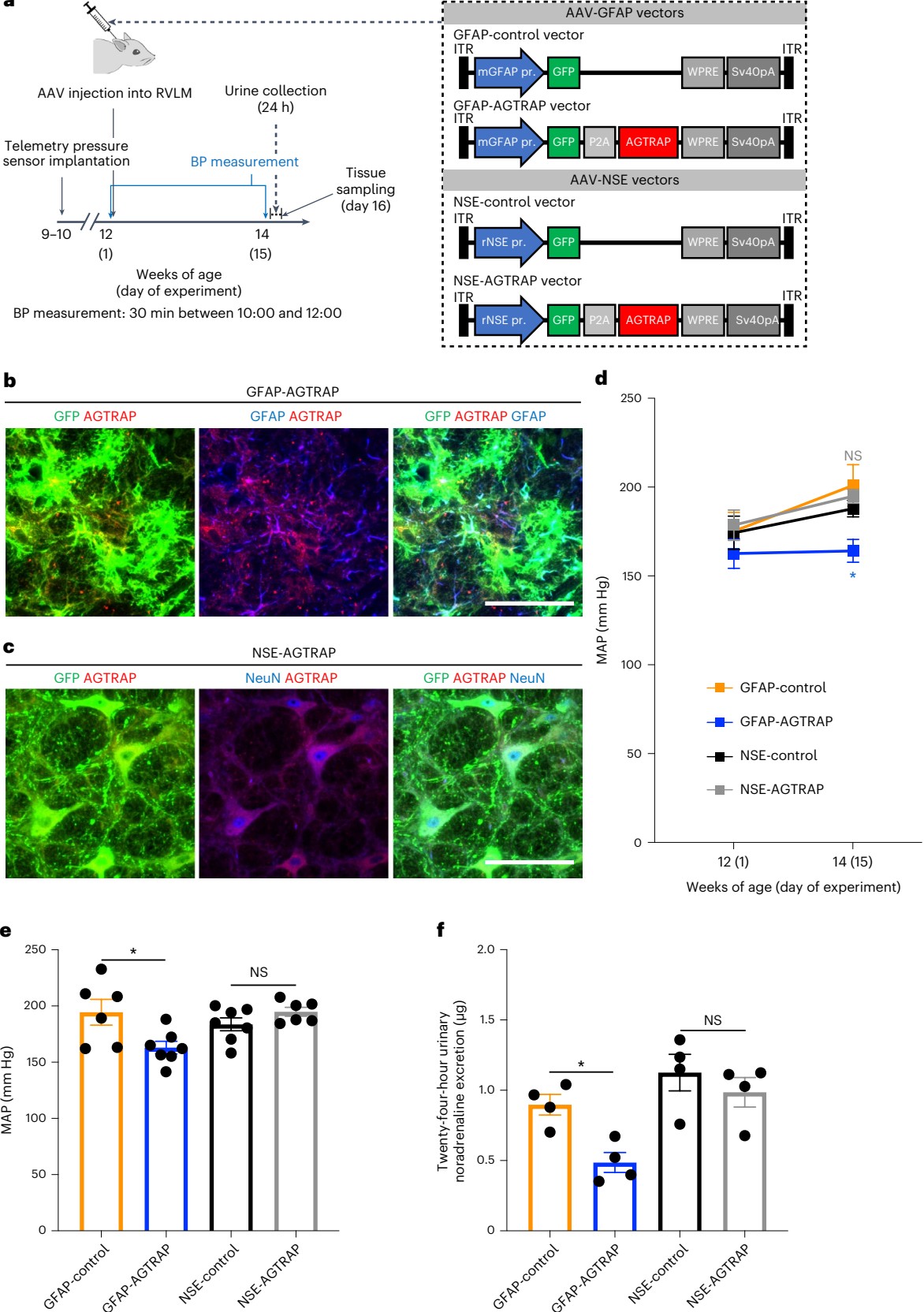

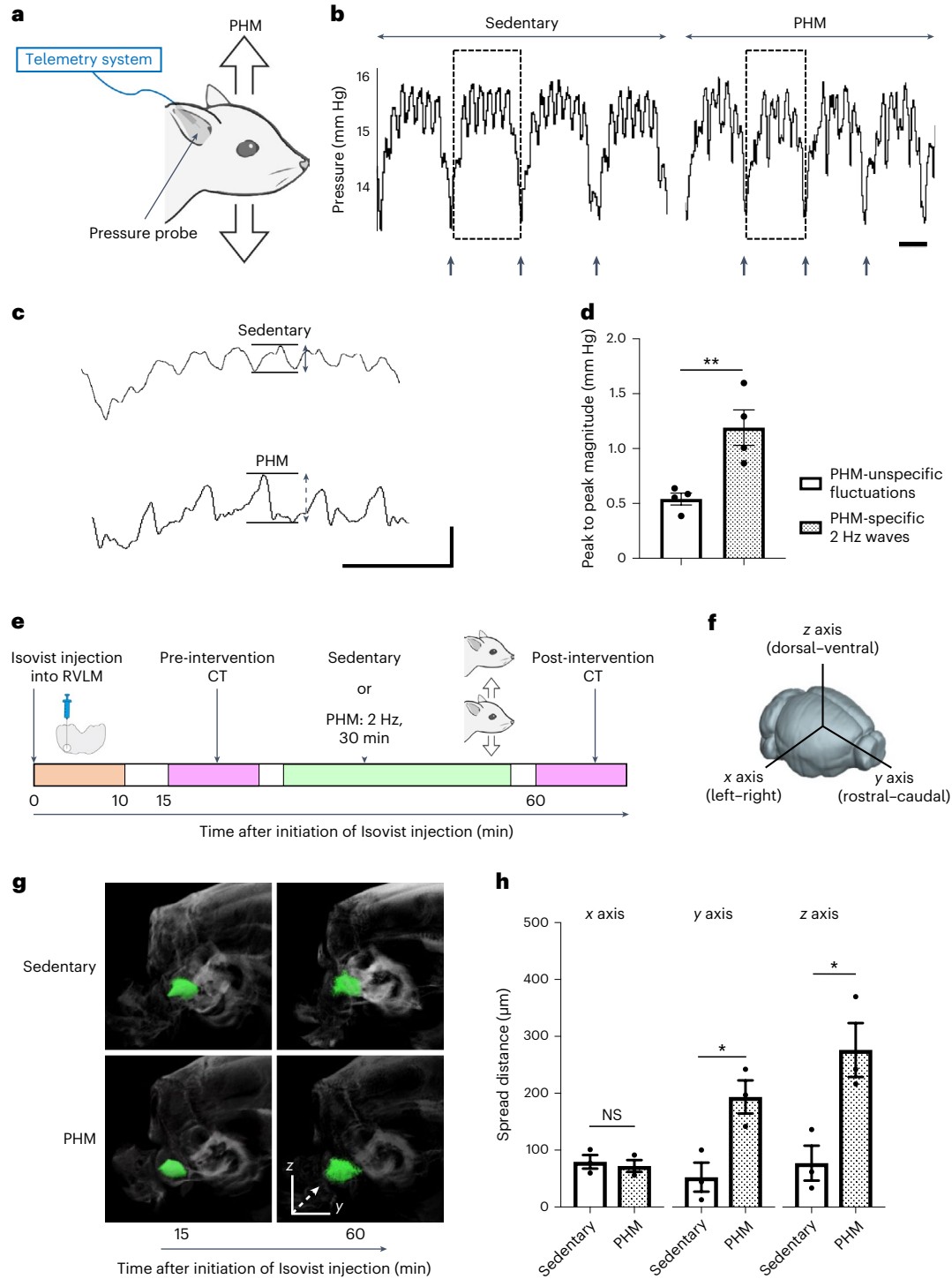

**Fig. 4 | PHM generates pressure waves of low amplitude, and facilitates interstitial-fluid movement (flow) in the rat RVLM. a**, Schematic of the pressure measurement in the rat RVLM. **b**, Representative pressure waves recorded in the rat RVLM during the sedentary condition and PHM. The arrows indicate the time of transition from inhalation to exhalation detected by simultaneous respiration monitoring. Scale bar, 1 s. Images are representative of four biologically independent experiments with similar results. **c**, Respiration-unsynchronized pressure changes. The respiration-synchronized pressure waves indicated by dotted rectangles in **b** are presented at a high magnification. Scale bars, 1 s (horizontal) and 1 mm Hg (vertical). Note that the 2 Hz pressure waves indicated by a two-headed dotted line arrow were specifically generated during PHM. **d**, The magnitude of PHM-specific and PHM-unspecific pressure changes unsynchronized with respiration. The peak-to-peak magnitudes indicated by two-headed arrows in **c** were quantified ($P = 0.0089$). $n = 4$ rats for each group, 10 segments analysed for each rat. **e**, Schematic of the experimental protocol for the μCT analysis of Isovist injected into the rat RVLM. **f**, Definition of the $x$ (left–right), $y$ (rostral–caudal) and $z$ (dorsal–ventral) axes used in this study. **g**, Representative Isovist spread presented on X-ray images. Isovist clusters are shown in green. The images in each row are from an individual rat, representative of three rats. The dashed arrow indicates the main direction of spreading in this sample. **h**, Quantification of Isovist spread along each axis ($P = 0.6666$ (left), $P = 0.0218$ (middle), $P = 0.0244$ (right)). $n = 3$ rats per group. Data are mean ± s.e.m. Statistical analysis was performed using unpaired two-tailed Student's $t$-tests; *$P < 0.05$, **$P < 0.01$.

calculated the average magnitude of interstitial-fluid-flow-derived shear stress exerted on rat RVLM cells during PHM (0.076–0.53 Pa; Supplementary Table 1).

## Fluid shear stress on astrocytes decreases AT1R expression in vitro

We next sought to determine what type of mechanical force was responsible for the PHM-induced decrease in AT1R expression in RVLM astrocytes (Fig. 1h,j) by in vitro experiments using cultured cells. Taking into account the approximate nature of our fluid shear stress calculation (Supplementary Table 1), we extensively examined whether the application of fluid shear stress or hydrostatic pressure change (HPC)—another type of mechanical intervention—modulated AT1R expression. On the basis of our calculation (Supplementary Table 1), we applied pulsatile fluid shear stress with an average magnitude of 0.05–0.7 Pa to cultured primary astrocytes, which were prepared from the astrocyte-GFP mice[47] (Extended Data Fig. 8a), using a system that we previously reported[23,24,41]. Quantitative PCR (qPCR) analysis revealed that application of fluid shear stress with at least 0.3 Pa magnitude (0.5 Hz, 30 min) significantly decreased AT1R expression in astrocytes for at least 24 h in an apparently magnitude-dependent manner (Fig. 5a and Extended Data Fig. 8b). By contrast, cyclical application of HPC, ranging from 1 to 40 mm Hg, did not significantly alter AT1R expression at ≤10 mm Hg and significantly increased AT1R expression at ≥20 mm Hg in cultured astrocytes (Fig. 5b). Consistently, immunostaining (Fig. 5c,d) and fluorescently labelled ligand (angiotensin II) binding (Fig. 5e,f) analyses of cultured astrocytes indicated that AT1R expression was significantly decreased by exposure to 30 min 0.7 Pa fluid shear stress. Collectively, fluid shear stress at magnitudes of less than 1 Pa, but not HPC, on astrocytes decreased AT1R expression in vitro. Notably, fluid shear stress application to Neuro2A cells, which exhibit neuronal phenotypes and morphology[48,49], did not decrease AT1R expression (Extended Data Fig. 8c–e).

The duration (>24 h) of fluid shear stress effects on AT1R expression in astrocytes (Fig. 5) poses a possibility of cumulative effects of fluid shear stress applied repeatedly at 24 h intervals. Nevertheless, 2 day PHM (30 min per day) alleviated the pressor and depressor responses to angiotensin II and AT1R blocker, respectively, injected into the RVLM of SHRSPs (Extended Data Fig. 9a–e), supporting the relevance of the relatively quick decrease in AT1R expression in our fluid shear stress experiments to our in vivo observations. Taken together, our in vitro findings are consistent with the notion that the fluid-shear-stress-mediated persistent decrease in AT1R expression is involved in the effects of daily PHM application on blood pressure (Fig. 1b,c) and AT1R expression in RVLM astrocytes (Fig. 1h,j) in SHRSPs. In contrast to PHM (Fig. 1b) and treadmill running[25,30,31], both of which required more than 2 weeks to decrease blood pressure in hypertensive rats, daily AT1R blocker administration has been reported to decrease the blood pressure of hypertensive rats in less than 1 week[50,51]. The decrease in AT1R signalling in RVLM astrocytes may take a considerably longer time to elicit its consequences on cardiovascular variables compared with the systemic RAS blockade[52]. Relatedly, PHM for 4 weeks in SHRSPs initiated during the plateau phase of their hypertension development (aged 21 weeks) did not significantly alter the blood pressure (Extended Data Fig. 10a–c), although it did decrease AT1R expression in RVLM astrocytes, but not neurons (Extended Data Fig. 10d–f). These findings imply that a complex mechanism links AT1R signalling in RVLM astrocytes to blood pressure regulation, and irreversible or refractory damage(s) may occur depending on various factors such as extended duration and aggravated seriousness of homeostasis-disrupting loads or stresses. For example, both vascular and renal functions in SHRSPs have been reported to be impaired in association with ageing (≥16 weeks) and the severity of hypertension (mean arterial pressure (MAP) ≥ 200 mm Hg)[53–56].

## Hindrance of interstitial-fluid movement by hydrogel introduction in the RVLM eliminates the ability of PHM to decrease AT1R expression in RVLM astrocytes and blood pressure in SHRSPs

To examine whether the interstitial-fluid movement in the RVLM mediated the effects of PHM on blood pressure and AT1R expression in the RVLM astrocytes in SHRSPs, we modulated the local interstitial-fluid dynamics. Following the procedure that we used to restrict the interstitial-fluid movement in the mouse PFC[24], we hindered the interstitial-fluid movement in situ by microinjecting mutually reactive polyethylene glycol (PEG) gel-precursor (pre-gel) solutions into the rat RVLM (Fig. 6a). Injected pre-gel solution spread over the rat RVLM, leading to hydrogel formation in the interstitial space in situ (Supplementary Fig. 1a). We previously showed that hydrogel introduction hinders only the fluid movement and does not restrict the diffusion of small molecules inside the gel[24,57]. Consistent with this, hydrogel introduction did not delay or attenuate the pressor and depressor responses to angiotensin II and angiotensin II antagonist, respectively, injected into the RVLM (Supplementary Fig. 1b–f), indicating rapid solute diffusivity through the hydrogel.

Hydrogel introduction into the bilateral RVLMs eliminated the ability of PHM to decrease blood pressure (Fig. 6b (black and orange lines) and 6c (columns 2 and 3)), urinary noradrenaline excretion (Fig. 6d (columns 2 and 3)) and AT1R expression in the RVLM astrocytes (Fig. 6e (rows 2 and 3) and 6f (columns 2 and 3)) in SHRSPs. By contrast, hydrogel introduction increased blood pressure (Fig. 6b (blue and orange lines) and 6c (columns 1 and 3)), noradrenaline excretion (Fig. 6d (columns 1 and 3)) and AT1R expression in RVLM astrocytes (Fig. 6e (rows 1 and 3) and 6f (columns 1 and 3)) of SHRSPs that were treated with PHM. The AT1R expression in RVLM neurons of SHRSPs remained unaltered irrespective of the combination of PHM and hydrogel introduction (Fig. 6e,g). These results suggest that hydrogel introduction in the RVLM disrupts the mechanism mediating the PHM-induced decrease in blood pressure, noradrenaline excretion and AT1R expression in RVLM astrocytes of SHRSPs. Consistent with this, the antihypertensive effect of daily treadmill running in SHRSPs was eliminated by hydrogel introduction in the RVLM (Supplementary Fig. 2a–c), supporting our hypothesis about the mechanism underlying the antihypertensive effects of exercise.

As was the case with the mouse PFC[24], hydrogel introduction (Supplementary Fig. 3a) did not affect the overall cell number or apoptosis (Supplementary Fig. 3b,c), survival or apoptosis of the RVLM astrocytes (Supplementary Fig. 3d,e) and neurons (Supplementary Fig. 3f,g), and the expression of pro-inflammatory cytokines (TNF-α and IL-1β) (Supplementary Fig. 3h) in the RVLM. Furthermore, hydrogel introduction in the rat RVLM did not significantly alter the intramedullary pressure (Supplementary Fig. 3i). Collectively, the loss of the PHM effects by hydrogel introduction into the bilateral RVLMs in SHRSPs (Fig. 6b–f) probably results from the hydrogel-mediated alteration in interstitial-fluid dynamics, rather than the decreased cell viability and/or enhanced inflammatory responses caused by the impaired nutrient supply, removal of metabolic wastes or persistent PEG existence/contact.

Consistent with these results supporting the importance of interstitial-fluid movement (flow) in the RVLM of SHRSPs, both their blood pressure and heart rate remained unchanged during the transition from before to after the initiation of PHM (Supplementary Fig. 4), precluding the response of baroreceptors, either carotid or aortic, to PHM. Furthermore, the activity of the aortic depressor nerve, which transmits afferent signals from baroreceptors and chemoreceptors located in the aortic arch[58], also remained unaltered from before to after PHM initiation (Supplementary Fig. 4). Thus, the baroreceptor response does not seem to be responsible for the antihypertensive effects of PHM on SHRSPs.

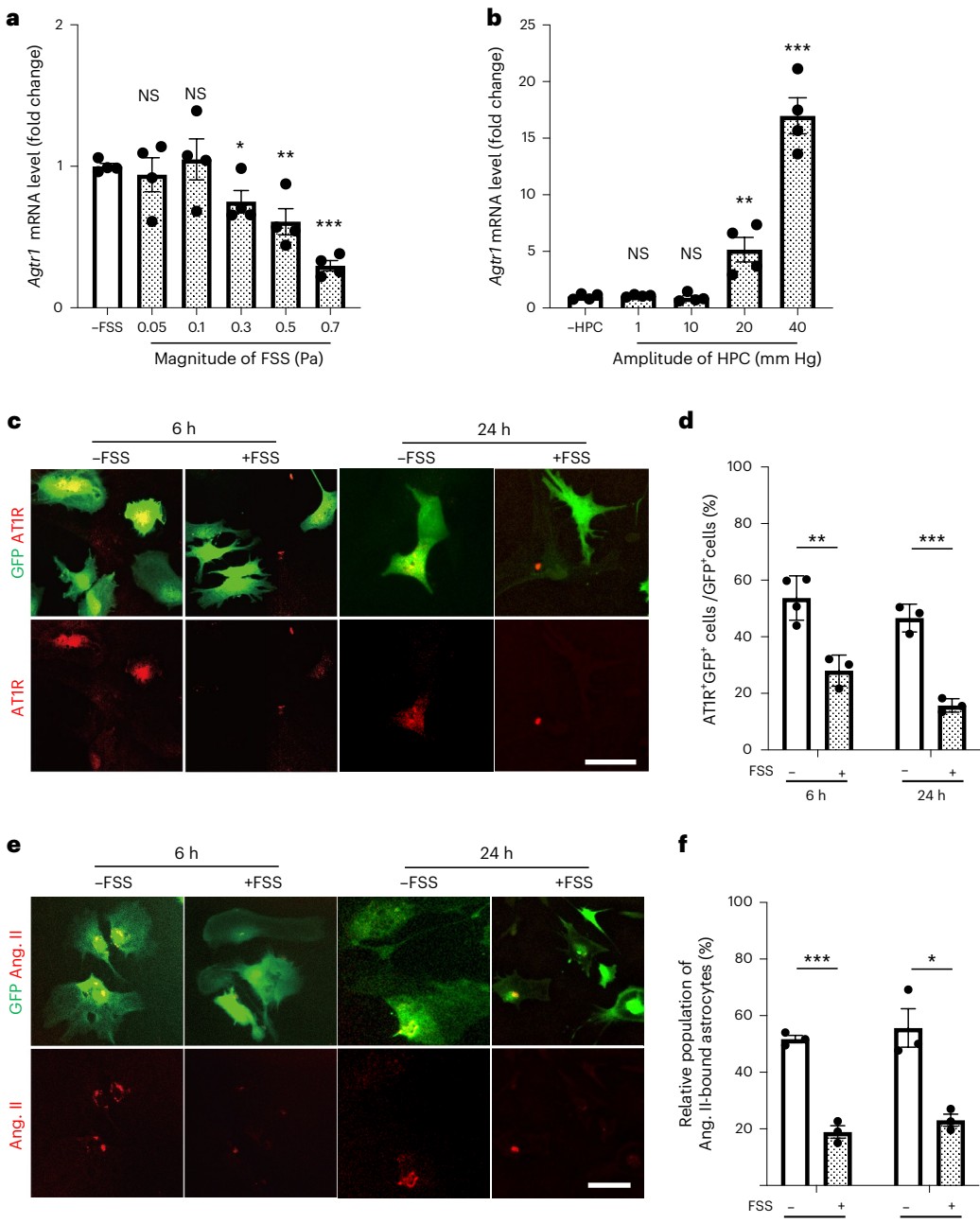

**Fig. 5 | Fluid shear stress on cultured astrocytes decreases their AT1R expression and angiotensin-II-binding potential in vitro. a–d**, AT1R expression in cultured astrocytes with or without exposure to fluid shear stress or HPC. Astrocytes prepared from the astrocyte-GFP mice, either left unexposed or exposed to pulsatile fluid shear stress (average 0.05–0.7 Pa, 0.5 Hz, 30 min) (**a**) or cyclical HPC (1–40 mm Hg, 0.5 Hz, 30 min) (**b**) were solubilized 24 h after the termination of intervention, and analysed using qPCR. *Agtr1* mRNA expression levels were normalized to *Gapdh* expression and scaled with the mean value from control samples (cells left unexposed to fluid shear stress or HPC) set as 1. **c**, Microscopy images of anti-AT1R (red) and anti-GFP (green) immunostaining of cultured astrocytes that were either left unexposed or exposed to pulsatile fluid shear stress (0.7 Pa, 0.5 Hz, 30 min) and fixed 6 or 24 h after the intervention. Images are representative of three or four biologically independent experiments with similar results. Scale bar, 50 μm. **d**, Relative population of AT1R+GFP+ double-

positive cells (cells were quantified as a ratio to total GFP+) in each sample. **e,f**, The effect of fluid shear stress on the angiotensin-II-binding potential of astrocytes. Cultured astrocytes were either left unexposed or exposed to fluid shear stress as described in **a**, **c** and **d**. Six or twenty-four hours after the cessation of the 30 min application of fluid shear stress (0.7 Pa, 0.5 Hz), cells were analysed using a fluorescent angiotensin-II-binding assay. **e**, Microscopy images. Images are representative of three biologically independent experiments with similar results. Scale bar, 50 μm. **f**, GFP+ cells with punctate red fluorescence (TAMRA–angiotensin-II-bound astrocytes) were quantified as the ratio (%) to total GFP+ cells in each sample. The images in each column of **c** and **e** are from an individual sample. Data are mean ± s.e.m. Statistical analysis was performed using unpaired two-tailed Student's *t*-tests; *$P < 0.05$, **$P < 0.01$, ***$P < 0.001$. Details of the statistical analyses are provided in the Supplementary Information.

## VOCR lowers the blood pressure of adult humans with hypertension

The results from our animal experiments reveal the antihypertensive effect of the mechanical accelerations generated in the head during treadmill running at a moderate velocity. This prompted us to test whether the application of mechanical intervention to the head lowered the blood pressure in people with hypertension. We observed that light jogging or fast walking (locomotion at a velocity of 7 km h⁻¹)

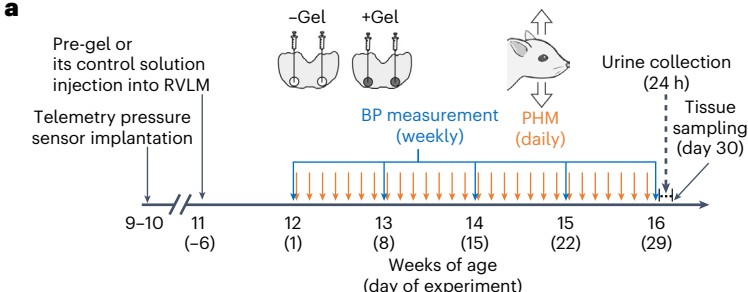

**Fig. 6 | Hydrogel introduction eliminates the decreasing effects of PHM on blood pressure and AT1R expression in RVLM astrocytes of SHRSPs.**
**a**, Schematic of the experimental protocol to analyse the effects of PHM with and without PEG hydrogel introduction in the bilateral RVLMs in SHRSPs. PHM was applied daily for a consecutive 28 days. **b–d**, Time courses (**b**) and values on day 29 (**c**) of MAP, and 24 h urinary noradrenaline excretion (**d**) in SHRSPs that were treated with various combinations of daily PHM application and hydrogel introduction in the bilateral RVLMs. Note the absence of significant differences in blood pressure (**b** and **c**) and urinary noradrenaline excretion (**d**) in SHRSPs with hydrogel-introduced RVLMs (+gel) between the groups with and without PHM. **e**, Micrographic images of anti-GFAP (green), anti-AT1R (red) and anti-NeuN (blue) immunostaining of the RVLM in SHRSPs that were treated with various combinations of hydrogel introduction in the bilateral RVLMs and 4 week PHM application. The arrows point to anti-AT1R immunosignals that overlap with anti-GFAP, but not anti-NeuN, immunosignals in the merged images. Scale bar, 50 μm. Images are representative of four rats. **f,g**, Quantification of AT1R-positive astrocytes (**f**) and neurons (**g**) in the RVLM. Note the absence of a significant difference in the ratio of AT1R⁺ astrocytes in SHRSPs with hydrogel-introduced RVLMs between the groups with and without PHM (**f**, columns 2 and 3). In total, 50 NeuN⁺ cells and 100 GFAP⁺ cells were analysed for each rat. Data are mean ± s.e.m. Statistical analysis was performed using two-way repeated-measures ANOVA (**b**) or one-way ANOVA (**c**, **d**, **f** and **g**) with Tukey's post hoc multiple-comparison test; *$P < 0.05$, ***$P < 0.001$. Details of the statistical analyses are provided in the Supplementary Information.

typically produces 2 Hz vertical acceleration waves with an amplitude of around 1.0*g* in the person's head (Supplementary Fig. 5a (top)). It was difficult to apply vertical forces to only the heads of the human participants safely without leading to considerable discomfort and distress. We therefore constructed a chair that could vertically oscillate at a frequency of 2 Hz (Supplementary Fig. 5b and Supplementary Video 2) and produce around 1.0*g* acceleration waves in the head of the occupant (Supplementary Fig. 5a (bottom)), although the other body parts were also exposed to cyclical vertical movements in this system. The waveform of the vertical acceleration (Supplementary Fig. 5a (bottom)) was also determined considering the participant's comfort, in addition to technical issues.

Given that previous reports regarding antihypertensive effects of aerobic exercise typically recommend at least 3–4 days per week (frequency) and at least 30 min per session or day (duration)[2], we set our regimen of vertically oscillating chair riding (VOCR) as 3 days per week (Monday, Wednesday and Friday, unless needed to assign otherwise for particular reasons such as public holidays) and 30 min per day. Our pilot study following protocol 1, in which we simply compared the participants' blood pressure and heart rate before and after VOCR for 4 weeks (12 times) (Supplementary Fig. 5c), showed that VOCR decreased the blood pressure of people with hypertension (Supplementary Fig. 5d).

We next conducted a human study of protocol 2, in which we followed the changes in the participants' blood pressure and heart rate minutely (Supplementary Fig. 6a). Encouraged by the positive results from the study of protocol 1, we adopted the same VOCR regimen as to its frequency (3 days per week) and duration (30 min per day). To detect the trends of blood-pressure and heart-rate changes more reliably by reducing the influences from interday variabilities, we followed and analysed the 'value of the week' (Methods). Serial blood sampling was performed for the participants to measure plasma catecholamines (adrenaline, noradrenaline and dopamine) and renin activity, as well as serum aldosterone and C-reactive protein (CRP) before and after the intervention period (Supplementary Fig. 6a). To conduct the second blood sampling on the next day of the last bout of VOCR, the intervention period was extended from 4 weeks (total of 12 times, typically 26 days) to 4.5 weeks (total of 14 times, 30–31 days) as blood sampling could not be done during weekends at our hospital. Systolic blood pressure (SBP), diastolic blood pressure (DBP) and MAP (value of the week) immediately after the intervention period significantly decreased compared with those immediately before the intervention period (Fig. 7a). Notably, the post-intervention follow-up showed that the blood-pressure-lowering effect apparently persisted for 4 weeks, but not for 5 weeks, after the last bout of VOCR (Fig. 7b). Similar to in our animal study, we did not observe significant changes in heart rate due to the VOCR intervention (Fig. 7a,b and Supplementary Fig. 5d). Significant differences were not detected in the blood levels of catecholamines, aldosterone, renin activity and CRP between before and after the VOCR intervention (Supplementary Fig. 6b).

The antihypertensive effect of VOCR observed in the participants of protocols 1 and 2 (Supplementary Table 2) prompted us to proceed to protocol 3, in which we examined whether non-oscillating chair riding (NOCR), a control for VOCR, affected blood pressure in adult humans with hypertension (Supplementary Fig. 7a). We asked the participants in protocol 3 for their agreement to continuous recording of beat-by-beat blood pressure and interbeat (*R*–*R*) intervals (RRIs). In those who agreed to these measurements, we analysed the SBP and RRI variabilities (Methods). Whereas NOCR for 4.5 weeks did not significantly affect the blood pressure of the participants (Fig. 8a (top)), VOCR significantly lowered it (Fig. 8a (bottom)), as in the studies following protocols 1 and 2. NOCR did not significantly alter the low-frequency (LF) power in SBP variability (Fig. 8b) or the ratio of LF/high frequency (HF) power (LF/HF ratio) in RRI variability (Fig. 8c). By contrast, VOCR significantly decreased the former (Fig. 8b) and elicited a decreasing

tendency in the latter (Fig. 8c). These findings indicate that VOCR decreases the vascular sympathetic nerve activity[59,60] and its dominance over the cardiac parasympathetic activity[59,61]; nevertheless, there remains some controversy regarding the use of the LF/HF ratio in RRI variability as an appropriate relevant indicator[62,63].

Collectively, our human studies suggest that VOCR, which reproduces mechanical accelerations in the head during light jogging or fast walking, has an antihypertensive and sympathoinhibitory effect in people with hypertension. Although our animal studies were conducted in only male rats, VOCR had antihypertensive effects in both male and female human participants (Supplementary Fig. 7b). Importantly, in the 33 VOCR participants (Supplementary Tables 2 and 3), no apparent adverse events, including motion sickness and low back pain, were observed or manifested in relation to the VOCR intervention.

## Discussion

The antihypertensive effects of physical exercise can involve a variety of events and processes, such as redox homeostasis and inflammation[30,31], in various tissues and organs, including the vessels/endothelium[64–66], skeletal muscle[67] and brain[31,68,69]. However, direct mechanical effects on the brain do not seem to have been considered in these previous studies. Here, PHM, which reproduced mechanical accelerations generated in the head during treadmill running, enabled us to examine the mechanical effects triggered by physical activity.

Whereas AT1R signalling in both neurons and astrocytes in the RVLM have been reported to be involved in regulating blood pressure[70,71], we observed that AT1R expression in RVLM astrocytes was increased in SHRSPs compared with in WKY rats (Fig. 1j). By contrast, AT1R expression in RVLM neurons was comparable between WKY rats and SHRSPs (Fig. 1i), although AT1R expression in RVLM neurons has been shown to have an important role in other animal model(s) of hypertension[70]. Together with the decreases in blood pressure and urinary noradrenaline excretion in SHRSPs in which RVLM astrocytes were transduced with the *Agtrap* gene (Fig. 3d–f), the intensity of AT1R signalling in RVLM astrocytes seems to be critically involved in the pathogenesis of hypertension and sympathetic hyperactivity in SHRSPs.

PHM for 4 weeks decreased urinary noradrenaline excretion and AT1R expression in RVLM astrocytes in SHRSPs to levels almost equivalent to those of WKY rats (Fig. 1f,j). However, PHM only partially alleviated the development of hypertension in SHRSPs (Fig. 1b,c) to an extent similar to the antihypertensive effects of treadmill running previously reported[25,28,30] or observed in this study (Supplementary Fig. 2b). Thus, it is evident that factors other than AT1R signalling in RVLM astrocytes also contribute to the pathogenesis of essential hypertension.

AT1R expression in the cultured astrocytes decreased on the application of fluid shear stress (Fig. 5a,c–f). This was consistent with our findings that PHM and treadmill running decreased AT1R expression in the RVLM astrocytes in SHRSPs (Fig. 1j and Extended Data Fig. 4c). However, AT1R expression level in the RVLM astrocytes was low in WKY rats even without PHM (Fig. 1j), and this may raise a concern regarding the physiological relevance of our in vitro fluid-shear-stress experiments using cultured astrocytes. Still, it has been reported that cultured astrocytes typically exhibit increased reactivity, and do not fully recapitulate the physiological astrocytes in vivo[72]. We suggest that the fluid-shear-stress-induced decrease in AT1R expression in cultured astrocytes that we observed represents the physiological functions of astrocytes, despite that their increased basal AT1R expression may relate to the non-physiological aspects of two-dimensional (2D) culture on stiff substrates (culture plastics). Cells in static culture are exposed to a complete absence of fluid shear stress, which may not be physiologically realized in vivo. Previous reports describe increased extracellular fluid in the brains of people with hypertension[73] and altered dynamics of the intracerebral interstitial fluid of SHRs[74]. Aberrant regulation of the function of RVLM astrocytes that relates to altered interstitial-fluid

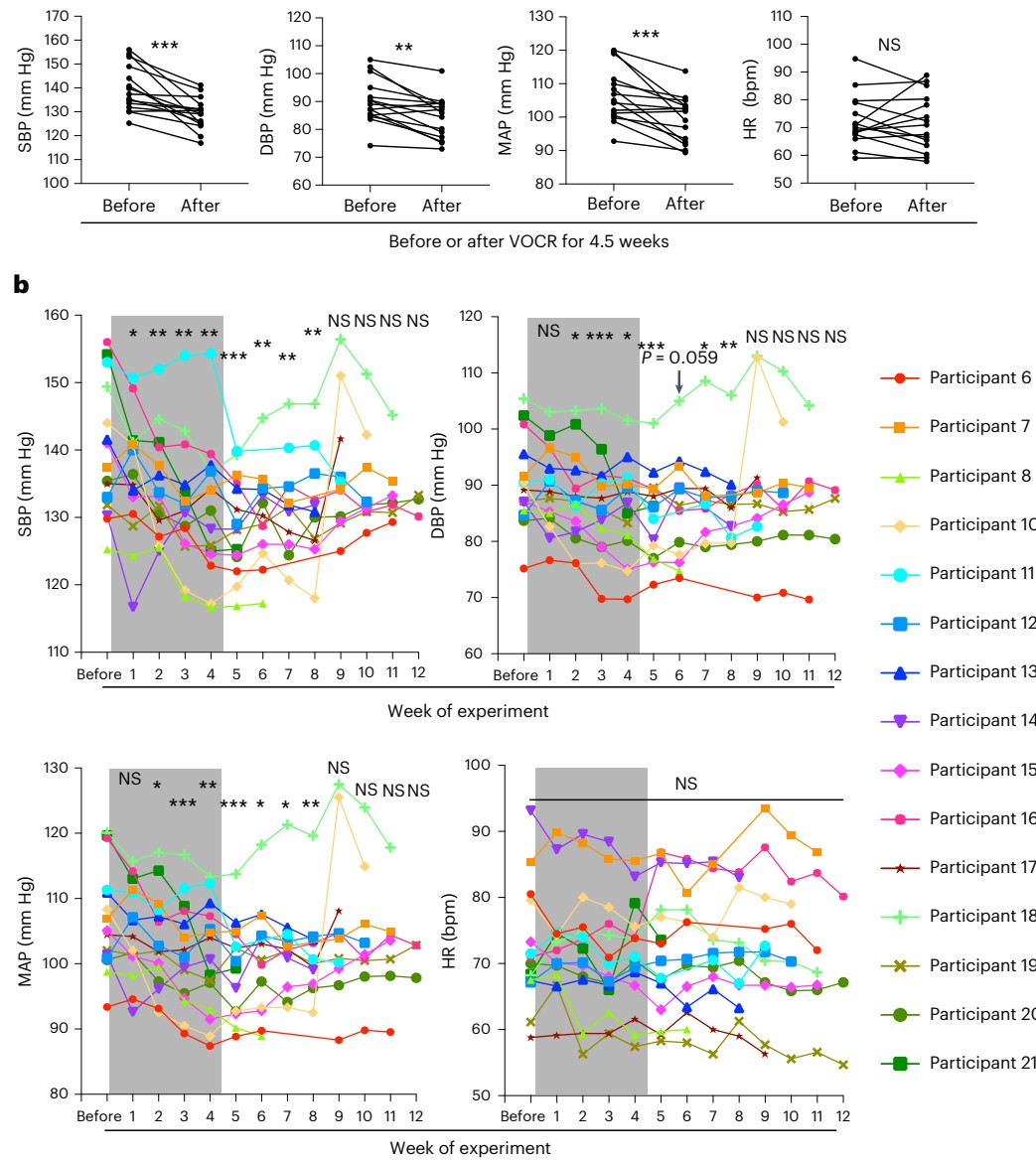

**Fig. 7 | VOCR has an antihypertensive effect on adult humans with hypertension. a**, The SBP, DBP, MAP and heart rate value of the week immediately before and after 4.5 week VOCR in the study of protocol 2 ($P = 0.0005$ (SBP), $P = 0.0011$ (DBP), $P = 0.0008$ (MAP), $P = 0.7845$ (heart rate); $n = 15$). **b**, The corresponding trajectories of the participants and statistical analysis of blood pressure and heart rate in the study of protocol 2. The grey rectangles indicate the VOCR intervention periods (4.5 weeks). The colours and symbols of individual lines correspond to individual participants (right), excluding participant 9 (Supplementary Table 2). Each value of the week was statistically compared with that of the week immediately before the initiation of VOCR intervention. Statistical analysis was performed using paired two-tailed Student's $t$-tests; *$P < 0.05$, **$P < 0.01$, ***$P < 0.001$. Details of the statistical analyses are provided in the Supplementary Information.

movement-derived fluid shear stress may underlie the pathogenesis of essential hypertension.

PHM did not significantly alter AT1R expression in the RVLM neurons of SHRSPs (Fig. 1i), and fluid shear stress did not decrease AT1R expression in cultured Neuro2A cells (Extended Data Fig. 8c–e). However, we do not suspect that these results represent the absence of sensitivity of neurons to fluid shear stress or other types of mechanical stimulation, particularly because we observed the PHM- and fluid-shear-stress-induced internalization of 5-HT$_{2A}$ receptor expressed in mouse PFC neurons and Neuro2A cells, respectively[24]. Alternatively, we speculate that PHM and fluid shear stress may mitigate the hyperexpression of AT1R related to the aforementioned pathological status of RVLM astrocytes in SHRSPs or to the unphysiological nature of cultured

astrocytes. Consistent with this notion, PHM did not significantly affect AT1R expression in RVLM astrocytes in normotensive WKY rats (Fig. 1j).

Consistent with the lack of strict cell specificity in many of cellular responses to mechanical forces[75], the fluid-shear-stress-induced decrease in AT1R expression, which was reported in vascular cells[19], was also observed in cultured astrocytes. We speculate that there may be common homeostasis regulatory mechanisms at the cellular level that involve fluid flow-derived shearing forces.

Given the time lag from the PHM-induced decrease in AT1R signalling in the RVLM astrocytes (within 2 days; Extended Data Fig. 9) to the decrease in basal blood pressure in SHRSPs (3 weeks or longer; Fig. 1b), it is unlikely that there is a direct connection between them. Alternatively, the link from AT1R signalling in RVLM astrocytes to the

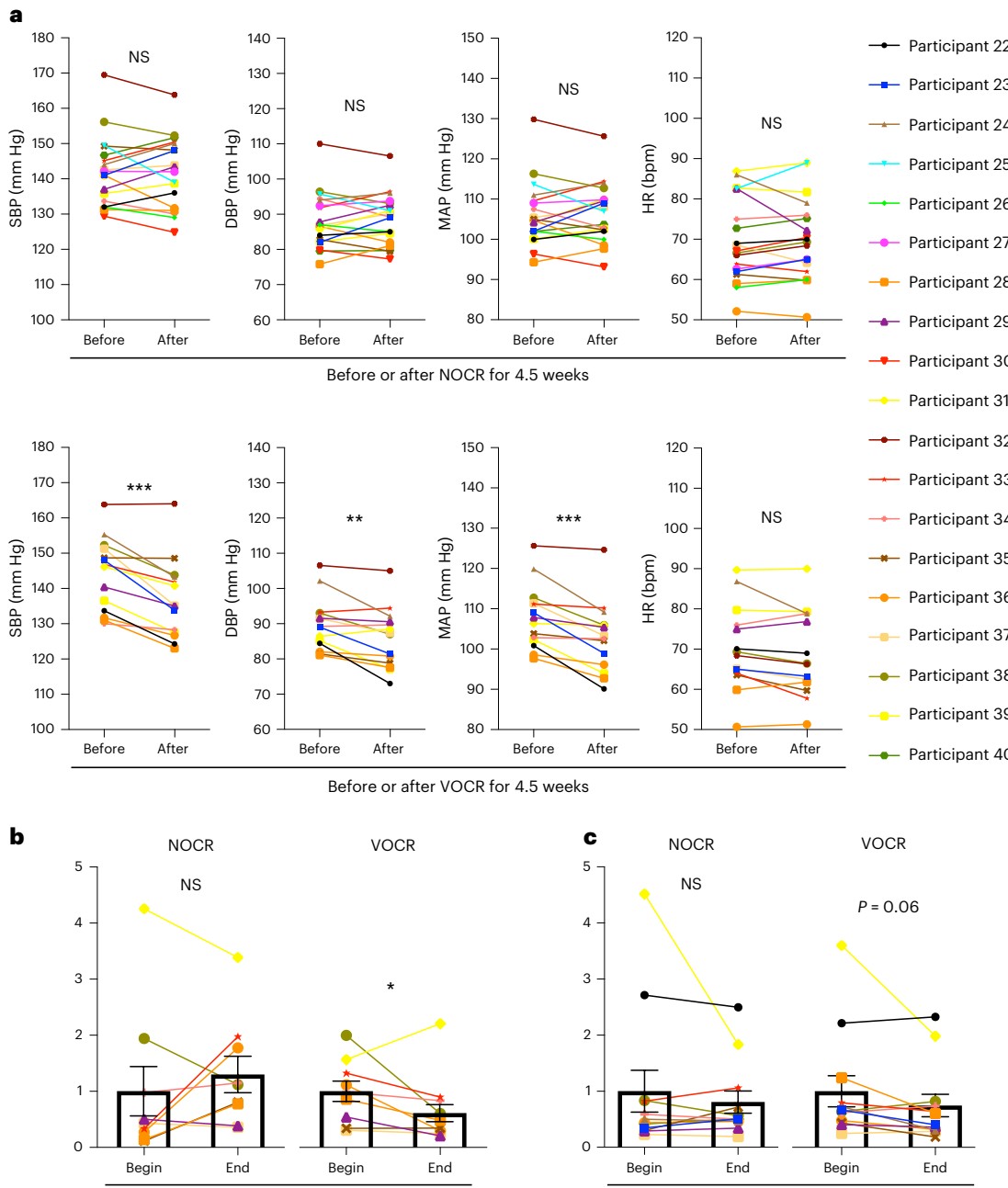

**Fig. 8 | VOCR, but not NOCR, has an antihypertensive and sympathoinhibitory effect in adult humans with hypertension. a**, The SBP, DBP, MAP and heart rate value of the week immediately before and after 4.5 week NOCR (top) and VOCR (bottom) in the study of protocol 3 (NOCR: $P = 0.9148$ (SBP), $P = 0.6597$ (DBP), $P = 0.7502$ (MAP), $P = 0.9002$ (heart rate), $n = 19$; VOCR: $P = 0.0001$ (SBP), $P = 0.0051$ (DBP), $P = 0.0006$ (MAP), $P = 0.0867$ (heart rate), $n = 14$). **b,c**, LF power in SBP variability (**b**) and the LF/HF ratio in RRI variability (**c**) at the beginning

(begin) and end (end) periods of intervention (NOCR and VOCR) scaled with the mean value from the beginning (the left column in each graph) set as 1 ($P = 0.492$, $n = 10$ (NOCR); $P = 0.016$, $n = 12$ (VOCR) (**b**); $P = 0.969$, $n = 12$ (NOCR); $P = 0.063$, $n = 12$ (VOCR) (**c**)). Data are mean ± s.e.m. Statistical analysis was performed using paired two-tailed Student's $t$-tests (**a**) or Wilcoxon signed-rank tests (**b** and **c**); *$P < 0.05$, **$P < 0.01$, ***$P < 0.001$.

basal blood pressure is presumably comprised of slow or chronic multifactorial processes. It has been reported that the hypothalamic paraventricular nucleus (PVN) integrates the signals concerning the factors that affect sympathetic activity and blood pressure, such as angiotensin II, proinflammatory cytokines and reactive oxygen species (ROS), from the basal forebrain, and sends the information to the RVLM[76–79]. AT1R signalling induces proinflammatory processes and ROS production in astrocytes[80]. Collectively, we assume that the persistent increase in AT1R expression in RVLM astrocytes of SHRSPs (for example, compare columns 1 and 3 in Fig. 1j) gives rise to sustained or chronic inflammation and oxidative stress in the aforementioned brain regions that are involved in the sympathetic activity and blood pressure regulation. Owing to the chronic nature of the increased AT1R signalling in the RVLM astrocytes in SHRSPs, the consequential inflammation and oxidative stress are also chronic, involving multiple factors but lacking the potential of quick responsive changes. As a result, the PHM-induced decrease in AT1R signalling in the RVLM astrocytes may not promptly lead to a decrease in the sympathetic outflow. Such a notion conforms

to a previous report describing the time course of exercise-induced attenuation of inflammation and oxidative stress in the PVN followed by the decrease in the basal blood pressure in SHR[31]. It also agrees with the involvement of multiple pro-inflammatory and anti-inflammatory and pro-hypertensive and anti-hypertensive factors in the PVN and lamina terminalis in the effects of exercise in hypertension-induced rats[69]. Furthermore, in contrast to an acute change in sympathetic activity, the consequence of a slow and moderate decrease in steady-state sympathetic outflow may involve relatively slow or time-consuming vascular responses and processes such as remodelling[81], which has been reported to be positively modulated by exercise[64,65].

On the basis of our hypothesis concerning the similarity in the pathogenesis of high blood pressure between human essential hypertension and SHRSPs, we conducted human studies in which we intended to reproduce the mechanical accelerations in the head that lowered blood pressure in SHRSPs. Although the mechanism behind the apparent antihypertensive effect of VOCR remains to be determined, the significant role of interstitial-fluid dynamics in the RVLM, which we demonstrated in our animal experiments, might be shared between humans and rats or other animals (Supplementary Fig. 8). Whereas plasma catecholamine levels were not significantly changed by the VOCR intervention in the human study of protocol 2 (Supplementary Fig. 6b), it is possible that the urinary noradrenaline measures collected over 24 h in our rat PHM experiments (Figs. 1f,3f and 6d) enhanced our ability to capture the sympathetic nerve activity under common ambulatory conditions[82]. The findings from the study of protocol 3 suggest a sympathoinhibitory effect of VOCR (Fig. 8b,c).

Physical exercise is broadly useful to maintain human health. Many aerobic exercises, including walking and running, involve impact-generating bodily actions that create sharp accelerations in the head at foot contact with the ground. Furthermore, the antihypertensive effects of PHM in the rostral–caudal direction (Extended Data Fig. 2), or with a lower peak magnitude (0.5$g$) or frequency (0.5 Hz) (Extended Data Fig. 3), may be relevant to the decrease in blood pressure caused by other forms of exercise such as swimming and bicycle riding[83,84]. We speculate that the beneficial effects of various types of exercise on a variety of brain-function-related diseases and health disorders may rely, at least in part, on modest changes in mechanical-stress distribution in the brain, which may prompt optimal fluid shear stress on intracerebral nervous cells. To the contrary, alterations in interstitial-fluid-movement-derived shear stress may underlie the pathogenesis of various brain disorders, particularly those related to physical inactivity or ageing.

## Limitations of the study

We used SHRSP as an animal model of essential hypertension. There are several differences between hypertension in SHRSPs and humans. SHRSPs develop hypertension in young adulthood, but not in middle age, as is characteristic of humans. Furthermore, SHRSPs cannot model environmental influences that trigger human hypertension, including increased salt intake, obesity and physical inactivity. Nevertheless, SHRSPs enabled us to obtain a chronic stable hypertensive condition with minimal interindividual variation, but without difficult or life-threatening technical interventions. We used only male rats for our animal studies, although sex is an important variable for nearly all diseases, including hypertension. This was because we intended to preclude or minimize the potential influence of oestrogen and progesterone, both of which basically act protectively on cardiovascular systems, including the heart and endothelium. Perhaps for the same reason, in many of or even most animal experiments in previous studies investigating the pathogenesis of cardiovascular diseases, male animals have been used unless there is particular reason to analyse female animals. In particular, we intended to be consistent with previous studies in which the antihypertensive effects of treadmill running in male SHRs or SHRSPs were demonstrated[25,28,30,31]. However, as we did not aim to

investigate only male-specific matters, we included human participants of both sexes. Given that VOCR had antihypertensive effects in both male and female participants (Supplementary Fig. 7b), we anticipate that the mechanical regulation of RVLM astrocytes demonstrated in this study is not specific to male individuals.

The decreases in blood pressure and urinary noradrenaline excretion with the exogenous expression of AGTRAP (Fig. 3d–f) indicate the causal or hierarchical relationship between AT1R signalling in RVLM astrocytes and blood pressure in SHRSPs. This supports the notion that the reduced AT1R expression in RVLM astrocytes mediates the blood-pressure-decreasing effects of PHM (Fig. 1b,c,h,j) and treadmill running (Extended Data Fig. 4a,c and Supplementary Fig. 2b,c). However, the decrease in blood pressure with the exogenous expression of AGTRAP in RVLM astrocytes was relatively short lasting (Extended Data Fig. 5g). Considering the persistent AAV-mediated protein expression in brain cells for a few months or longer[85], it is unlikely that the short-lasting decrease in blood pressure with the exogenous expression of AGTRAP in RVLM astrocytes results from the short duration of AGTRAP overexpression. Relatedly, it has been reported that the transgenic mice in which AGTRAP was specifically overexpressed in the renal tubules did not present a significant decrease in basal blood pressure, although these mice did present a reduced blood-pressure-increasing response to pressor-dose infusion of angiotensin II[38]. By contrast, the deletion of endogenous angiotensin II type 1a receptor in the renal proximal tubules decreased the basal blood pressure in mice[39,40], indicating the persistent connection between attenuated AT1R signalling in the renal tubules and basal blood-pressure lowering. Considering these previous reports together with the short duration of the basal blood-pressure lowering as a result of the exogenous expression of AGTRAP in RVLM astrocytes (Extended Data Fig. 5g), we speculate that the decreasing effect of AGTRAP overexpression on steady-state AT1R signalling is relatively short lasting due to a compensatory or neutralizing mechanism. This issue makes it difficult to reasonably combine the AAV-mediated astrocyte-specific AGTRAP overexpression in the RVLM with PHM or treadmill running, of which the antihypertensive consequence becomes significant in ≥3 weeks (Fig. 1b and Supplementary Fig. 2b). Further investigation is needed to decipher the details of the connection between AT1R signalling in RVLM astrocytes and basal blood-pressure regulation in the context of hypertension.

We estimated the magnitude of the fluid shear stress exerted on the cells in the rat RVLM to be 0.076−0.53 Pa (Supplementary Table 1). However, a fluid shear stress of 0.1 Pa, which is within this range, did not significantly alter AT1R expression in cultured astrocytes (Fig. 5a). 2D cell culture experiments are non-physiological, particularly in light of the 3D nature of the microenvironments in vivo, and do not entirely recapitulate physiological conditions. Furthermore, our calculation of the fluid-shear-stress magnitude in vivo is approximate. We speculate that the aforementioned discrepancy derives from the approximate nature of our estimation of fluid shear stress in vivo and/ or the non-physiological nature of the in vitro fluid-shear-stress experiments using cultured cells. Considering this issue, we have made our in vitro studies extensive by testing not only fluid shear stress but also HPC with various magnitudes and amplitudes (Fig. 5). Although further study is required to determine the magnitude of mechanical forces at the cell level generated during the application of PHM or treadmill running, and the consequent cellular responses in vivo more precisely, our experimental results support the notion that fluid shear stresses of lower than 1 Pa are responsible for the PHM-induced decreases in AT1R expression in RVLM astrocytes in rats. Due to the easy detachment of mouse cerebral-cortex-derived or hippocampus-derived primary neurons from the substrates by fluid shear stress, we tested Neuro2A cells, an alternative of cultured neuronal cells, which stably adhered to the substrates through fluid shear stress of magnitudes up to around 1 Pa (ref. 24).

At present, we do not have experimental data with which we can explain why PHM did not lower the blood pressure in SHRSPs during the plateau phase of their hypertension development (that is, at ≥21 weeks of age), although it attenuated AT1R expression in RVLM astrocytes (Extended Data Fig. 10). Regarding such disconnection of AT1R signalling in the RVLM astrocytes from the basal blood pressure control in SHRSPs, we speculate that it originates from irreversible or refractory organic changes in one or more elements of the aforementioned link between them and/or other factors affecting blood pressure. Chronic inflammation and oxidative stress are implicated in severe degenerative organic damage in various tissues and organs, including the blood vessels[86], kidneys[87] and brain[88]. In particular, it has been reported that hypertension can become irreversible when renal function is severely impaired[89]. From the apparently limited contribution of systemic RAS to the antihypertensive effects of exercise revealed by a meta-analysis[90], impairments in renal function may negatively affect blood pressure regulation fairly independently of PHM, and override its antihypertensive effect in SHRSPs. Consistent with our speculation about the involvement of impaired renal function in the lack of PHM effect in SHRSPs aged ≥21 weeks, severe kidney damage was observed in 20-week-old SHRSPs[91].

We did not comprehensively analyse the effects of PHM on brain functions, but focused on the study of the RVLM. PHM may modulate AT1R signalling in other brain regions that participate in the regulation of sympathetic nerve activity, including the anteroventral third ventricle, PVN and nucleus tractus solitarii[6,9]. It is technically difficult to specifically interfere solely with interstitial flow in the brain or in other tissues or organs in living animals. Nonetheless, introduction of hydrogel into the RVLM of SHRSPs eliminated the decreasing effects of PHM on blood pressure and urinary noradrenaline excretion (Fig. 6b–d) as well as the antihypertensive effect of treadmill running (Supplementary Fig. 2), supporting the critical role of the RVLM. As hydrogel may exert yet unknown effects, experiments of hydrogel introduction may not entirely prove the contribution of interstitial-fluid movement. For example, introduction of hydrogel may alter the stiffness and elasticity of the extracellular matrix, which is known to affect neurological physiology, pathology and development[92]. Regardless, from the unaltered cell survival and apoptosis, pro-inflammatory protein expression and pressure in the hydrogel-introduced RVLM in SHRSPs (Supplementary Fig. 3), substantially detrimental or favourable processes are unlikely to be responsible. Given the involvement of ROS in hypertension in SHRSPs[93] and the unaltered blood pressure in hydrogel-introduced SHRSPs (Supplementary Fig. 2b), the immediate reduction in ROS by PEG itself[94] also seems to not be responsible. Although further studies are required to strictly determine the specific role of interstitial-fluid dynamics in the RVLM, our findings conform to the notion of its importance in blood-pressure control.

We conducted microinjection into the rat RVLM at a rate of 0.03–0.2 µl min$^{-1}$ (Methods), approximately 4–20 times faster than that reportedly associated with excellent preservation of central nervous tissue[95]. Although microinjection of 0.1–0.2 µl min$^{-1}$ has been successfully used in animal (rat) brain research[96–99], our approach may have compromised the brain region to some extent.

In contrast to the case of PHM in rats, VOCR in humans generates vertical accelerations at various body parts in addition to the head. We therefore cannot preclude the possibility that the effects of VOCR also involves mechanical regulation of tissues and organs other than the brain. For example, the antihypertensive effect of human VOCR became significant in about 2 weeks, approximately as quickly as or even a little more quickly than that of rat PHM (comparison of the MAP between SHRSPs in Fig. 1b and humans in Fig. 7b), despite the lower frequency of VOCR (PHM, 7 days per week; VOCR, 3 days per week). This could be attributed to some additional influence of cyclical vertical motion of body parts other than the head.

Our clinical studies are based on a small number of participants (Supplementary Table 2 and 3) with a fixed condition (frequency of 2 Hz, peak acceleration of around 1.0$g$, 30 min per day, 3 days per week, 12–14 rides), and our findings would benefit from further analysis, with a much larger sample size and using varying conditions, to determine the optimal VOCR.

The antihypertensive outcome of isometric exercise[100] cannot be explained by direct mechanical effects on the brain. Nevertheless, the application of moderate exercise-mimicking mechanical intervention is expected to be highly safe with minimal possibility of adverse effects, providing a therapeutic/preventative strategy for physical disorders including those resistant to conventional treatments such as drug administration. Mechanical interventions may bring considerable benefits to those who cannot receive them from exercise owing to physical disabilities.

## Methods

### Animal experiments and human studies

Animals were housed under a 12 h–12 h light–dark cycle with controlled temperature (22–24 °C) and humidity (50–60%), and treated with humane care under approval from the Animal Care and Use Committee of National Rehabilitation Center for Persons with Disabilities (approval number, 30-07). Male SHRSP/Izm and WKY/Izm rats were provided by the Disease Model Cooperative Association and astrocyte-GFP mice (*Aldh1L1-GFP* mice)[47] were obtained from GENSAT, acclimatized to the laboratory environments for at least 1 week, randomly divided into experimental groups and used for experiments.

All of the participants in our human studies provided written informed consent. The studies were approved by the Ethics Committees of the Iwai Medical Foundation and the National Rehabilitation Center for Persons with Disabilities (approval number, 30-01).

### Chemicals and antibodies

All of the chemicals were purchased from Sigma-Aldrich unless noted otherwise. Primary antibodies and their dilution rates used for immunostaining in this study are as follows: mouse monoclonal anti-GFAP (MAB360, Millipore, 1:1,000); rabbit polyclonal anti-GFAP (Z0334, Dako, 1:1,000); chicken polyclonal anti-GFAP (ab4674, Abcam, 1:2,000); rabbit polyclonal anti-cleaved caspase-3 (9661, Cell Signalling Technology, 1:1,000); mouse monoclonal anti-NeuN (MAB377, Millipore, 1:200); rabbit polyclonal anti-NeuN (ABN78, Millipore, 1:1,000), rabbit polyclonal anti-AT1R (HPA003596, Sigma-Aldrich, 1:200); rabbit polyclonal anti-AGTRAP (HPA044120, Sigma-Aldrich, 1:1,000); rabbit polyclonal anti-GFP (598, MBL, 1:2,000); chicken polyclonal anti-GFP (ab13970, Abcam, 1:2,000); and mouse monoclonal anti-TUJ-1 (ab78078, Abcam, 1:1,000). Secondary antibodies conjugated with Alexa Fluor 350, 488, 568, 633 and 647 (Thermo Fisher Scientific) were used at a dilution of 1:400. Cell nuclei were stained with DAPI (D9542, Sigma-Aldrich). The primary antibodies and their dilution rates used for immunoblot analysis were as follows: rabbit polyclonal anti-TNF-α (ab66579, Abcam, 1:250); rabbit polyclonal anti-IL-1β (ab9722, Abcam, 1:250); rabbit polyclonal anti-GAPDH (5174, Cell Signaling Technology, 1:10,000). Horseradish peroxidase-conjugated anti-rabbit IgG (H+L) secondary antibody (W401B, Promega) was used at a dilution of 1:5,000 for anti-TNF-α or anti-IL-1β blotting, and 1:10,000 for anti-GAPDH blotting.

### PHM application to rats

Rats were treated with PHM in a prone position using a platform that we developed to move the heads of rodents up and down[27] (Supplementary Video 1). During PHM, the animals were kept anaesthetized with 1.5% isoflurane except for in the µCT study, in which we used intraperitoneal injection of 2 mg per kg of midazolam (Sandoz), 2.5 mg per kg of butorphanol (Meiji Seika) and 0.15 mg per kg of medetomidine (Kyoritsu Seiyaku) for anaesthesia. The body temperature of tested

animals was maintained using a light heater. The PHM system was set up to reproduce the head motion (5 mm, 2 Hz) of treadmill running (20 m min⁻¹), which made 1.0 $g$ vertical acceleration peaks in the heads of rats examined[24]. The control rats in the PHM experiments were anaesthetized in the same manner, and placed in a prone position with their heads on the platform that was left static.

### Treadmill running of rats
Rats were subjected to compulsive running using a belt drive treadmill equipped with an electrical shock system (MK-680S, Muromachi). We habituated the rats to the treadmill system by placing them in the machine several times without turning on the treadmill belt during the acclimatization period. The electrical stimulation was turned on only once or twice during the first 5 min of the 30 min treadmill running on the first day of the 4 week treadmill running period. Thereafter, we did not need to turn on the electrical shock system to have the animals keep running, perhaps because the velocities that we employed (20 m min⁻¹) were moderate. The control rats in the treadmill running experiments were placed onto the belt daily for 30 min without turning on the treadmill.

### Measurement of blood pressure and heart rate of rats using radio telemetry
A telemetry pressure probe equipped with a microelectromechanical systems-based sensor (length, 2.0 mm; width, 0.47 mm; thickness, 0.60 mm; Millar) was implanted into the abdominal aorta of a rat at 9–10 weeks of age, according to a surgical procedure described previously[101]. Rats were allowed to recover for at least 14 days before the initiation of experimental interventions or analyses. During the periods of experimentation that involved repeated blood pressure and heart rate measurements over multiple weeks, the blood pressure and heart rate were monitored and recorded for continuous 30 min every 7 days between 10:00 and 12:00 by a multichannel amplifier and signal converter (LabChart 8, ADInstruments).

### Measurement of urinary noradrenaline excretion of rats
Urine excreted during the indicated 24 h period was collected under an acidic condition using a metabolic cage (KN-646, Natsume Seisakusho) connected to a glass flask containing 10 ml of 6 N HCl, and stored at −80 °C until assayed. Excretion of urinary noradrenaline was calculated by multiplying its concentration measured using an enzyme-linked immunosorbent assay (ELISA) kit (KA1891, Abnova) with the urine volume.

### Tissue preparation and immunostaining (immunohistochemical or immunocytochemical analysis)
Rats were anaesthetized with intraperitoneal injection of midazolam, butorphanol and medetomidine, and perfused transcardially with 4% paraformaldehyde (PFA; TAAB Laboratories Equipment) in phosphate-buffered saline (PBS (137 mM NaCl, 10 mM $Na_2HPO_4$, 2.7 mM KCl, 1.5 mM $KH_2PO_4$)). The brainstems were excised and post-fixed with 4% PFA in PBS overnight at 4 °C. The tissues were cryoprotected by soaking in 20% sucrose/PBS for 24 h and in 30% sucrose/PBS for an additional 24 h at 4 °C. Fixed brainstems were frozen in optimal cutting temperature compound (Sakura Finetek) and cut into 16-μm-thick coronal sections using a cryostat (CM3050S; Leica Microsystems). The sliced sections were permeabilized and blocked with 0.1% Tween-20 and 4% donkey serum (Merck Millipore) in Tris-buffered saline, and stained by incubating with primary antibodies at the appropriate dilutions followed by their species-matched secondary antibodies. Cell nuclei were stained with DAPI (Sigma-Aldrich). The slides were mounted with ProLong Gold Antifade Reagent (Thermo Fisher Scientific) and images were captured using the BZ-9000 digital microscope system (Keyence). Quantitative immunohistochemical analysis (that is, cell counting) of the RVLM was performed in the approximately same relative section

for each rat (11.3–12.3 mm caudal to the bregma, 1.5–2.5 mm lateral to the midline, 0–1 mm dorsal to the ventral surface of the medulla).

For immunocytochemistry, cultured cells were fixed with 4% PFA in PBS for 20 min at room temperature and permeabilized and blocked with 0.1% Triton X-100 and 10% fetal bovine (FBS; GE Healthcare Life Science) in PBS for 30 min at room temperature. The cells were then incubated with primary antibodies for 2 h and then with secondary antibodies for 1 h at room temperature.

### Microinjection into the rat RVLM
Rats were anaesthetized with intraperitoneal injection of midazolam, butorphanol and medetomidine except for angiotensin II or valsartan injection studies, in which we used 1.2–1.4 g per kg of urethane (Sigma-Aldrich), and then microinjected as described previously[102]. In brief, a 25s-G microsyringe (Hamilton) was stereotaxically positioned on the anaesthetized rats after exposure of the dorsal surface of the medulla. The needle placement was defined according to an atlas of the rat with stereotaxic coordinates[103]; anteroposterior angle: 18°, 1.8 mm lateral to the calamus scriptorius, 3.5 mm ventral to the dorsal surface of the medulla. The placement of the needle tip in RVLM was confirmed by ensuring the pressure response to a test-dose injection of L-glutamate[12,104] (100 nl of 1 mmol l⁻¹ in PBS). Microinjection of various compounds or mediums was made through a needle reinserted at the same coordinates with fixed infusion rates using a microsyringe pump instrument (KD Scientific). Except for the experiments to analyse the pressor or depressor responses, we held the syringe for 5 min after the injection to avoid reflux, pulled out the needle carefully, and sutured the skin. The volumes and rates of microinjection were as follows; angiotensin II (Auspep) and valsartan (Tocris Bioscience): 100 nl of 1 mmol l⁻¹ in PBS at 0.1 μl min⁻¹; AAV solutions: 300 nl at 0.03 μl min⁻¹; PEG solutions: 1 μl at 0.1 μl min⁻¹; Isovist: 1 μl at 0.2 μl min⁻¹.

### Analysis of pressor/depressor responses
Rats implanted with a telemetry pressure probe were anaesthetized with urethane, and analysed for pressor/depressor responses. Monitoring blood pressure, we injected angiotensin II or valsartan (100 pmol) stereotaxically into the unilateral RVLM according to the microinjection procedures described above. The injection side (right or left) was chosen randomly. When both pressor and depressor responses were analysed, at least 2 h elapsed between the injections of angiotensin II and valsartan (Fig. 2a). The maximal MAP change elicited from the baseline was statistically analysed[28].

### Production of AAV vectors
To achieve astrocyte- and neuron-specific transduction, we used AAV9 vectors expressing a transgene under the control of mouse *Gfap* and rat *Eno2* promoters, respectively. The astrocyte-specific *Gfap* promoter consists of 0.6 kb hybrid fragments containing ABC1D genomic regions upstream of the mouse *Gfap* gene[105]. The neuron-specific *Eno2* promoter is composed of the 1.2 kb genomic region upstream of the rat *NSE* gene[106]. Full-length rat *Agtrap* cDNA was synthesized (Eurofins Genomics) and inserted into the plasmid pAAV-GFAP-GFP-P2A-Cre-WPRE-SV40pA and pAAV-NSE-GFP-P2A-Cre-WPRE-SV40pA to generate pAAV-GFAP-GFP-P2A-AGTRAP-WPRE-SV40pA and pAAV-NSE-GFP-P2A-AGTRAP-WPRE-SV40pA. pAAV-GFAP-GFP-WPRE-SV40pA and pAAV-NSE-GFP-WPRE-SV40pA were used for experimental controls. Recombinant single-stranded AAV2/9 vectors were produced by transfection of HEK293T cells (Thermo Fisher Scientific) with the respective pAAV expression plasmid, pAAV2/9 and a helper plasmid (Stratagene) as previously described[107]. After collecting the conditioned medium, the viral particles were precipitated using polyethylene glycol 8000 and iodixanol continuous gradient centrifugation as previously described[108]. The genomic titre of purified AAV9 vectors was determined by qPCR targeting the WPRE sequence.

## Measurement of pressure in the rat medulla

Intramedullary pressure was measured using a blood pressure telemeter according to the procedure that we described previously[24,109]. To place the pressure sensor in the rat medulla, we made a ~1 mm hole in the occipital bone (2 mm lateral to the midline, 2 mm rostral to the caudal margin of occipital bone) using a dental handpiece (Osada Electric), stereotaxically inserted a 20-G needle (Terumo) as a guide sheath (4 mm in depth from the occipital bone surface) at an anteroposterior angle of 20° and sealed it in an airtight manner using dental cement (GC) to avoid pressure escape. We macroscopically observed that the needle tip was placed into the rat RVLM after these procedures in our pilot experiments. However, for this pressure measurement, we did not strictly confirm the placement of the needle tip in the RVLM using the aforementioned test-dose injection of L-glutamate. We therefore designate this experiment as an intramedullary pressure measurement. During the pressure measurement, respiration was monitored using a pulse transducer (TN1012/ST; ADInstruments) attached to the tested rats. Low-pass (50 Hz) filtered intramedullary pressure waves were analysed using LabChart 8 software.

## In vivo analysis of the distribution dynamics of interstitial fluid in the rat RVLM using µCT

Isovist (Bayer) was stereotaxically microinjected into the RVLM of anaesthetized 12-week-old male WKY rats according to the procedure described above, and visualized using µCT (inspeXio SMX-100CT, Shimadzu). After Isovist injection, the rats were analysed using two serial brain µCT scans between which PHM was either applied or not applied (kept sedentary) for 30 min (Fig. 4e). µCT images were analysed using software for 3D morphometry (TRI/3D-BON-FCS64, RATOC System). Voxels with ≥1.02 times signal intensity compared with that of air was defined as an Isovist cluster in the rat RVLM.

## Hydrogel introduction in the rat RVLM

Just before use, a premixture of PEG with functional groups (25 g l⁻¹ in PBS) was prepared from tetra-armed thiol-terminal (TetraPEG-SH) (Yuka-Sangyo) and acrylate-terminal (Tetra-PEG-ACR) (JenKem Technology) PEG solutions as we previously described[24]. Tetra-armed polyethylene glycol without functional groups (25 g l⁻¹ in PBS) was used as an ungelatable control. For the analysis of hydrogel distribution in the rat RVLM, we used Tetra-PEG-SH fluorescently labelled with a thiol-reactive probe (Thermo Fisher Scientific or Merck). Microinjection of PEG solutions into the rat RVLM was conducted as described above.

To specifically analyse the consequences of PHM and hydrogel introduction by minimizing possible invasive influences of the microinjection itself, we gave 1 week recovery time before the first blood pressure measurement, and then applied PHM to the rats (daily 30 min, 14 or 28 days). Immediately after the post-PHM 24 h urine collection (Fig. 6a), rats were euthanized by transcardial infusion of PFA and processed for histological analysis.

## Multiphoton microscopy imaging and analysis of the interstitial space structure/orientation and dimension in the rat RVLM

We prepared PFA-fixed lower brainstem tissue samples from 12-week-old WKY rats 24 h after introducing fluorescently labelled hydrogel conjugated with Alexa Fluor 594 C5 maleimide (A10256, Thermo Fisher Scientific) to their RVLM as described above. We used a multiphoton microscope (FVMPE-RS-UPSP2, Olympus), which is suitable for viewing a deep part of a relatively large sample. This was because we wanted to minimize the possible influence of plastic deformation/distortion or disruption of hydrogels (and tissues) that might occur during the experimental procedures (for example, sample sectioning) particularly in the vicinity of the cut section. The sample was partially embedded in 2% (w/v) agarose (A9414; Sigma-Aldrich) and immersed in PBS. An excitation laser (Mai Tai DeepSee, Spectra-Physics) was tuned at 800 nm, and the beam was focused with an objective lens (XLPLN25XWMP2, Olympus). The fluorescence was collected with the same objective lens, and detected using a GaAsP detector after passing through a red band-pass filter (BA575-645, Olympus). Considering the relative extension of multiphoton excitation in the incident direction of excitation laser beam[110], which is parallel to the optical axis of objective lens, we acquired and analysed images of the three mutually orthogonal planes (*xy*, *xz* and *yz* planes; Extended Data Figs. 6a–c and 7a–c).

Image analysis to determine the structure/orientation of the interstitial space was conducted as follows. Approximately 20 µm stacked long rectangular voxel images (20 voxels of 1 µm length) were resampled to 100 cubic voxels (0.2072 µm on each side) and smoothed to a width of 5 voxels (around 1 µm). Following the previous reports on the occupancy of interstitial space in the brain[98,111], we extracted the top 20% high-fluorescence voxel population as clusters representing the interstitial space (Extended Data Fig. 6a (bottom)). For huge clusters (>5,000 voxels), the bottom 50% population of low-fluorescence voxels in the cluster was removed to reduce the size of the cluster. These procedures, together with a clearance of clusters of <50 voxels enabled us to apply elliptic fitting to determine the major and minor axes of each processed cluster on the stacked plane (Extended Data Fig. 6b). For each sample, we analysed 100 slices, focusing on the plane with the highest mean fluorescence intensity along the stack direction.

To quantitatively analyse the cross-sectional area of the interstitial space, we resampled ~4 µm stacked long rectangular voxel images to 20 cubic voxels (Extended Data Fig. 7a). On the basis of the voxel number of each individual cluster (Extended Data Fig. 7b), we drew the distribution of its cross-sectional area (Extended Data Fig. 7c). Analysing the probability density by fitting to the log-normal distribution[112,113], we determined the mode and full width at half maximum (FWHM)[114] of the cross-sectional areas of the individual interstitial spaces (Extended Data Fig. 7c). We used MATLAB (v.2021a; MathWorks) for these image analyses.

## MRI scanning and orientation analysis of the rat lower brainstem

Male WKY rats (aged 12 weeks) were scanned in the 7.0-Tesla MRI system (Biospec 70/30-USR, Bruker BioSpin) using 2D rapid acquisition with relaxation enhancement (RARE). Parameters for the sagittal $T_2$-RARE sequence were as follows: echo time (TE), 33 ms; repetition time (TR), 3,600 ms; slice thickness (SL), 0.75 mm; field of view (FOV), 38.4 mm × 38.4 mm; voxel, 0.15 mm × 0.15 mm × 0.75 mm; RARE factor, 8; number of averages, 4. We analysed the MRI images using SPM12 on MATLAB (https://www.fil.ion.ucl.ac.uk/spm/software/spm12/). Each image was smoothed with 0.24 mm FWHM, and registered to the template atlas of the Fischer 344 rat brain[115]. Assuming an equivalent weight for each voxel, we determined the centre of mass of the lower brainstem in each image in a particular plane (*xy*, *xz* or *yz* plane) and drew its centroidal line by combining the least squares and sigmoid fitting. As the *y* axis was actually the direction of the maximum diameter of this part of the brain, the longest centroidal line was drawn by fitting to the mass centres determined in the *xz* plane images (Extended Data Fig. 6d).

## Calculation of the magnitude of fluid shear stress on the cells in the rat RVLM during PHM

We calculated the interstitial-fluid flow-derived fluid shear stress imposed on cells in the rat RVLM during PHM by referring to the findings from our µCT analysis (Fig. 4g,h) and multiphoton microscopy analyses (Extended Data Figs. 6 and 7) according to Henry Darcy's law, which defines the flux density of penetrating fluid per unit time[116]. On the basis of the PHM-induced enhancement of Isovist spread (Fig. 4g,h), we estimated the velocity of the interstitial-fluid flow (*u*) during PHM as 0.4–0.6 µm s⁻¹, two- to three-times that in the brain of sedentary rodents, which is reported to be 0.2 µm s⁻¹ (refs. 44,45).

We then calculated the Darcy permeability ($K_p$), which relates to the flow or pressure drop of liquid across a porous structure[117], in the rat RVLM tissue. The Kozeny–Carman equation enables us to calculate the permeability given pore-size/sphericity data[118,119], where a value of 4.5–5.5 was considered to be the Kozeny constant[118–120] (Supplementary Table 1a) on the basis of the apparent structure/orientation of the interstitial space in the rat RVLM (Extended Data Fig. 6e and 7d). We referenced the interstitial space dimension (cross-sectional area or diameter) estimated by analysing the microscopy images (Extended Data Fig. 7c). Fluid shear stress ($\tau$) at the interstitial cell surface was calculated as described in Supplementary Table 1 (ref. 121).

## Cell culture

Primary cultures of astrocytes were prepared from the cortex of neonatal (one day old) mouse pups as described previously[122,123] with some modifications. The brains were excised from astrocyte-GFP mice[47] immediately after euthanasia by $CO_2$ inhalation (typically 4–5 pups at one time), and the cortices were dissected to small pieces in DMEM (FUJIFILM Wako Pure Chemical) using forceps under a stereomicroscope (SV-11; Zeiss). The dissected pieces of cortex tissue were collected by centrifugation at 190$g$ for 3 min, suspended and incubated in an isotonic solution (124 mM NaCl, 5 mM KCl, 3.2 mM MgCl$_2$, 0.1 mM CaCl$_2$, 26 mM NaHCO$_3$, 10 mM D-glucose, 100 IU ml$^{-1}$ penicillin, 100 μg ml$^{-1}$ streptomycin) containing 0.1% trypsin, 0.67 mg ml$^{-1}$ hyaluronidase and 0.1 mg ml$^{-1}$ deoxyribonuclease I at 37 °C for 10 min to dissociate cells. Subsequently, an equal volume of DMEM/F12 (Thermo Fisher Scientific) containing 10% FBS, 100 IU ml$^{-1}$ penicillin and 100 μg ml$^{-1}$ streptomycin was added to neutralize the trypsin. After multiple pipetting for further homogenization with a 10 ml serological pipet, the cell suspension was cleared of tissue debris using a 70 μm nylon-mesh strainer (Corning Life Sciences). Cells were then washed twice with DMEM/F12 containing 10% FBS, 100 IU ml$^{-1}$ penicillin and 100 μg ml$^{-1}$ streptomycin, plated onto a poly-D-lysine-coated T75 flask (Corning Life Sciences, Corning) and kept in a humidified incubator (5% $CO_2$, 95% air, 37 °C). The culture medium was replaced with fresh medium every 3 days, and it typically took 7–10 days for the cells to become confluent in the flask. Astrocyte-enriched cell population was then obtained by physically detaching the other types of cells, such as microglia and oligodendrocytes. After shaking the flask overnight using an orbital shaker (200 rpm with BR-40LF; TAITEC), the detached cells were removed together with the medium. Cells that remained attached to the flask were replated on poly-D-lysine-coated 10 cm dishes (Corning Life Sciences) so that the cell density became approximately one-third (typically 3–4 dishes were used to achieve this). The medium was replaced with fresh medium every 3 days until the cells reached 80–90% confluence and were used for experiments. We confirmed that >95% of the cells were GFP positive (Extended Data Fig. 8a), indicating a high degree of astrocyte purity.

Mouse neuroblastoma-derived Neuro2A cells (provided by T. Yokota), which exhibit neuronal phenotypes and morphology[48,49], were cultured in DMEM supplemented with 10% FBS, 100 IU ml$^{-1}$ penicillin and 100 μg ml$^{-1}$ streptomycin in a humidified incubator (5% $CO_2$, 95% air, 37 °C).

## Application of fluid shear stress to astrocytes or Neuro2A cells in culture

Astrocytes or Neuro2A cells grown in a poly-D-lysine-coated 35-mm culture dish (Corning Life Sciences, Corning) were exposed to pulsatile fluid shear stress (average 0.7 Pa) for 30 min. As we previously reported[23,24,41], a parallel-plate flow chamber and a roller pump (Masterflex, Cole-Parmer) were used to apply fluid shear stress. The flow chamber, which was composed of a cell culture dish, a polycarbonate I/O unit and a silicone gasket, generated a flow channel with a length of 23 mm, width of 10 mm and height of 0.5 mm. Astrocytes or Neuro2A cells were seeded at a density of $8 \times 10^5$ cells per 8.0 cm$^2$. To maintain the

pH and temperature of culture medium, we used a 5% $CO_2$-containing reservoir and a temperature-controlled bath.

## Application of HPC to astrocytes in culture

Astrocytes seeded at a density of $8 \times 10^5$ cells per 8.0 cm$^2$ were exposed to cyclical HPC with various amplitudes at a frequency of 0.5 Hz for 30 min using a custom-made pressure system described previously[41]. The system consists of a cell culture dish, a polycarbonate pressure chamber, a silicone gasket, an O-ring, a quartz glass, two holding jigs, a thermostatic chamber and a syringe pump. The system was completely airtight, enabling precise and strict pressure control with the syringe pump. To maintain the temperature of culture medium, the entire system was placed in a 37 °C incubator during HPC application. In both fluid shear stress and HPC experiments, cell adhesion and morphology were observed using a light microscope (DM IRE2; Leica Microsystems).

## qPCR with reverse transcription analysis

Total RNA (500 ng) extracted from cell culture was processed for reverse transcription using the ISOGEN II (NIPPON GENE) and PrimeScript RT reagent Kit (TaKaRa). The resulting cDNA was analysed using qPCR using glyceraldehyde-3-phosphate dehydrogenase (*Gapdh*) as an internal control in the Applied Biosystems 7500 Real Time PCR System with the Power SYBR Green PCR Master Mix (Thermo Fisher Scientific).

The primers (sense and antisense, respectively) were as follows: mouse *Agtr1a* (AT1R-encoding gene): 5′-AAAGGCCAAGTCGCACTCAAG-3′ and 5′-TCCACCTCAGAACAAGACGCA-3′; mouse *Gapdh*: 5′-GCAAAGTGGAGATTGTTGCCAT-3′ and 5′-CCTTGACTGTGCCGTTGAATTT-3′; and WPRE (for genomic titration of purified AAV9 vectors): 5′-CTGTTGGGCACTGACAATTC-3′ and 5′-GAAGGGACGTAGCAGAAGGA-3′.

## Fluorescent angiotensin-II binding assay

Six or twenty-four hours after the termination of fluid shear stress application, cultured astrocytes were incubated with the angiotensin II type 2 receptor inhibitor PD123319 ($10^{-6}$ mol l$^{-1}$ in PBS; ab144564, Abcam), for 20 min, and then with tetramethylrhodamine (TAMRA)-labelled angiotensin II ($10^{-8}$ mol l$^{-1}$ in PBS; AS-61181, AnaSpec) for 30 min. After three washes with PBS, the samples were fixed and immunostained with anti-GFP antibodies to strengthen the GFP-derived green fluorescence signals and corroborate our analysis on astrocytes prepared from astrocyte-GFP mice as well as to secure the binding of fluorescent angiotensin II. Green and red fluorescence was viewed using a fluorescence microscope (BZ-9000 HS; Keyence). The samples from astrocytes that were left unexposed to fluid shear stress were prepared and viewed in the same manner, and provided an experimental control.

## TUNEL assay

Rat RVLM sections were stained using a terminal deoxynucleotidyl-transferase-mediated dUTP nick-end labelling (TUNEL) kit (Biotium) according to the manufacturer's protocols, counterstained with DAPI and then viewed using a ×20 objective with a fluorescence microscope (BZ-9000 HS; Keyence). The nuclei of apoptotic cells were determined by the existence of green fluorescent patches, and cell apoptosis was quantified with reference of their counts to the total numbers of nuclei defined by DAPI staining.

## Immunoblot analysis

Rat RVLM tissue was excised immediately after cervical dislocation, mechanically homogenized, solubilized with RIPA buffer (25 mM Tris·HCl pH 7.6, 150 mM NaCl, 1% NP-40, 1% sodium deoxycholate, 0.1% SDS) with protease and phosphatase inhibitors (78440; Thermo Fisher Scientific), and analysed using SDS–PAGE followed by anti-TNF-α, anti-IL-1β and anti-GADPH immunoblotting. Specific signals were visualized and quantified using the Odyssey infrared imaging system (LI-COR Biosciences) and ImageJ (NIH).

## Analysis of rat ADNA and blood pressure during the transition from before to after the initiation of PHM

Aortic depressor nerve activity (ADNA) recording was conducted in isoflurane-anaesthetized (2%) 12–16-week-old male SHRSPs as described previously[124] with some modifications. In brief, using a stereomicroscope (M80; Leica), we surgically exposed and isolated the left aortic depressor nerve through an anterior neck approach. We then placed a bipolar cuff electrode below the nerve bundle and embedded the nerve–electrode contacting complex in a two-component silicone gel (932; Wacker Chemie). Multifibre ADNA was 50–10,000 Hz band-pass filtered and amplified with a differential preamplifier (MEG-5200, Nihon Kohden). The ADNA was identified using its unique spontaneous activity synchronous with the arterial systolic cycle as an indicator. Blood pressure was monitored using a pressure transducer (DX-360, Nihon Kohden)/polygraph system (UBS-100-6, Unique Medical) connected to a catheter inserted into the right femoral artery. The PHM cycle was also monitored using a piezoelectric pulse transducer (MLT1010, ADInstruments). ADNA, blood pressure and PHM cycle were digitized using the Power1401 mkII system and Spike2 software (Cambridge Electronic Design) at a sampling frequency of 3 kHz. The recorded ADNA was then processed for DC offset removal, rectified, integrated and smoothed with a time constant of 3 ms by Spike2. The integrated value of ADNA measured after euthanasia by isoflurane overdose was considered to be background noise and subtracted from the entire data collected. ADNA was then normalized in each rat (Supplementary Fig. 4a). The relative ADNA value (%) averaged for each heartbeat was scaled with the mean of 10 heartbeats immediately before the PHM initiation set as a baseline (100%) (Supplementary Fig. 4b). The SBP, DBP, MAP and heart rate were determined using Spike2.

## Measurement of accelerations in the human head

To measure the accelerations at the human head during treadmill running or VOCR, we fixed an accelerometer (NinjaScan-Light, Switchscience) to the forehead with a surgical tape. Vertical acceleration was evaluated using the software application provided from the manufacture.

## Design and participants of the clinical study on antihypertensive effects of VOCR

We conducted single-arm (protocol 1 and 2) and non-randomized controlled (protocol 3) clinical studies (UMIN000040420). Two male and three female participants aged 37–60 years participated in the study of protocol 1 (Supplementary Fig. 5c), which was carried out at the affiliated health services facility of Iwai Medical Foundation (Iwai Keiaien). In total, 16 male and 14 female participants aged 23–85 years participated in the studies of protocol 2 (Supplementary Fig. 6a) and 3 (Supplementary Fig. 7a), which were carried out at the National Rehabilitation Center for Persons with Disabilities Hospital. Three male and two female participants participated in both the protocol 2 and 3 studies. In protocol 3, the participants were assigned to either NOCR or VOCR, depending on their intention and choice, whereas 14 individuals underwent both NOCR and VOCR after we obtained their consent (Supplementary Table 3). In principle, for the NOCR and VOCR participants, NOCR intervention was administered before VOCR, considering that the antihypertensive effect of VOCR remains for a while after its termination (Fig. 7b). In only one participant (participant 39), VOCR intervention was performed before NOCR. She first participated in the VOCR study on her agreement and choice, and then agreed to the NOCR study after the completion of the VOCR intervention. We confirmed that her blood pressure had stably returned to the pre-VOCR level before starting the NOCR intervention; her NOCR was initiated 10 weeks after the last bout of VOCR. Continuous beat-by-beat recording of blood pressure and RRI was conducted for the participants who agreed to these measurements in protocol 3. As the system for continuous blood pressure recording became available after we

started the protocol 3 study, two participants in the NOCR group (22 and 23) were analysed by continuous RRI recording alone (that is, void of continuous blood pressure recording). Participants 32 and 39 were excluded from continuous blood pressure and RRI recording, despite agreeing to these measurements, because arrhythmia was revealed on electrocardiography (the first RRI recording) and confirmed to be atrial fibrillation by their primary care physicians (Supplementary Table 3).

Participants were considered to be eligible if they were 20–75 (protocols 1 and 2) or ≥20 (protocol 3) years old and confirmed to have 130–160 mm Hg of SBP at the time of interview for informed consent and eligibility check. Participants with mental or psychological illnesses, history or presence of cardiovascular events, history or presence of severe dysfunction of the liver, kidney, lung, gastrointestinal tract and spine, or the presence of acute injuries or diseases (such as recent traumas and infectious diseases) were excluded, with the exception of those who were given permission for participating in this study from their primary care physicians. Whereas antihypertensive medication did not disqualify the participants (Supplementary Table 2 and 3), they were advised not to change their medication from at least one month before to the first bout of NOCR or VOCR through the study period (that is, up to 8 weeks after the last bout of NOCR or VOCR). At a certain (approximately fixed) timepoint in the morning (typically just before breakfast), they conducted three consecutive measurements of blood pressure (mm Hg) and heart rate (bpm) using an automated upper-arm-cuff sphygmomanometer, and recorded the values from all of those measurements. These procedures of blood pressure measurement and recording were performed in accordance with the Japanese Society of Hypertension Guidelines for the management of hypertension (JSH2019)[125]. The participants were directed to start periodical (≥3 days per week) blood pressure/heart rate measurements at least 2 weeks (protocols 1 and 2) or 1 week (protocol 3) before the initiation of the intervention (that is, NOCR or VOCR) and continue to measure blood pressure/heart rate throughout the study period using the same sphygmomanometers. Particularly, during the studies of protocol 2 and 3 (that is, the studies at the National Rehabilitation Center for Persons with Disabilities), the participants were advised to record all the data of those measurements. Those whose blood pressure lowered below the eligibility requirement of the study (≥130 mm Hg of SBP) before the initiation of intervention (NOCR or VOCR) were eliminated from the study. The participants were directed to be rested and keep calm for at least 1 min before starting to measure blood pressure/heart rate. The mean blood pressure/heart rate value from three measurements was defined as the value of the day, and used for statistical analysis. When blood pressure and heart rate were measured and recorded on ≥3 days during a particular week in the studies of protocol 2 and 3, the mean of all of the values of the day through the week was defined as the value of the week. For the participants who agreed, periodical blood pressure/heart rate measurement and recording (≥3 days per week) was extended up to 8 weeks after the last bout of VOCR (protocol 2 and 3). MAP was calculated using a standard formula as follows: MAP = DBP + 1/3(SBP − DBP).

The human research participants provided written informed consent for publication of potentially identifiable information in Supplementary Tables 2 and 3 of the Supplementary Information. The participants in the studies of protocol 2 and 3 received monetary compensations in accordance with the institutional rule of the National Rehabilitation Center for Persons with Disabilities.

## Blood sampling and measurement of parameters in the plasma and serum of humans

Blood sampling in the human study of protocol 2 was conducted between 12:00 and 15:00. The participants were rested in a sitting position for at least 10 min before starting the sampling procedures. After plasma (for catecholamines and renin activity) and serum (for aldosterone and CRP) separation by centrifugation, we outsourced the measurement of parameters to be tested (BML).

**Continuous beat-by-beat blood pressure and RRI recording to evaluate vascular sympathetic nerve activity and balance of cardiac sympathetic/parasympathetic nerve activity, respectively**

In the study of protocol 3, we evaluated the vascular sympathetic nerve activity and the balance between the cardiac sympathetic and parasympathetic nerve activities (that is, sympatho-vagal balance) in the participants who agreed to these measurements. We analysed the LF (0.04–0.15 Hz) power of the power spectrum density of SBP variability[59,60] and the LF/HF (0.15–0.4 Hz) ratio of the power spectrum density of RRI variability[59,61], respectively, at the beginning and end periods of the intervention (NOCR and VOCR) (Supplementary Fig. 7a). The power spectrum densities of the SBP and RRI variability signals were calculated by applying a Fourier transform with a Hanning window[126]. Continuous blood pressure recording was performed using a continuous non-invasive arterial pressure system that measures blood pressure through an automated inflatable cuff wrapped around the proximal phalanx of index or middle finger (LiDCOrapid V3, Merit Medical Japan)[127]. Continuous RRI recording was conducted using a wearable electrocardiography system (myBeat; Union Tool)[128]. Although we monitored blood pressure and RRI continuously for 10 min at each recording, we analysed the data from the central 256 s (4.27 min) of the last 5 min. It took us around 5 min to prepare for continuous blood pressure and RRI monitoring and confirm the proper system operation before starting blood pressure and RRI recording. Therefore, the analysed data were from the participants who had rested on the chair for at least 10 min. The average values from five (that is, 5 days) measurements at the beginning and end periods of intervention (NOCR or VOCR) were statistically analysed. In the case that the blood pressure was not determined or SBP was recorded as <50 mm Hg, the measurement was considered to be failure. For RRI analysis, we first removed RRIs of <0.5 s and >1.5 s as erroneous values. RRI values outside the range of 0.6 to 1.4 times the average of the 10 immediately preceding ones were also removed owing to a lack of reliability. In the case that the sum of time for blood pressure recording failures exceeded 4% or, for RRI, value removal exceeded 10% during the last 5 min of recording, the continuous blood pressure or RRI measurement on that particular day was deemed to be unsuccessful and was excluded from the statistical analysis. Nevertheless, all statistically analysed data were the mean values from at least four measurements except for the 'end' RRI variability for participant 31's NOCR, which was from three measurements owing to two failures in recording.

### Statistical analysis

All quantitative data are presented as mean ± s.e.m. Parametric statistical analyses were conducted using paired or unpaired two-tailed Student's $t$-tests for two-group comparison (except for SBP and RRI variabilities) using Prism software (v.8; GraphPad Software). SBP and RRI variabilities were analysed using Wilcoxon signed-rank tests, using R software (https://www.r-project.org/) in the human study of protocol 3. One-way ANOVA and one- or two-way repeated measures ANOVA with Tukey's, Dunnett's or Bonferroni's post hoc test were conducted for multiple (≥3) group comparison using Prism. $P < 0.05$ was considered to be significant.

### Reporting summary

Further information on research design is available in the Nature Portfolio Reporting Summary linked to this article.

### Data availability

The main data supporting the findings of this study are available within the Article and its Supplementary Information. The raw data generated during the study are too large to be publicly shared, yet they are available for research purposes from the corresponding author on reasonable request. The template atlas of the Fischer 344 rat brain is available at Zenodo[115]. Source data are provided with this paper.

### Code availability

Source code used in this work is available for non-commercial purposes from the corresponding author on request. The MATLAB scripts to compute the cluster-angle and aspect-ratio distributions from multiphoton microscope images (Extended Data Figs. 6c and 7c) and to compute the lower brainstem and longest centroidal line of this part of the brain (Extended Data Fig. 6d) are available at Zenodo (https://doi.org/10.5281/zenodo.7936475).

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

## Acknowledgements

We thank K. Nakanishi, K. Hamamoto and N. Kume for their support. This work was in part supported by Intramural Research Fund from the Japanese Ministry of Health, Labour and Welfare; Grants-in-Aid for Scientific Research from the Japan Society for the Promotion of Science (KAKENHI 15H01820, 15H04966, 18H04088, 20K21778 and 21H04866 to Y.S.; 20K19367 to N.S.; 17H02127 and 18H03138 to T.O.; 19K06899 and 22K06454 to A.K.); Brain Mapping by Integrated Neurotechnologies for Disease Studies (Brain/MINDS, JP20dm0207057 and JP21dm0207111 to H.H.) and Research and Development Grants for Comprehensive Research for Persons with Disabilities (21dk0310116j0201 to Y.S. and D.Y.) from the Japan Agency for Medical Research and Development (AMED); the JST-Mirai Program (JPMJMI20D4 to Y.S. and D.Y.); the Naito Science & Engineering Foundation (to Y.S.); and the Uehara Memorial Foundation (to Y.S.).

## Author contributions

S.M. and N.S. conducted most of the animal and cell experiments. Y.S. and N.S. performed the human studies. Y.S. conceived the research, designed the study and led the project. K.S. and T.K. provided technical advice for the experiments involving measurement of cardiovascular variables. N.S., K.Takano., S.M., M.S. and Y.S. wrote the manuscript. T.M. and A.T. contributed to the design and construction of the machine for PHM. D.Y. helped the in vitro fluid shear-stress experiments. D.Y. and K.F. performed the calculation of in vivo fluid shear stress. T. Sakai and Y.Y. developed and provided the PEG hydrogel system. A.K. and H.H. prepared and provided the AAV vectors. S. Sato and S. Saito conducted the multiphoton microscopy and MRI experiments, respectively, and K.Takano. analysed the images. H.T. conducted the beat-by-beat blood pressure, heart rate and ADNA analyses for rats. K.Y. analysed the data from human continuous blood pressure monitoring and RRI recording. K.Tomiyasu., T.K., M.A., H.I., Y.M. and T.O. supported Y.S. in the human studies. K.S., T. Saito, S.T. M.S., T.O., H.O. and M.N. provided technical, advisory and financial support.

## Competing interests

S.M., T.M., T.O., A.T. and Y.S. joined the application of a patent for the vertically oscillating chair, which has been awarded in Japan, the US, the EU and China (JP6592834; US16/616,935; EP18806753.2; CN201880033284.0) and is under review in India (IN201927048891). The other authors declare no competing interests.

## Additional information

**Extended data** is available for this paper at https://doi.org/10.1038/s41551-023-01061-x.

**Correspondence and requests for materials** should be addressed to Yasuhiro Sawada.

[1]Department of Rehabilitation for Motor Functions, National Rehabilitation Center for Persons with Disabilities, Tokorozawa, Japan. [2]Department of Orthopaedic Surgery, Graduate School of Medicine, The University of Tokyo, Tokyo, Japan. [3]Department of Cell Biology, National Cerebral and Cardiovascular Center, Suita, Japan. [4]Division of Advanced Applied Physics, Institute of Engineering, Tokyo University of Agriculture and Technology, Koganei, Japan. [5]Department of Rehabilitation for Brain Functions, National Rehabilitation Center for Persons with Disabilities, Tokorozawa, Japan. [6]Department of Neurophysiology & Neural Repair, Gunma University Graduate School of Medicine, Maebashi, Japan. [7]Department of Cardiovascular Medicine, Faculty of Medical Sciences, Kyushu University, Fukuoka, Japan. [8]Department of Cardiology, Graduate School of Medicine, International University of Health and Welfare, Okawa, Japan. [9]Department of Chemistry and Biotechnology, Graduate School of Engineering, The University of Tokyo, Tokyo, Japan. [10]Tokorozawa Heart Center, Tokorozawa, Japan. [11]Inanami Spine & Joint Hospital/Iwai Orthopaedic Medical Hospital, Iwai Medical Foundation, Tokyo, Japan. [12]Center of Sports Science and Health Promotion, National Rehabilitation Center for Persons with Disabilities, Tokorozawa, Japan. [13]Department of Assistive Technology, National Rehabilitation Center for Persons with Disabilities, Tokorozawa, Japan. [14]Department of Rehabilitation Medicine, Graduate School of Medicine, The University of Tokyo, Tokyo, Japan. [15]Department of Cardiac Physiology, National Cerebral and Cardiovascular Center, Suita, Japan. [16]Department of Advanced Medical Technologies, National Cerebral and Cardiovascular Center, Suita, Japan. [17]Department of Medical Physics and Engineering, Division of Health Sciences, Osaka University Graduate School of Medicine, Suita, Japan. [18]School of Biological and Environmental Sciences, Kwansei Gakuin University, Sanda, Japan. [19]Department of Health and Sports, Niigata University of Health and Welfare, Niigata, Japan. [20]Institute of Fluid Science, Tohoku University, Sendai, Japan. [21]Department of Clinical Research, National Rehabilitation Center for Persons with Disabilities, Tokorozawa, Japan. [22]These authors contributed equally: Shuhei Murase, Naoyoshi Sakitani. ✉e-mail: ys454-ind@umin.ac.jp

**a**

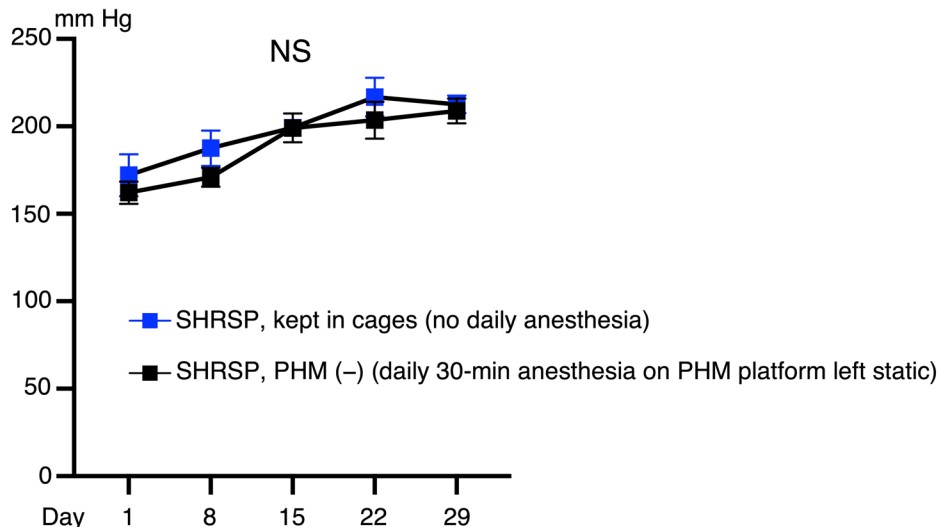

**b**

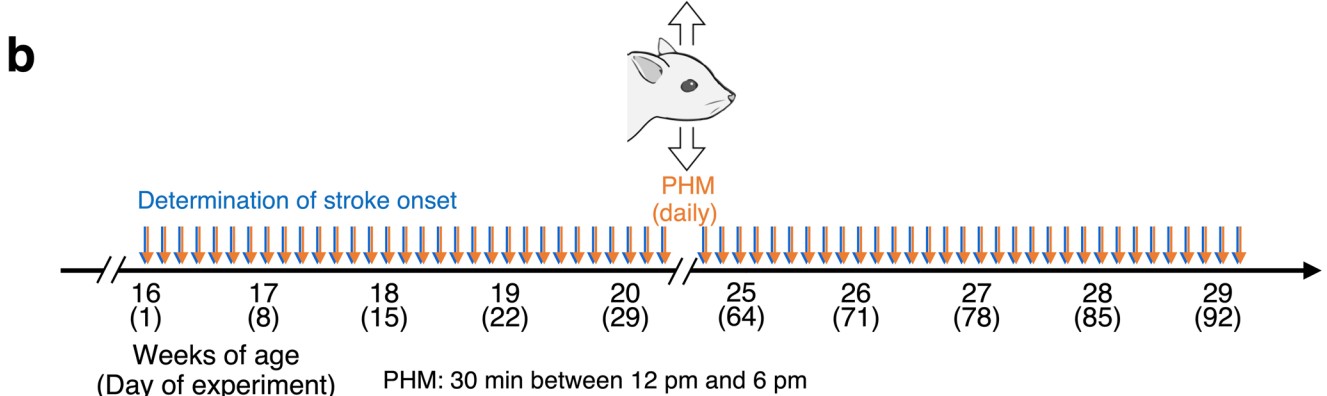

**c**

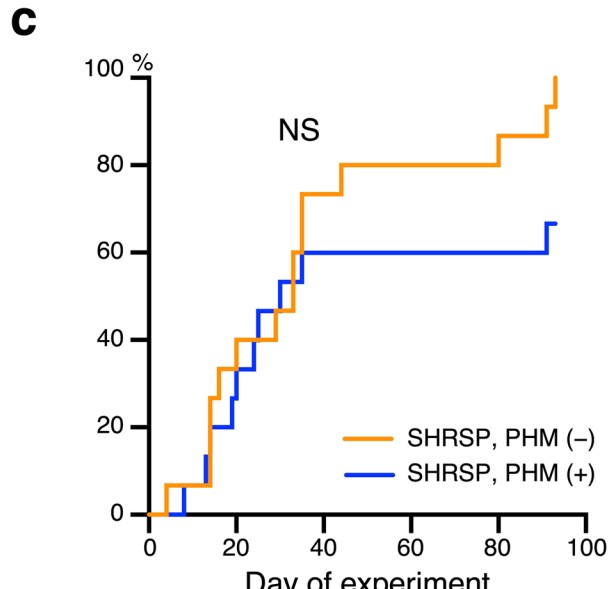

**d**

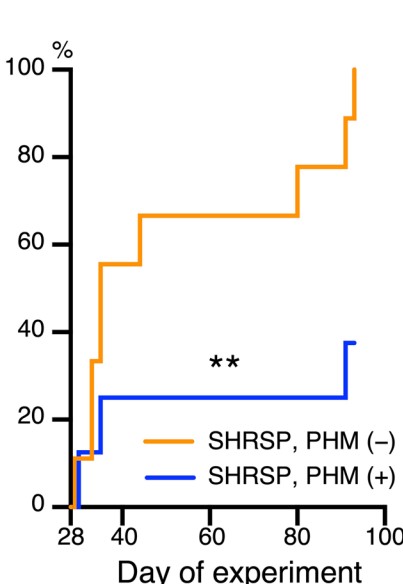

**Extended Data Fig. 1 | See next page for caption.**

**Extended Data Fig. 1 | Daily anaesthesia alone does not affect blood pressure, and ≥ 4 weeks of daily PHM decreases or delays stroke incidence in SHRSPs.** **a**, Time courses of MAP in SHRSPs either routinely kept in cages (no anaesthesia) or subjected to daily anaesthesia on the platform of PHM machine without turning on its switch [PHM (−)] [$P > 0.9999$ for Day 15, Day 22, and Day 29. $n = 6$ rats for no daily anaesthesia; $n = 8$ rats for PHM (−)]. **b**, Schematic representation of the experimental protocol to analyse the effects of PHM on the stroke incidence in SHRSPs. **c,d**, Kaplan-Meier curve of the stroke incidence in SHRSPs with and without PHM represented as PHM (+) and PHM (−), respectively. Data from SHRSPs that did not have a stroke during the first four weeks after the initiation of intervention (daily PHM or anaesthesia alone) is shown in **(d)** [**c**: $P = 0.1179$. $n = 15$ rats for PHM (−); $n = 15$ rats for PHM (+). **d**: $P = 0.0093$. $n = 9$ rats for PHM (−); $n = 8$ rats for PHM (+)]. Data are presented as mean ± s.e.m. **$P < 0.01$; NS, not significant; two-way repeated measures ANOVA with Bonferroni's post hoc multiple comparisons test **(a)** or long-rank test **(c,d)**.

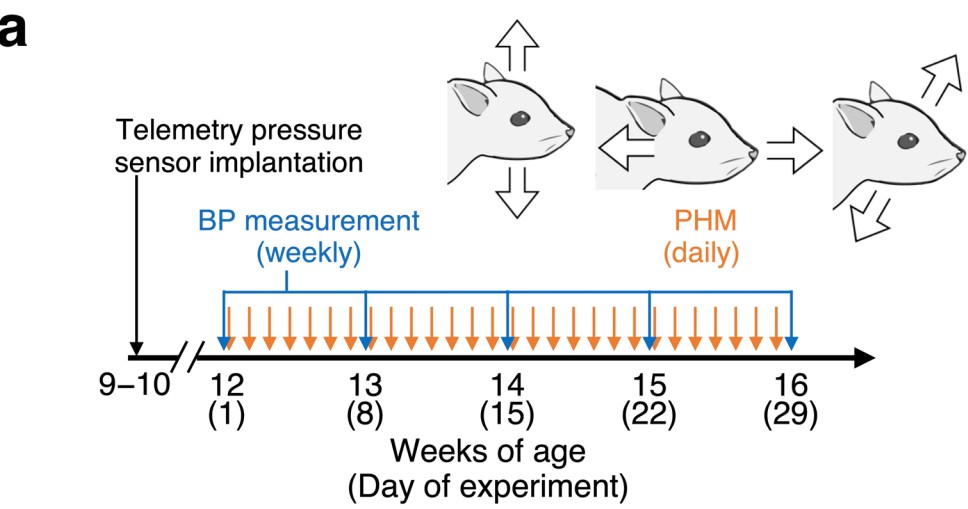

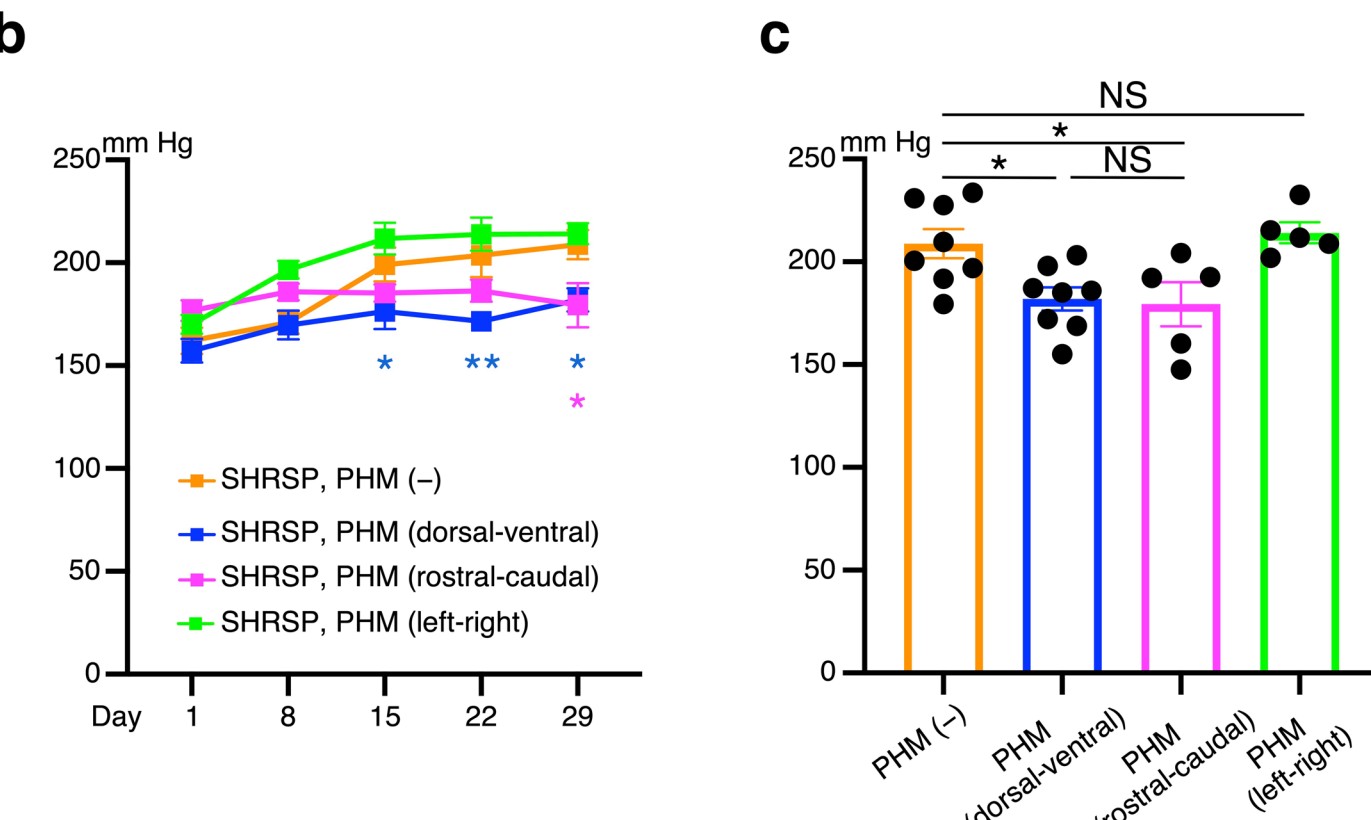

**Extended Data Fig. 2 | PHM in the rostral–caudal direction, but not left–right, direction lowers blood pressure in SHRSPs. a**, Schematic representation of the experimental protocol to analyse the effects of PHM on the blood pressure in rats. The cartoon represents PHM in the dorsal-ventral (left), rostral-caudal (centre), and left-right (right) directions. **b,c**, Time courses **(b)** and values on Day 29 **(c)** of MAP in SHRSPs, subjected to either daily PHM or anaesthesia only [**b**, orange vs. blue: $P = 0.0376$ for Day 15, $P = 0.0018$ for Day 22, $P = 0.0105$ for Day 29; orange vs. magenta: $P = 0.4131$ for Day 15, $P = 0.2448$ for Day 22, $P = 0.0148$ for Day 29; orange vs. green: $P = 0.4869$ for Day 15, $P = 0.6439$ for Day 22, $P = 0.9273$ for Day 29. **c**: $P = 0.0363$ for column 1 vs. 2, $P = 0.0465$ for column 1 vs. 3, $P = 0.9576$ for column 1 vs. 4, $P = 0.9948$ for column 2 vs. 3. $n = 8$ rats for PHM (−) and PHM (dorsal-ventral); n = 5 rats for PHM (rostral-caudal) and PHM (left-right)]. The data for columns 1 and 2 in **(c)** are respectively identical to those for columns 3 and 4 in Fig. 1c. Data are presented as mean ± s.e.m. *$P < 0.05$; **$P < 0.01$; NS, not significant; two-way repeated measures ANOVA with Dunnett's post hoc multiple comparisons test **(b)** or one-way ANOVA with Tukey's post hoc multiple comparisons test **(c)**.

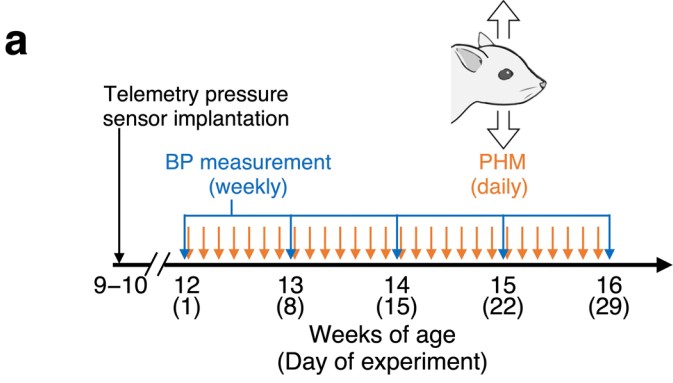

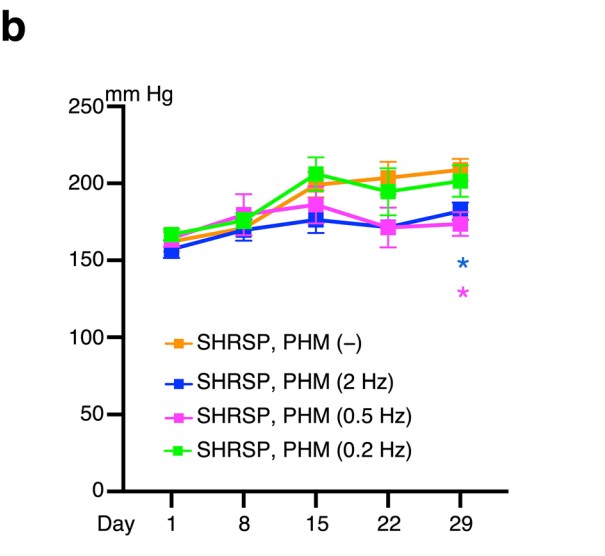

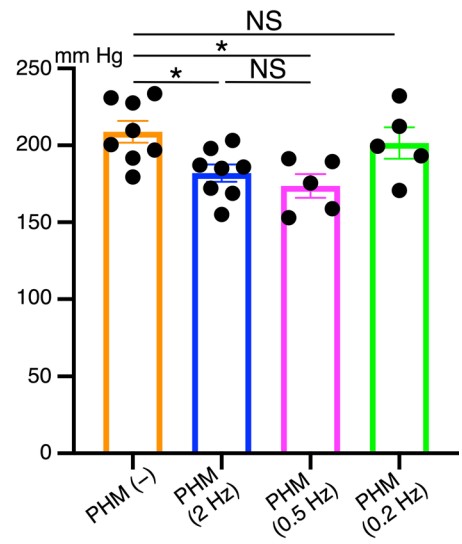

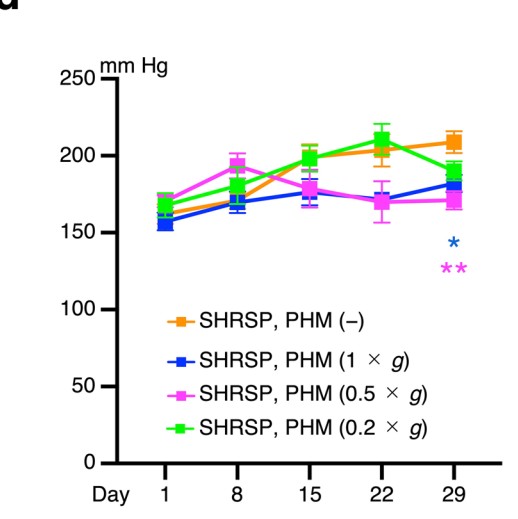

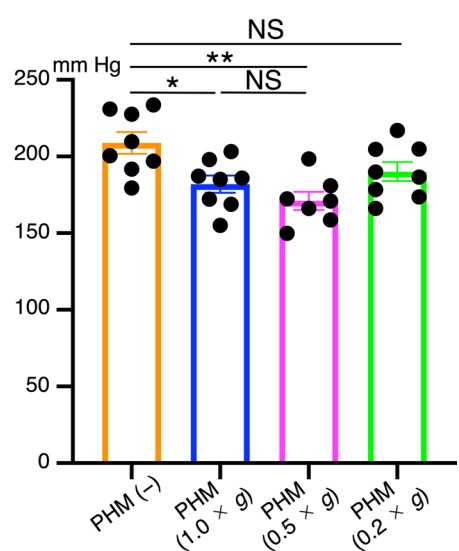

**Extended Data Fig. 3 | See next page for caption.**

**Extended Data Fig. 3 | Antihypertensive effects of PHM at different frequencies and magnitudes. a**, Schematic representation of the experimental protocol to analyse the effects of PHM on the blood pressure in SHRSPs. **b–e**, Time courses **(b,d)** and values on Day 29 **(c,e)** of MAP in SHRSPs, subjected to daily PHM with either different frequencies **(b,c)** or peak accelerations **(d,e)**. [**b,c**, $n = 8$ rats for PHM (−) and PHM (2 Hz); $n = 5$ rats for PHM (0.5 Hz) and PHM (0.2 Hz). **d,e**, $n = 8$ rats for PHM (−), PHM ($1.0 \times g$), and PHM ($0.2 \times g$); n = 7 rats for

PHM ($0.5 \times g$)]. The data for columns 1 and 2 in **(c,e)** are respectively identical to those for columns 3 and 4 in Fig. 1c. Data are presented as mean ± s.e.m. *$P < 0.05$; **$P < 0.01$; NS, not significant; two-way repeated measures ANOVA with Dunnett's post hoc multiple comparisons test **(b,d)** or one-way ANOVA with Tukey's post hoc multiple comparisons test **(c,e)**. Details of statistical analyses are provided in Supplementary Information.

**a**

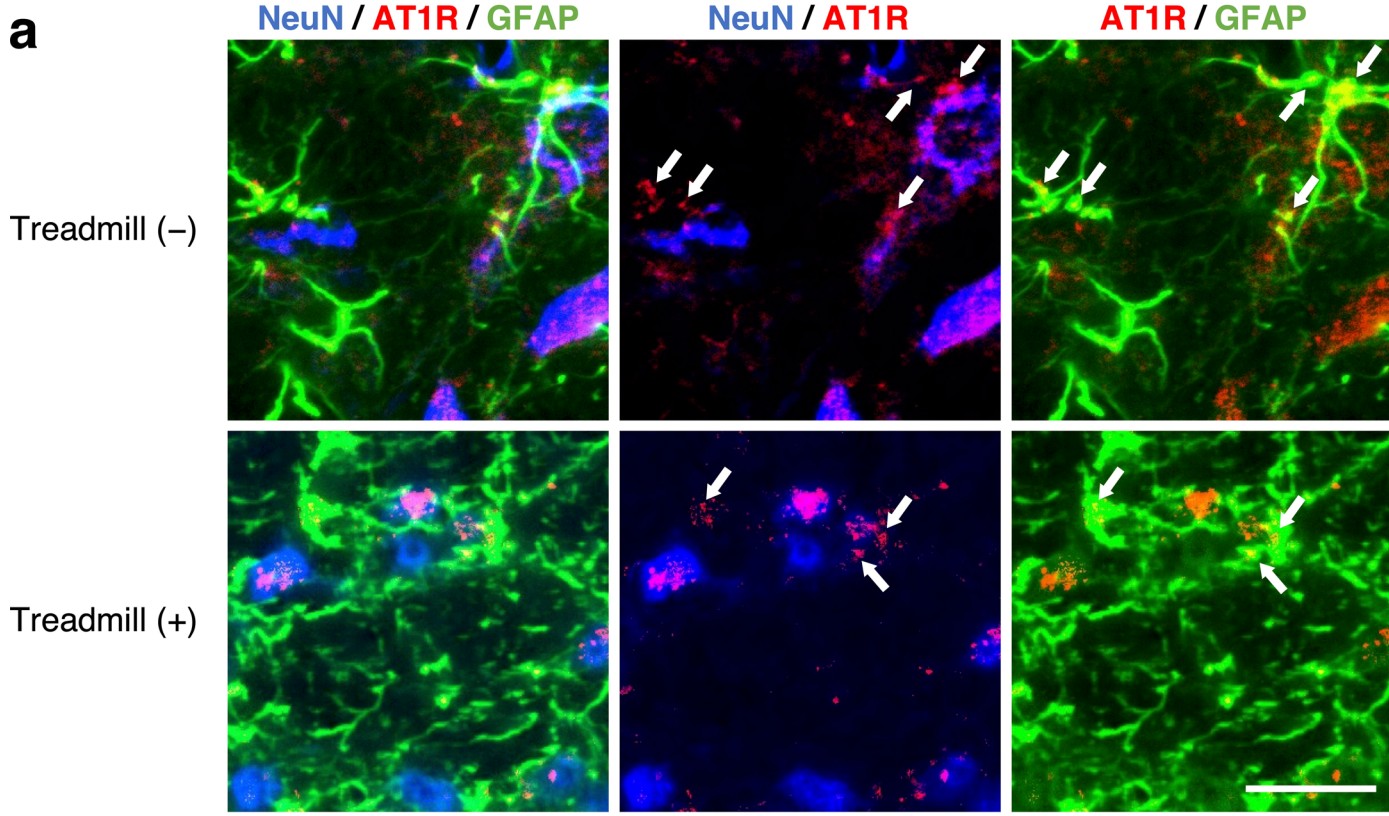

**b**

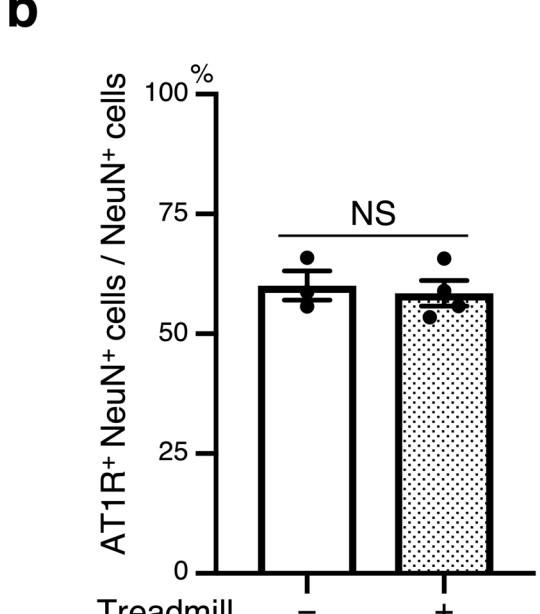

**c**

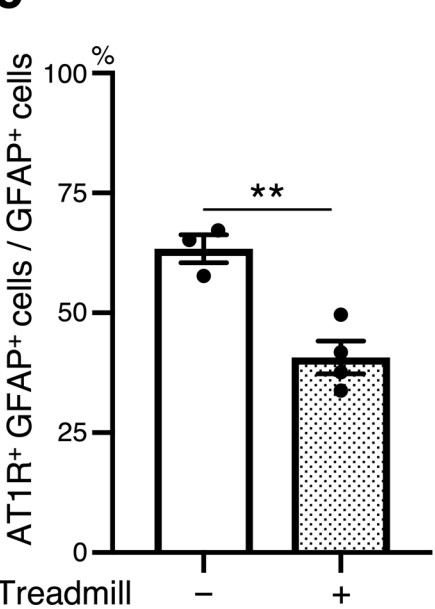

**Extended Data Fig. 4 | Treadmill running decreases AT1R expression in the RVLM astrocytes in SHRSPs. a**, Micrographic images of anti-NeuN for mature neurons (blue), anti-GFAP for glial fibrillary acidic protein-positive astrocytes (green) and anti-AT1R for angiotensin II type 1 receptor (red) immunostaining of the RVLM of SHRSPs, either placed in the static treadmill machine or subjected to treadmill running at a velocity of 20 m/min (30 min/day, 28 days). Arrows point to anti-AT1R immunosignals that overlap with anti-GFAP, but not anti-NeuN, immunosignals in merged images. Scale bar, 50 μm. Three images for treadmill (−) are from a particular section of a rat whereas those for treadmill (+) are from a different individual rat. They are representative of three [treadmill (−)] or four [treadmill (+)] rats. **b,c**, Quantification of AT1R-positive neurons **(b)** and astrocytes **(c)** in the RVLM of SHRSPs with or without 4-week treadmill running. Fifty NeuN⁺ cells and 100 GFAP⁺ cells were analysed for each rat [**b**: $P = 0.7056$. **c**: $P = 0.0048$. $n = 3$ rats for treadmill (−); $n = 4$ rats for treadmill (+)]. Data are presented as mean ± s.e.m. **$P < 0.01$; NS, not significant, unpaired two-tailed Student's $t$-test.

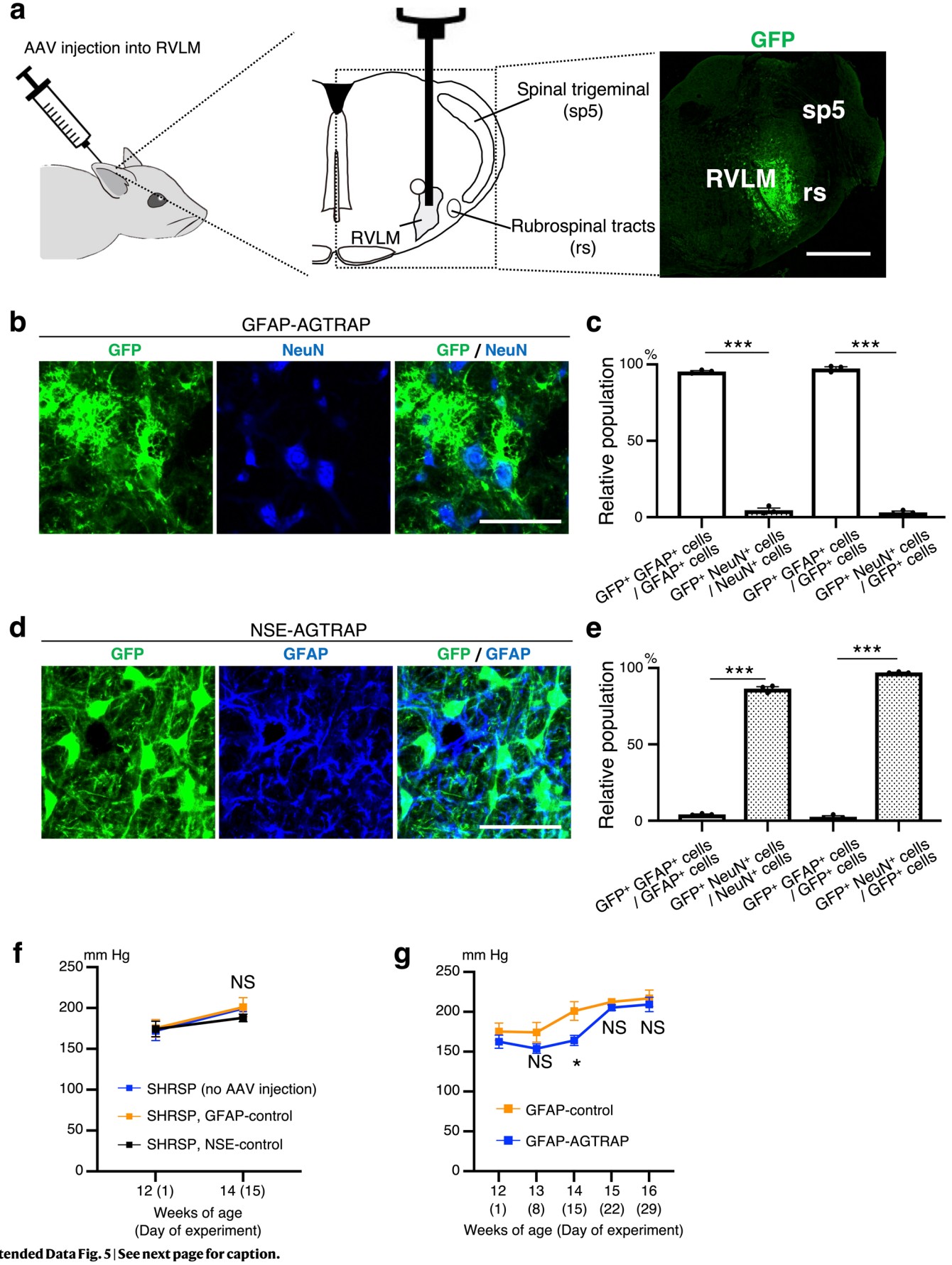

**Extended Data Fig. 5 | See next page for caption.**

**Extended Data Fig. 5 | AAV-mediated transduction of RVLM astrocytes or neurons in SHRSPs. a**, Schematic representation of the injection of AAV9 vectors to the RVLM. Micrographic image is representative of three rats analysed in Fig. 3b. GFP-derived fluorescence indicates cells expressing the transgene. Scale bar, 1 mm. **b,c**, Efficiency and specificity of astrocyte-specific expression of the transgene. **(b)** Micrographic images of GFP (green) and anti-NeuN immunostaining for mature neurons (blue) of the RVLM of SHRSPs analysed in Fig. 3b. Scale bar, 50 μm. Images are representative of three rats. **(c)** Relative populations (%) of GFP/ GFAP/ double positive (GFP⁺ GFAP⁺) or GFP/ NeuN/ double positive (GFP⁺ NeuN⁺) cells were calculated by referring their numbers to those of GFP⁺, GFAP⁺ or NeuN⁺ cells ($n$ = 3 rats for each group). **d,e**, Efficiency and specificity of neuron-specific expression of the transgene. **(d)** Micrographic images of GFP (green) and anti-GFAP immunostaining (blue) of the RVLM of SHRSPs analysed in Fig. 3c. Scale bar, 50 μm. Images are representative of three

rats. **(e)** Efficiency and specificity quantified as in **(c)**. **f**, MAP in SHRSPs injected with the control vectors. Blood pressure was measured and MAP was quantified as in Fig. 1b ($n$ = 6 rats for GFAP-control; $n$ = 7 rats for NSE-control). The data for blue line are identical to those demonstrated with blue line in Extended Data Fig. 1a. **g**, Four-week time courses of MAP. The experimental protocol was identical to that for Fig. 3 except for the observation period after the AAV injection. The data for Day 1 and Day 15 are identical to those for GFAP-control and GFAP-AGTRAP shown in Fig. 3d,e (GFAP-control: $n$ = 6 rats for Day 1, Day 8, and Day 15; $n$ = 3 rats for Day 22 and Day 29. GFAP-AGTRAP: $n$ = 7 rats for Day 1, Day 8, and Day 15; $n$ = 3 rats for Day 22 and Day 29). Data are presented as mean ± s.e.m. *$P$ < 0.05; ***$P$ < 0.001; NS, not significant; unpaired two-tailed Student's $t$-test **(c,e)** and two-way repeated measures ANOVA with Tukey's **(f)** or Bonferroni's **(g)** post hoc multiple comparisons test. Details of statistical analyses are provided in Supplementary Information.

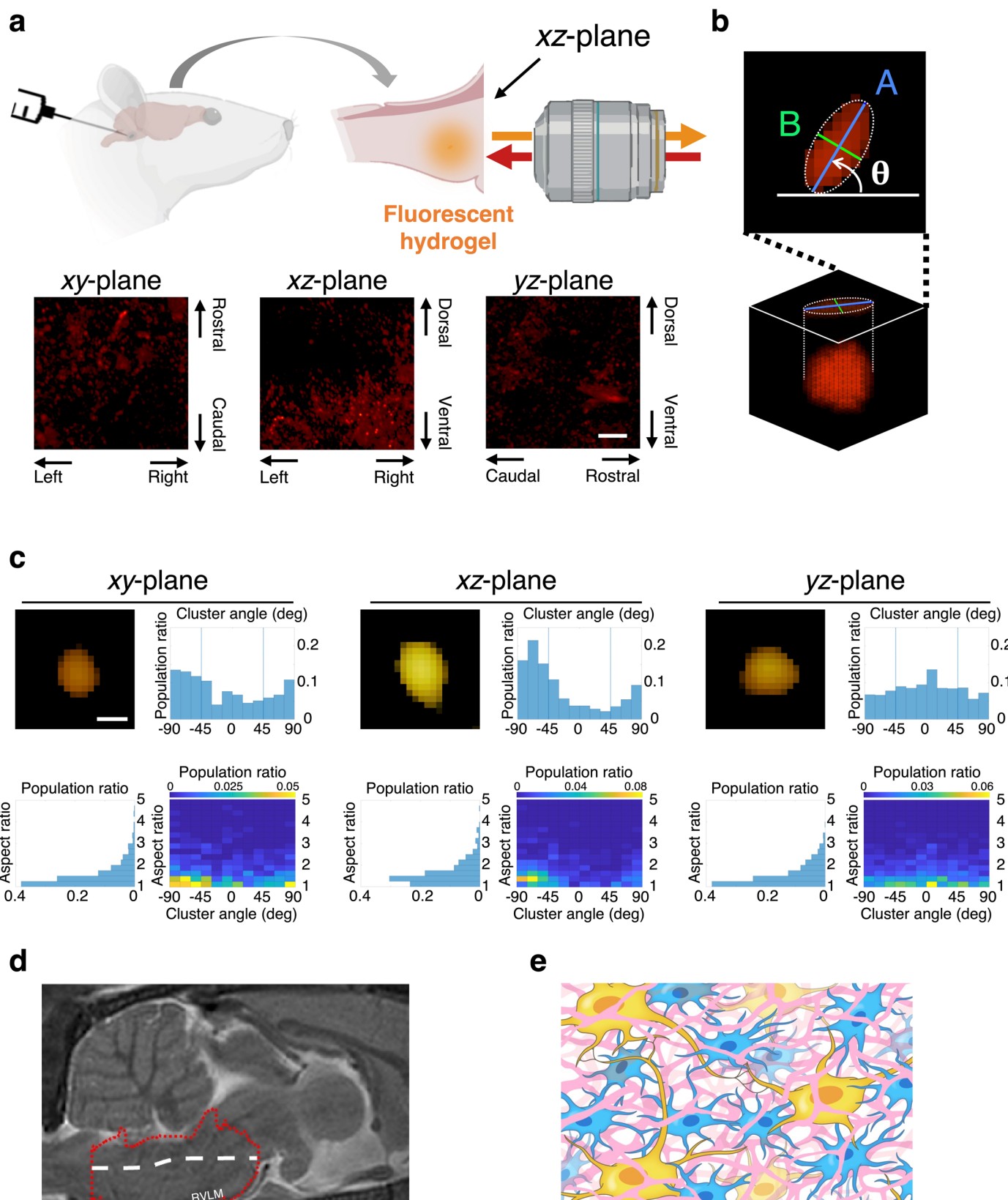

**Extended Data Fig. 6 | See next page for caption.**

**Extended Data Fig. 6 | Structure and orientation of interstitial space in the rat RVLM. a,** Multiphoton microscopic imaging of fluorescent hydrogel-introduced rat RVLM in three mutually orthogonal planes. Top: schematic illustration of hydrogel introduction and *xz*-plane imaging. Bottom: processed volume images (see Methods) of interstitial space-representing fluorescence clusters projected onto the *xy*-, *xz*-, and *yz*-planes. Scale bar, 20 μm. **b,** Determination of the cluster angle and calculation of the aspect ratio. Interstitial space-representing fluorescence clusters were extracted, and their projection and elliptic fitting on the stacked plane were conducted as described in the Methods. The cluster angle (θ) in degrees was defined and analysed as positive (counter-clockwise; see arc arrow) or negative (clockwise) from the horizontal line in each plane. The aspect ratio was calculated as A/B. **c,** Distribution of the cluster angle and aspect ratio. Volume image of a cluster of typical shape and orientation (top left), histograms of the angle distribution (top right) and aspect ratio (bottom left), and 2-D histogram of the cluster orientation/aspect ratio (bottom right) are shown for the

*xy*-, *xz*-, and *yz*-planes. From the location of seemingly unimodal peaks in the 2-D histograms for the *xy*- and *xz*-planes, the *y*- and *z*-axes appear to be dominant over the *x*-axis concerning the cluster orientation. Furthermore, the *y*-axis appears slightly dominant over the *z*-axis (loose peak on the 0–15° block in the *yz*-plane). Scale bar, 1 μm. **d,** Representative sagittal MR image of the rat brainstem (0.06 mm off the median plane). Red dotted line-indicated area and white broken line represent the lower brainstem and longest centroidal line of this part of the brain, respectively (see Methods). Note that the centroidal line is approximately along the *y*-axis with a slight rostrally upward inclination. Scale bar, 5 mm. Data shown in **(c,d)** represent at least three biologically independent samples (rats) with similar results. **e,** Diagram of the mesoscale structure/orientation of interstitial space in the rat RVLM. The interstitial space (depicted in pink) is not randomly structured but oriented approximately along the direction of the centroidal line of this part of the brain with relatively minor lateral communications. Blue and yellow cells represent astrocytes and neurons, respectively.

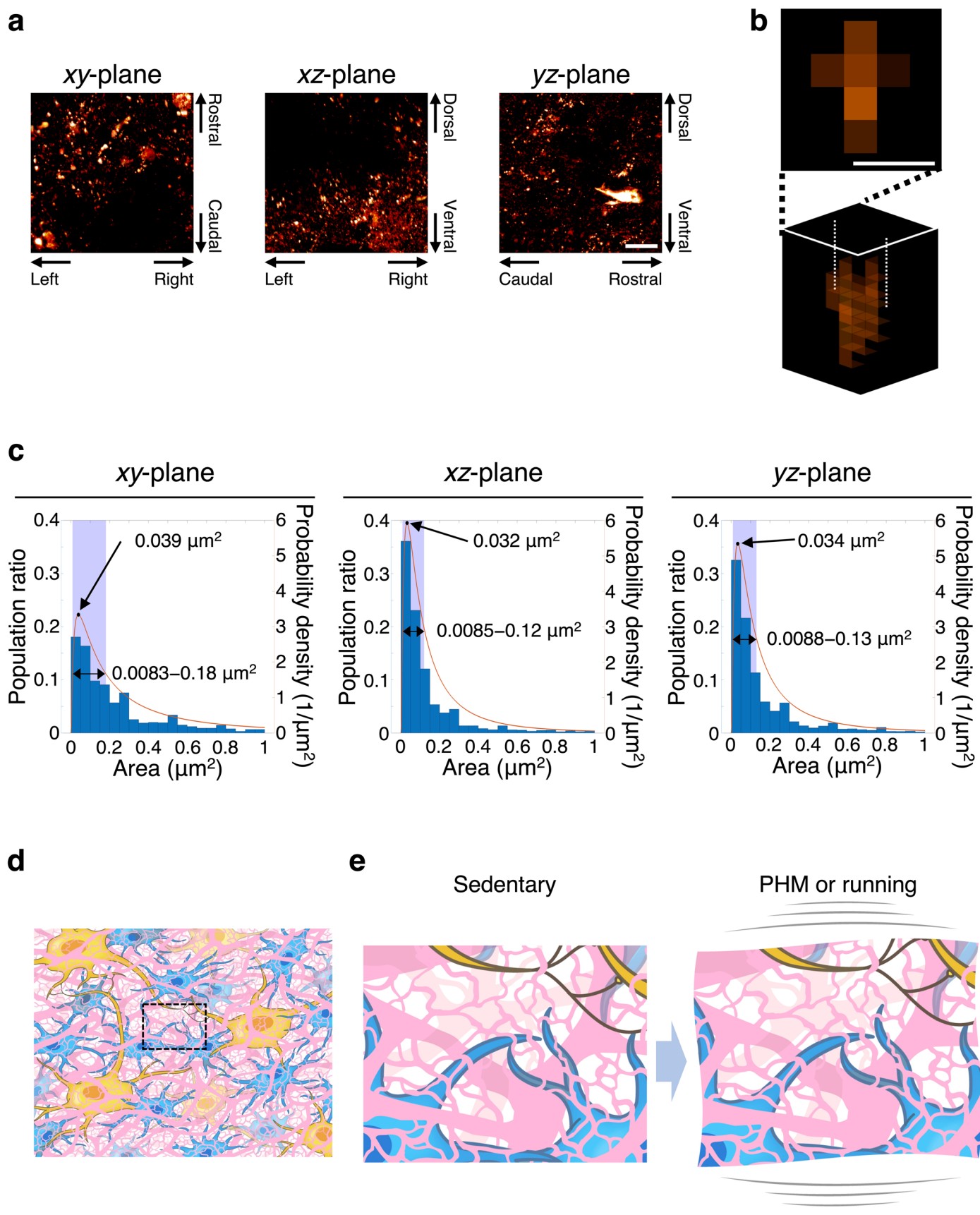

**Extended Data Fig. 7 | See next page for caption.**

**Extended Data Fig. 7 | Dimension (cross-sectional area) of interstitial space in the rat RVLM. a**, Multiphoton microscopic images of interstitial space-representing fluorescence clusters projected onto *xy*- (left), *xz*- (center), and *yz*- (right) planes. Scale bar, 20 µm. **b**, Projection of 4-µm stack images of an individual cluster. White dotted lines indicate the ends of the long axis of the cluster. Scale bar, 0.5 µm. **c**, Distribution of the cross-sectional areas of individual clusters. Red curves represent the probability densities revealed by fitting to log-normal distribution[112,113], and black circles indicate the modes. Two-way arrows are located at the half maximum, and purple zones indicate the full widths at half maximum (FWHM)[114]. **d**, Diagram of the interstitial space in the rat RVLM.

Dotted rectangle indicates the area illustrated in **(e)**. Because we analysed the interstitial space with imaging experiments based on the spread of the injected gelling fluorescent PEG solutions, we assumed the interstitial space to have a continuous structure. Nevertheless, we do not preclude the possible existence of dead ends of interstitial space. **e**, Illustration of cyclic microdeformation during PHM or treadmill running that generates small pressure changes in the rat RVLM, facilitating or promoting interstitial fluid movement in situ. Interstitial spaced is depicted in pink, and blue and yellow cells represent astrocytes and neurons, respectively, in **(d,e)**.

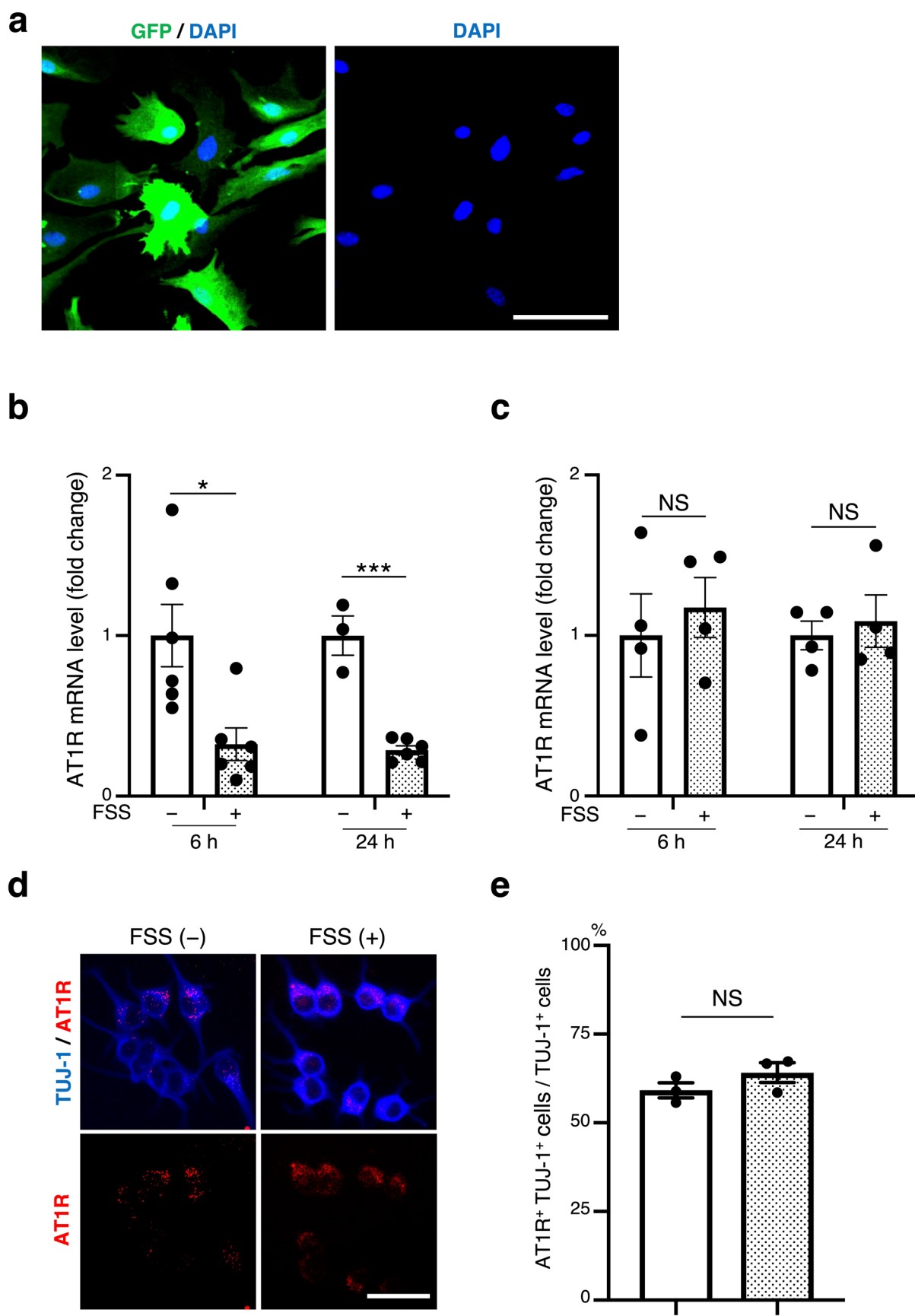

Extended Data Fig. 8 | See next page for caption.

**Extended Data Fig. 8 | Preparation of astrocyte-enriched primary culture, and lack of a decreasing effect of FSS on AT1R expression in cultured neuronal cells. a**, Representative micrographic image of anti-GFP immunostaining (green) and DAPI staining (blue) of astrocyte-enriched culture prepared from the astrocyte-GFP mice. Scale bar, 50 μm. Note that most of the cells are GFP-positive. **b,c**, Effects of FSS on the AT1R mRNA expression in astrocytes and Neuro2A cells. AT1R mRNA expression in cultured astrocytes **(b)** and Neuro2A cells **(c)** 6 or 24 h after 30-min FSS application (0.7 Pa, 0.5 Hz) was analysed as in Fig. 5a [**b**: $P = 0.0116$ for 6 h, $P < 0.0001$ for 24 h. $n = 6$ for 6 h; $n = 3$ for 24 h/FSS (−); $n = 6$ for 24 h/FSS (+). **c**: $P = 0.6065$ for 6 h, $P = 0.6490$ for 24 h.

$n = 4$ for each group]. **d**, Microscopic images of anti-AT1R (red) and anti-TUJ-1 (blue) immunostaining of Neuro2A cells, either left unexposed or exposed to pulsatile FSS (0.7 Pa, 0.5 Hz, 30 min), and fixed 24 h after the intervention. Images are representative of three biologically independent experiments with similar results. Scale bar, 50 μm. **e**, Relative population of AT1R/TUJ-1/ double positive (AT1R$^+$ TUJ-1$^+$) cells quantified as a ratio to total TUJ-1-positive (TUJ-1$^+$) cells in each sample ($P = 0.2308$. More than 100 TUJ-1-positive cells were analysed in each sample. $n = 3$). Data are presented as mean ± s.e.m. *$P < 0.05$; ***$P < 0.001$; NS, not significant; unpaired two-tailed Student's $t$-test **(b,c,e)**.

**a**

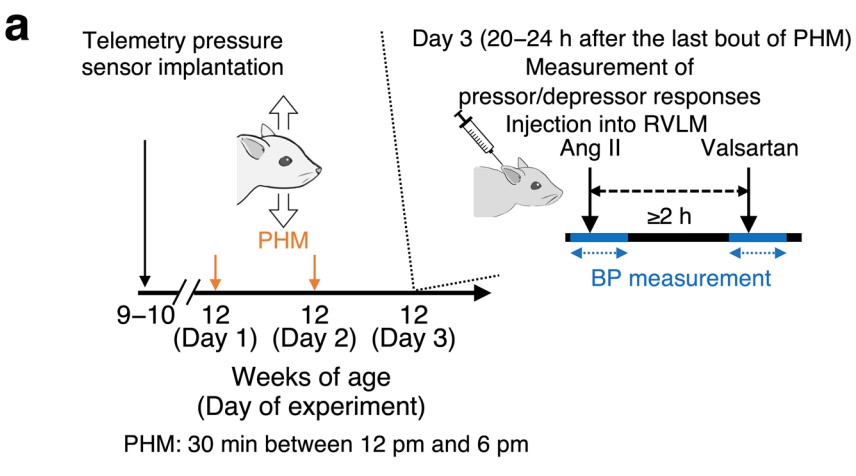

**b**

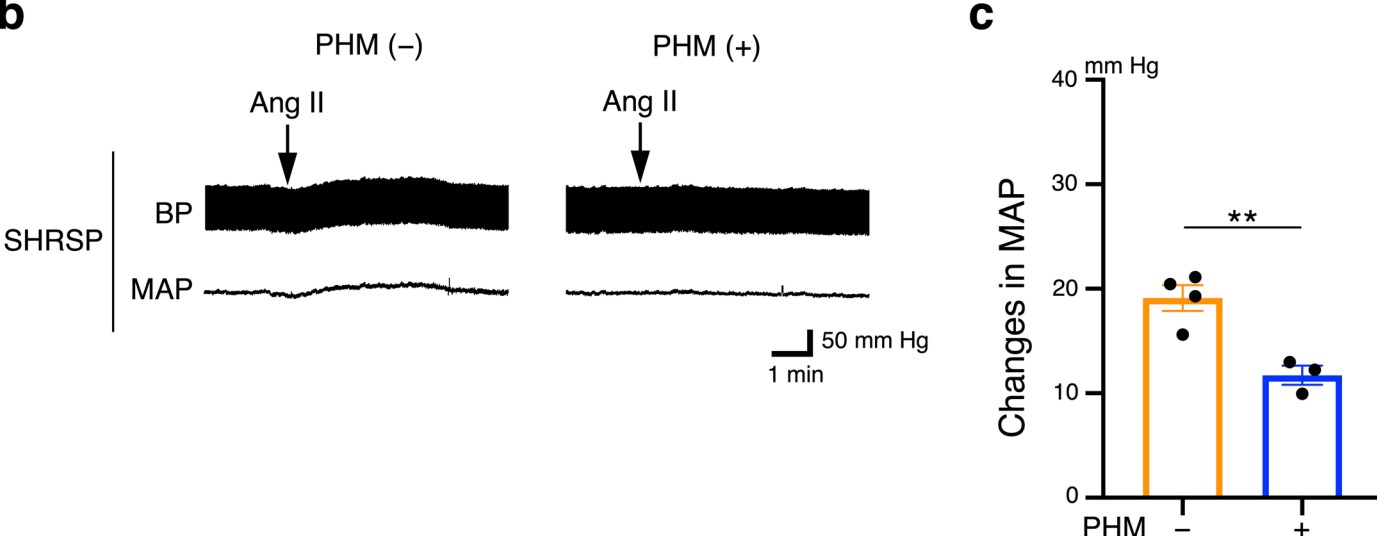

**d**

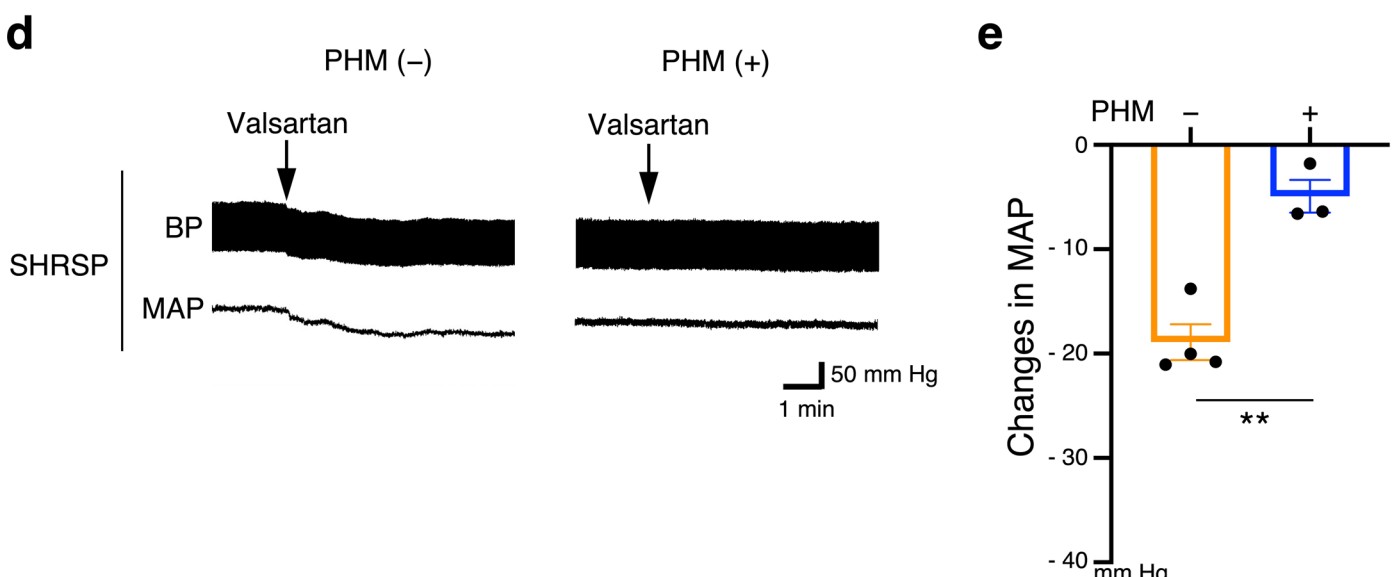

**Extended Data Fig. 9 | Two-day PHM alleviates the sensitivity of RVLM in SHRSPs to angeotensin II or valsartan. a**, Schematic representation of the experimental protocol. Pressor and depressor responses were analysed as in Fig. 2. **b–e**, Pressor (**b,c**) and depressor (**d,e**) responses in SHRSPs with and without 2-day PHM. (**b,d**) Representative trajectories of the blood pressure (top in each panel) and MAP (bottom in each panel). Arrows point to the time of the initiation of RVLM injection of Ang II (**b**) or valsartan (**d**). Right-angled scale bars, 1 min / 50 mm Hg. (**c, e**) MAP change caused by Ang II (**c**) or valsartan (**e**) injection [**c**: $P = 0.0064$. **e**: $P = 0.0022$. $n = 4$ rats for PHM (−); $n = 3$ rats for PHM (+)]. Data are presented as mean ± s.e.m. **$P < 0.01$; unpaired two-tailed Student's $t$-test.

**a**

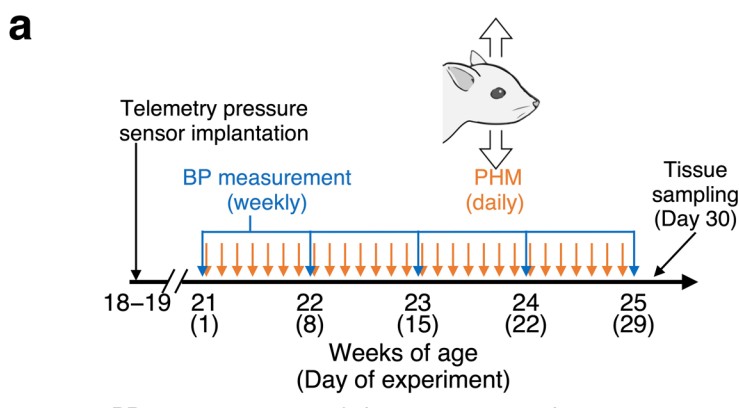

Telemetry pressure sensor implantation

BP measurement (weekly)

PHM (daily)

Tissue sampling (Day 30)

18–19 | 21 (1) | 22 (8) | 23 (15) | 24 (22) | 25 (29)

Weeks of age (Day of experiment)

BP measurement: 30 min between 10 am and 12 pm

PHM: 30 min between 12 pm and 6 pm

**b**

NS

mm Hg

PHM (−)
PHM (+)

Day 1 8 15 22 29

**c**

NS

mm Hg

PHM − +

**d**

NeuN / AT1R / GFAP    NeuN / AT1R    AT1R / GFAP

PHM (−)

PHM (+)

**e**

*

AT1R⁺ GFAP⁺ cells / GFAP⁺ cells

PHM − +

**f**

NS

AT1R⁺ NeuN⁺ cells / NeuN⁺ cells

PHM − +

**Extended Data Fig. 10 | See next page for caption.**

**Extended Data Fig. 10 | PHM does not lower blood pressure in SHRSPs, but decreases AT1R expression in RVLM astrocytes during the plateau phase of hypertension development. a**, Schematic representation of the experimental protocol to analyse the effects of PHM. **b,c**, Time courses (**b**) and values on Day 29 (**c**) of MAP in SHRSPs, subjected to either daily PHM or anaesthesia only (**b**: $P > 0.9999$ for Day 15, Day 22, and Day 29. **c**: $P = 0.9167$. $n = 8$ rats for each group). **d**, Micrographic images of anti-GFAP for glial fibrillary protein-positive astrocytes (green), anti-AT1R for angiotensin II type 1 receptor (red), and anti-NeuN for mature neurons (blue) immunostaining of the RVLM of SHRSPs, either left sedentary (top) or subjected to PHM (bottom) under anaesthesia (30 min/day, 28 days). Arrows point to anti-AT1R immunosignals that overlap with anti-GFAP, but not anti-NeuN, immunosignals in merged images. Three images for PHM (−) are from a particular section of a rat whereas those for PHM (+) are from a different individual rat. Scale bar, 50 µm. Images are representative of three rats. **e,f**, Quantification of AT1R-positive astrocytes (**e**) and neurons (**f**) in the RVLM of SHRSPs, either left sedentary or subjected to PHM. One-hundred GFAP-positive (GFAP⁺) cells (**e**) and 50 NeuN-positive (NeuN⁺) cells (**f**) were analysed for each rat (**e**: $P = 0.0173$. **f**: $P = 0.7812$. $n = 3$ rats for each group). Data are presented as mean ± s.e.m. *$P < 0.05$; NS, not significant; two-way repeated measures ANOVA with Bonferroni's post hoc multiple comparisons test (**b**) or unpaired two-tailed Student's *t*-test (**c,e,f**).

# Reporting Summary

## Statistics

For all statistical analyses, confirm that the following items are present in the figure legend, table legend, main text, or Methods section.

| n/a | Confirmed | |
|---|---|---|
| ☐ | ☒ | The exact sample size (*n*) for each experimental group/condition, given as a discrete number and unit of measurement |
| ☐ | ☒ | A statement on whether measurements were taken from distinct samples or whether the same sample was measured repeatedly |
| ☐ | ☒ | The statistical test(s) used AND whether they are one- or two-sided *Only common tests should be described solely by name; describe more complex techniques in the Methods section.* |
| ☐ | ☒ | A description of all covariates tested |
| ☐ | ☒ | A description of any assumptions or corrections, such as tests of normality and adjustment for multiple comparisons |
| ☐ | ☒ | A full description of the statistical parameters including central tendency (e.g. means) or other basic estimates (e.g. regression coefficient) AND variation (e.g. standard deviation) or associated estimates of uncertainty (e.g. confidence intervals) |
| ☐ | ☒ | For null hypothesis testing, the test statistic (e.g. *F*, *t*, *r*) with confidence intervals, effect sizes, degrees of freedom and *P* value noted *Give P values as exact values whenever suitable.* |
| ☒ | ☐ | For Bayesian analysis, information on the choice of priors and Markov chain Monte Carlo settings |
| ☒ | ☐ | For hierarchical and complex designs, identification of the appropriate level for tests and full reporting of outcomes |
| ☒ | ☐ | Estimates of effect sizes (e.g. Cohen's *d*, Pearson's *r*), indicating how they were calculated |

*Our web collection on statistics for biologists contains articles on many of the points above.*

## Software and code

Policy information about availability of computer code

| Data collection | Microsoft Excel (Version 16.65, Microsoft, Redmond, WA), LabChart (Version 8, ADInstruments, Bella Vista, Australia), Spike2 software (Version 7.20, Cambridge Electronic Design, Cambridge, UK). |
|---|---|
| Data analysis | Prism software (Version 8, GraphPad Software, San Diego, CA), R software (Version 4.1.2, https://www.r-project.org/), MATLAB (Version 2021a, Math Works, Natic, MA). The MATLAB scripts to compute the cluster-angle and aspect-ratio distributions from multiphoton microscope images (Extended Data Fig. 6c and 7c) and to compute the lower brainstem and longest centroidal line of this part of the brain (Extended Data Fig. 6d) are available at https://doi.org/10.5281/zenodo.7936475. |

For manuscripts utilizing custom algorithms or software that are central to the research but not yet described in published literature, software must be made available to editors and reviewers. We strongly encourage code deposition in a community repository (e.g. GitHub). See the Nature Portfolio guidelines for submitting code & software for further information.

## Data

Policy information about availability of data

All manuscripts must include a data availability statement. This statement should provide the following information, where applicable:

- Accession codes, unique identifiers, or web links for publicly available datasets
- A description of any restrictions on data availability
- For clinical datasets or third party data, please ensure that the statement adheres to our policy

> The main data supporting the findings of this study are available within the paper and its Supplementary Information. The raw data generated during the study are too large to be publicly shared, yet they are available for research purposes from the corresponding author on reasonable request. The template atlas of the Fischer 344 rat brain is available at https://doi.org/10.5281/zenodo.3900544.

## Research involving human participants, their data, or biological material

Policy information about studies with human participants or human data. See also policy information about sex, gender (identity/presentation), and sexual orientation and race, ethnicity and racism.

| | |
|---|---|
| Reporting on sex and gender | We used only male rats for our animal studies (see below). However, because we did not aim to investigate only male-specific matters, we included human participants of both sexes in our clinical study. Vertically oscillating chair riding had antihypertensive effects in both male and female humans (Supplementary Fig. 7b). We list the participants' sex in Supplementary Tables 2 and 3. |
| Reporting on race, ethnicity, or other socially relevant groupings | The eligibility to participate in the clinical study did not involve any particular race, ethnicity, or other socially relevant groupings. However, all the applicants for this study were Japanese, and hence we did not control for race and ethnicity as possible confounding variables in our analyses. |
| Population characteristics | Both females and males were considered eligible if they were 20 years old or older and confirmed to have 130–160 mm Hg of systolic blood pressure at the time of interview for informed consent and eligibility check. Subjects with mental or psychological illnesses, history or presence of cardiovascular events, history or presence of severe dysfunction/disorder of liver, kidney, lung, gastrointestinal tract, and spine, or presence of acute injuries/diseases (such as recent traumas and infectious diseases) were excluded, with the exception of those who were given permission for participating in this study from their primary care physicians. Two male and three female subjects aged 37–60 years participated in the study of protocol 1. Sixteen males and 14 females aged 23–85 years participated in the studies of protocols 2 or 3. Three males and two females participated in the studies of both protocols 2 and 3. |
| Recruitment | For the study of protocol 1, participants were recruited through printed advertisements placed in Iwai Orthopaedic Medical Hospital and the affiliated health services facility of Iwai Medical Foundation (Iwai Keiaien). For the studies of protocols 2 and 3, participants were recruited through printed advertisements placed in the National Rehabilitation Center for Persons with Disability Hospital and the Tokorozawa Heart Center. There may be potentially self-selection bias related to the recruitment of the participants in our human studies. For example, the participants may have had more interest in their health condition as compared to ordinary people. Although this may have influenced the results of our human study, antihypertensive or sympathoinhibitory effects were not observed in non-oscillating chair riding. This supports the notion that vertically oscillating chair riding is specifically effective. |
| Ethics oversight | The study of protocol 1 was approved by the Ethics Committee of the Iwai Medical Foundation, and the studies of protocols 2 and 3 were approved by the Ethics Committee of the National Rehabilitation Center for Persons with Disabilities. All participants in our human studies provided written informed consent. |

Note that full information on the approval of the study protocol must also be provided in the manuscript.

# Field-specific reporting

Please select the one below that is the best fit for your research. If you are not sure, read the appropriate sections before making your selection.

☒ Life sciences  ☐ Behavioural & social sciences  ☐ Ecological, evolutionary & environmental sciences

For a reference copy of the document with all sections, see nature.com/documents/nr-reporting-summary-flat.pdf

# Life sciences study design

All studies must disclose on these points even when the disclosure is negative.

| | |
|---|---|
| Sample size | No statistical methods were used to predetermine sample size. At least three biologically independent experimental replicates were performed for statistical analyses, according to standard scientific conventions. In the case of experiments using hypertensive rats, we in principle used similar sizes to those in published experiments to analyse the effects of exercise on hypertensive rats (Kishi et al. Clin Exp Hypertens 2012; Bertagnolli et al. Am J Hypertens 2008; Agarwal et al. Hypertension 2009). We tried to reach a conclusion for each individual experiment, using the smallest sample size possible. |

| | |
|---|---|
| Data exclusions | No data were excluded. |
| Replication | Information on replication, or on the independent performance of the experiments and measurements, is provided in the figure legends and Methods. Unless stated otherwise, the experiments or measurements were replicated or performed at least three times independently. |
| Randomization | All the animals and cells were randomly assigned to experimental groups. |
| Blinding | Blinding was irrelevant to the study. All animal and cell experiments were carried out by researchers who prepared and analysed the samples. |

# Reporting for specific materials, systems and methods

We require information from authors about some types of materials, experimental systems and methods used in many studies. Here, indicate whether each material, system or method listed is relevant to your study. If you are not sure if a list item applies to your research, read the appropriate section before selecting a response.

| Materials & experimental systems | | Methods | |
|---|---|---|---|
| **n/a** | **Involved in the study** | **n/a** | **Involved in the study** |
| ☐ | ☒ Antibodies | ☒ | ☐ ChIP-seq |
| ☐ | ☒ Eukaryotic cell lines | ☒ | ☐ Flow cytometry |
| ☒ | ☐ Palaeontology and archaeology | ☐ | ☒ MRI-based neuroimaging |
| ☐ | ☒ Animals and other organisms | | |
| ☒ | ☐ Clinical data | | |
| ☒ | ☐ Dual use research of concern | | |
| ☒ | ☐ Plants | | |

## Antibodies

| | |
|---|---|
| Antibodies used | Mouse monoclonal anti-GFAP cloneGA5 (cloneGA5, MAB360; Millipore) ; rabbit polyclonal anti-GFAP (Z0334; Dako) ; chicken polyclonal anti-GFAP (ab4674; Abcam) ; rabbit polyclonal anti-cleaved caspase-3 (9661; Cell Signaling Technology) ; mouse monoclonal anti-NeuN (clone A69, MAB377; Millipore) ; rabbit polyclonal anti-NeuN (ABN78; Millipore) , rabbit polyclonal anti-AT1R (HPA003596; Sigma-Aldrich) ; rabbit polyclonal anti-AGTRAP (HPA044120; Sigma-Aldrich) ; rabbit polyclonal anti-GFP (598; MBL) ; chicken polyclonal anti-GFP (ab13970; Abcam) ; mouse monoclonal anti-TUJ-1 (clone 2G10, ab78078; Abcam); rabbit polyclonal anti-TNF-alpha (ab66579; Abcam) ; rabbit polyclonal anti-IL-1beta (ab9722; Abcam); rabbit polyclonal anti-GAPDH (5174; Cell Signaling Technology). Secondary antibodies conjugated with Alexa Fluor 350, 488, 568, 633, and 647 (Thermo Fisher Scientific). Horseradish peroxidase-conjugated anti-rabbit IgG (H + L) secondary antibody (W401B; Promega). |
| Validation | All antibodies were obtained from commercial manufacturers. Anti-AT1R antibody was validated by verifying the consistency with anti-hemagglutinin (HA) (mouse monoclonal, clone HA.C5, ab18181; Abcam) immunostaining of HA-tagged AT1R exogenously expressed in HEK293 cells, which do not express AT1R endogenously. All the other antibodies were employed based on the validation statements provided on the manufacturers' websites. |

## Eukaryotic cell lines

Policy information about cell lines and Sex and Gender in Research

| | |
|---|---|
| Cell line source(s) | Neuro2A cell line (mouse neuroblastoma), which was provided by Dr. T. Yokota (Tokyo Medical and Dental University). |
| Authentication | Although the expression of neuronal proteins was confirmed, the cell line used was not authenticated. |
| Mycoplasma contamination | The cell line was not tested for mycoplasma contamination. However, when we conducted DAPI staining, we did not see a dotted stain around the nuclei, which is characteristic of mycoplasma contamination. |
| Commonly misidentified lines (See ICLAC register) | No commonly misidentified cell lines were used. |

## Animals and other research organisms

Policy information about studies involving animals; ARRIVE guidelines recommended for reporting animal research, and Sex and Gender in Research

| | |
|---|---|
| Laboratory animals | Male SHRSP/Izm and WKY/Izm rats were provided by the Disease Model Cooperative Association (Kyoto, Japan) and used for experiments at the age of 9 to 21 weeks after acclimation for at least 1 week. The astrocyte-GFP mice (Aldh1L1-GFP mice) obtained from GENSAT and bred in-house were housed with free access to water and standard rodent chow under a 12/12 h light–dark cycle with controlled temperature (22–24°C) and humidity (50–60%). One-day-old or two-day-old mice of both genders were used for astrocyte preparation. |

| Wild animals | The study did not involve wild animals. |
|---|---|
| Reporting on sex | We used only male rats for our animal studies, although sex is an important variable for nearly all diseases, including hypertension. This was because we intended to preclude or minimize the potential influence of oestrogen and progesterone, both of which basically act protectively on cardiovascular systems, including the heart and endothelium. Perhaps for the same reason, in many or even most of the animal experiments in previous studies investigating the pathogenesis of cardiovascular diseases, male animals have been used unless there is particular reason to analyse female animals. In particular, we intended to be consistent with previous studies that investigated antihypertensive effects of treadmill running in male SHRs or SHRSPs. |
| Field-collected samples | The study did not involve samples collected from the field. |
| Ethics oversight | The animal study was approved by the Animal Care and Use Committee of the National Rehabilitation Center for Persons with Disabilities. |

Note that full information on the approval of the study protocol must also be provided in the manuscript.

# Magnetic resonance imaging

## Experimental design

| Design type | — |
|---|---|
| Design specifications | — |
| Behavioral performance measures | — |

## Acquisition

| Imaging type(s) | Structural |
|---|---|
| Field strength | 7 Tesla |
| Sequence & imaging parameters | 2D rapid acquisition with relaxation enhancement (RARE). Parameters for the sagittal T2-RARE sequence were as follows; echo time (TE): 33 ms, repetition time (TR): 3600 ms, slice thickness (SL): 0.75 mm, field of view (FOV): 38.4 × 38.4 mm, voxel: 0.15 × 0.15 × 0.75 mm3. RARE factor:8, number of Averages: 4. |
| Area of acquisition | Whole brain |

Diffusion MRI ☐ Used ☒ Not used

## Preprocessing

| Preprocessing software | SPM12 on MATLAB2021a |
|---|---|
| Normalization | We mainly used inverse normalization. First, we used the 'Old normalize' tool of SPM12 to compute individual data normalization. Then the brainstem template was inverse-normalized into each individual volume. |
| Normalization template | An MRI-Derived Neuroanatomical Atlas of the Fischer 344 Rat Brain (Version v4) [Data set]. Zenodo. https://doi.org/10.5281/zenodo.3900544. |
| Noise and artifact removal | — |
| Volume censoring | — |

## Statistical modeling & inference

| Model type and settings | — |
|---|---|
| Effect(s) tested | — |

Specify type of analysis: ☐ Whole brain ☒ ROI-based ☐ Both

Anatomical location(s) Brainstem

| Statistic type for inference<br>(See Eklund et al. 2016) | — |
|---|---|
| Correction | — |

## Models & analysis

| n/a | Involved in the study |
|-----|------------------------|
| ☒ | ☐ Functional and/or effective connectivity |
| ☒ | ☐ Graph analysis |
| ☒ | ☐ Multivariate modeling or predictive analysis |

