## [Peer Review File · Nature Biomedical Engineering]

Interstitial-fluid shear stresses induced by vertically oscillating head motion lower blood pressure in hypertensive rats and humans

Corresponding author: Yasuhiro Sawada

Editorial note

This document includes relevant written communications between the manuscript's corresponding author and the editor and reviewers of the manuscript during peer review. It includes decision letters relaying any editorial points and peer-review reports, and the authors' replies to these (under 'Rebuttal' headings). The editorial decisions are signed by the manuscript's handling editor, yet the editorial team and ultimately the journal's Chief Editor share responsibility for all decisions.

Any relevant documents attached to the decision letters are referred to as **Appendix #**, and can be found appended to this document. Any information deemed confidential has been redacted or removed. Earlier versions of the manuscript are not published, yet the originally submitted version may be available as a preprint. Because of editorial edits and changes during peer review, the published title of the paper and the title mentioned in below correspondence may differ.

Correspondence

Thu 30 Jul 2020

Decision on Article nBME-20-1499

Dear Prof Sawada,

Thank you again for submitting to *Nature Biomedical Engineering* your manuscript, "Mechanical impact on the head has an antihypertensive effect". The manuscript has been seen by 3 experts, whose reports you will find at the end of this message. You will see that although the reviewers have some good words for the work, they articulate concerns about the degree of support for the claims, and in this regard provide useful suggestions for improvement. We hope that with significant further work you can address the criticisms and convince the reviewers of the merits of the study. In particular, we would expect that a revised version of the manuscript provides:

*Clarification of the causative mechanisms for the effects of the passive head movements with further experimental evidence, as suggested by all the Reviewers.

*Clarification of the calculation of shear forces during running, and further justification that fluid shear stress influences blood pressure with experimental evidence, as suggested by all the Reviewers.

*Further time control data to align the in vitro and in vivo experiments as suggested by Rev #2.

*Clarification of the data pertaining to sympathetic tone, with experimental evidence as suggested by Reviewer #2.

*Clarification of the statistics as suggested by Rev #3.

When you are ready to resubmit your manuscript, please upload the revised files, a point-by-point rebuttal to the comments from all reviewers, the (revised, if needed) reporting summary, and a cover letter that explainsthe main improvements included in the revision and responds to any points highlighted in this decision.

Please follow the following recommendations:

- * Clearly highlight any amendments to the text and figures to help the reviewers and editors find and understand the changes (yet keep in mind that excessive marking can hinder readability).
- * If you and your co-authors disagree with a criticism, provide the arguments to the reviewer (optionally, indicate the relevant points in the cover letter).
- * If a criticism or suggestion is not addressed, please indicate so in the rebuttal to the reviewer comments and explain the reason(s).
- * Consider including responses to any criticisms raised by more than one reviewer at the beginning of the rebuttal, in a section addressed to all reviewers.
- * The rebuttal should include the reviewer comments in point-by-point format (please note that we provide all reviewers will the reports as they appear at the end of this message).
- * Provide the rebuttal to the reviewer comments and the cover letter as separate files.

We hope that you will be able to resubmit the manuscript within 25 weeks from the receipt of this message. If this is the case, you will be protected against potential scooping. Otherwise, we will be happy to consider a revised manuscript as long as the significance of the work is not compromised by work published elsewhere or accepted for publication at *Nature Biomedical Engineering*. Because of the COVID-19 pandemic, should you be unable to carry out experimental work in the near future we advise that you reply to this message with a revision plan in the form of a preliminary point-by-point rebuttal to the comments from all reviewers that also includes a response to any points highlighted in this decision. We should then be able to provide you with additional feedback.

We hope that you will find the referee reports helpful when revising the work. Please do not hesitate to contact me should you have any questions.

Best wishes,

Michelle

Dr Michelle Korda
Senior Editor, Nature Biomedical Engineering

Reviewer #1 (Report for the authors (Required)):

In this interesting paper, the authors claim that passive head motion of hypertensive rats generated at their heads during treadmill running at a moderate velocity decreases the expression of angiotensin II type 1 receptor (AT1R) in astrocytes in their rostral ventrolateral medulla, which in turn decreases blood pressure. Overall, the paper is compelling and the authors are to be commended for executing a large number of complementary, multidisciplinary experiments leading to significant amounts of data. If the authors' hypothesis is correct, this work will have profound implications for the study and treatment of hypertension in the context of cardiovascular disease. However, because the central ideal is indeed exciting, the burden of proof is high and as such, several parts of the paper have significant flaws that need to be addressed.

1) First, while the overall scientific premise is interesting, could all of this be predominantly due to a classic baroreceptor response? Indeed, when the head is being moved up and down repeatedly, carotid and aortic stretching can occur, which will lead to decrease in blood pressure. While the AT1R seems to be involved, could that signaling pathway be dwarfed by the parasympathetic baroreceptor response?

2) The PHM needs to be better characterized in this paper. What happens when PHM directionality, frequency, and amplitude all are modulated systemically? How do these variables each affect the duration of the decrease in BP? Are there plateau effects with each variable? Rigorous characterization of these parameters could help make the authors' cases.

3) The human experiments comprising vertically oscillating chair riding utilize a different mode of force application, which could confound results. This needs to be addressed and discussed

4) The experiments involving the conjecture that effects of interstitial fluid pressure on astrocytes is a major part of the underlying mechanism unfortunately are more correlative and demonstrate more of an association than causation as many confounders can be taking place. First, there is little evidence that astrocytes experience the changes in interstitial pressure as postulated in Figure 4. The contrast agent data demonstrates effects that occurs at a much higher size scale than at the cellular level. In fact, it is unclear that astrocytes within the complex anatomy of the medulla even are exposed to the contrast agent let alone the hydrodynamic forces thereof. In addition, the in vitro fluid stress experiments occur in a system that again is non-physiologic and do not recapitulate the in vivo conditions of the astrocytes' microenvironment. Finally, the PEG injection experiments may be causing other effects aside from modulating AT1R expression and therefore obvious issues that need to be excluded are local inflammation or increased intracranial pressure from the gel, which could affect blood pressure. All of these issues need to be definitively addressed and/or clarified.

5) Importantly, the authors should better place their work in the context of others already published in the literature. In fact, if done correctly, this can better bolster their argument.

Minor points

1) The phrase "Mechanical Impact" in the title implies blunt head trauma, which is not the case and therefore should be removed or rephrased.

2) Limitations of the animal hypertension model should be mentioned and the model used should be better justified

3) Is there attenuation of the deleterious effects of the mouse when PHM is applied? Is stroke incidence decreased?

Reviewer #2 (Report for the authors (Required)):

This is an ambitious study that proposes a novel mechanism to explain the antihypertensive effects of exercise in arterial hypertension. The authors conclude from a combination of in vivo and in vitro studies in stroke-prone spontaneously hypertensive rats (SHRSP), an animal model for primary hypertension, as well as an intervention study in hypertensive patients, that interstitial fluid movements triggered by physical movements in the rostral ventrolateral medulla (RVLM), a brain area important for the regulation of blood pressure, lower sympathetic tone via an AT-1 receptor (AT1R) dependent mechanism and thereby induce a reduction in blood pressure.

Although this new idea is appealing, I do not find that the data presented sufficiently support the overall hypothesis. I also have some conceptual difficulties in reconciling it with the existing basic scientific and clinical evidence.

Here is the list of my questions and concerns (all major), which I hope the authors will address.

1. The experimental results presented by the authors show that passive head movements (PHM), which are calculated to replicate the acceleration effects of running, have quantitatively comparable effects on blood pressure to treadmill running in SHRSP. However, this is only correlative evidence and does not indicate any causal relationships. The relevance of direct mechanical effects on the RVLM may be better tested by investigating whether pre-treatment by PHM or prevention of interstitial fluid movements can significantly reduce the antihypertensive effects of running in SHRSP.

2. All in vivo experiments in SHRSP were carried out when the blood pressure of the animals was still rising considerably. In fact, PHM appears to prevent a further increase in blood pressure rather than exerting a blood pressure lowering effect (see fig. 1B and fig. 6B). This raises the question of whether the effects of PHM observed in SHRSP may be more of a developmental nature. The antihypertensive effects of PHM should therefore also be shown in a phase of stable hypertension. If this is not possible in SHRSP, SHR could also be investigated.
3. There is a major discrepancy in the time course of in vivo responses to PHM (between day 8 and day 15) and in vitro responses that showed a rapid and strong inhibitory effect of pulsatile fluid shear stress on the expression of AT1R in cultured primary astrocytes, which was even more pronounced after 24 hours (Fig. 5). Even assuming that cumulative effects are necessary for the in vivo effect of increased fluid shear stress, a much earlier response should be expected if the in vitro model is valid. One way to test this could be to perform the experiment shown in fig. 2 after only two days of treatment with PHM. Please also provide time control data for 6 h and 24 h for the cell culture experiments.
4. Do SHRSP show unaltered responses to Ang II after the introduction of the hydrogel?
5. The treatment with VOCR had a remarkably strong hypotensive effect (in one person the mean blood pressure dropped by almost 50 mmHg!). According to the authors' hypothesis, this reduction in blood pressure should be triggered mechanistically by a damping of sympathetic activity, however VOCR had practically no impact on all the indicators of sympathetic tone studied. To me, these results therefore seem to refute rather than support the authors' hypothesis. The authors should provide direct evidence of an inhibitory influence on sympathetic tone, e.g. by nerve conduction or measurements of stable metabolites of the sympathetic neurotransmitters in urine. In addition, time control without VOCR is indispensable to allow a meaningful interpretation of the specific effects of VOCR.
6. AT1R antagonists are one of the most widely used drugs in the treatment of arterial hypertension. Since they can permeate the blood-brain barrier (PMID: 10882779), one would expect that exercise in hypertensive patients treated with sartans (or ACE inhibitors) should yield much weaker antihypertensive effects. To my knowledge, this has not yet been described. Four of the 21 patients in this study were also treated with a sartan. Did these patients show a lower blood pressure response to VOCR than the other subjects?
7. Other forms of exercise such as swimming, cycling or isometric exercise, which should have considerably less mechanical effects, produce equally large decreases in blood pressure as running. How do you explain this? Why, then, does running (because of the additional mechanically mediated effect) not lead to a significantly greater lowering of blood pressure than does isometric exercise, for example?
8. In the discussion, the authors state that the magnitude of shear forces caused by interstitial fluid movement in the RVLM is as large as the shear forces acting on the vascular endothelium. In view of the strength of the blood flow in the circulation, this does not seem plausible to me.

Reviewer #3 (Report for the authors (Required)):

OVERALL SUMMARY

In this study, the authors investigate the role of head movement as a byproduct of exercise in reducing essential hypertension. In order to isolate exercise-induced head movement (which they term passive head movement, or PHM) from the other effects of exercise, they implement a technique which they have previously described in order to oscillate the heads of rats in a programmed fashion. In significant advances, they show that PHM reduces blood pressure in a rat model of hypertension whereas it has no effect on a WT control. They propose that this reduction in blood pressure is driven by PHM-induced fluid shear stress which decreases the expression of AT1R on astrocytes in the rostral ventrolateral medulla and affects blood pressure via the sympathetic nervous system. They present comprehensive evidence to support this claim. Finally, they investigate the effect of a similar approach, an oscillating chair, to reduce blood pressure in hypertensive humans. They find that it is effective and may act through a similar mechanism, providing a potential translational advancement. Overall, this study is interesting, comprehensive, and has significant ramifications relevant to human health, but key technical details, controls, and analyses are missing.

MAJOR COMMENTS

1. The central premise of the proposed mechanism is that PHM causes changes in hydrostatic pressure which are sufficient to induce fluid shear stress (FSS) on the astrocytes within the RVLM. The authors present data which qualitatively indicates that PHM causes the induction of flow within the interstitial fluid (telemetry data from an implanted sensor, uCT of contrast agent). However, the effects of FSS on cells is dependent on the quantitative nature of the FSS (i.e., the magnitude of the shear matters as well as the dynamics of the flow). To validate their proposed mechanism, the authors performed an in vitro experiment wherein astrocytes and neurons were subjected to pulsatile fluid shear. To justify the parameters used in their in vitro experiments, the authors perform a simple calculation (which they confusingly refer to as a simulative calculation, despite no simulations conducted) based on data collected via uCT scans. These calculations are extremely simple and are very unlikely to be accurate. Critically, it is not clear whether this analysis is appropriate for oscillatory flow, as it does not seem possible to measure the dynamics of the flow from their uCT experiments. Additionally, the details of what was done are not presented clearly. For instance, it is unclear how x,y,z coordinates in the contrast agent data relate to a cell. In addition to addressing the above points, a diagram showing the anatomy of the interstitial space/fluid with astrocytes and neurons, how fluid flow would occur, and the orientation of the pertinent axes would be helpful.
2. Given the data presented in the paper, there is no reason to assume that this simple model, even when explained in greater detail, accurately estimates the FSS experienced by astrocytes. This is highlighted by the fact that the high end for FSS values they estimate are on a similar order of magnitude to that experienced by endothelial cells within the vasculature (Resnick et al., 2003, *Prog. Biophys Mol. Biol.*). Additionally, if other velocity components were used these estimates would increase by a factor of 20-25. Furthermore, these estimates inform their choice of FSS magnitude for use in cell culture. The authors should investigate other, more accurate methods of modeling/simulating interstitial fluid flow in the brain in response to movement and provide extensive justification for the estimate of FSS experienced by individual cells. A nice review of modeling within the brain interstitial system can be found in Lei et al., 2017, *Prog. Neurobiol.* Additionally, see work of Jennifer Munson, Roger Kamm, and Melody Swartz.
3. The use of uCT tracking of contrast agent may not capture the native fluid motion induced via PHM. Simply injecting a small volume of solution into the RVLM is likely to cause disruption of the tissue architecture surrounding the injection site. It is therefore likely that the effect PHM has on an injected bolus of fluid, in a region of the brain which may have been mechanically compromised, is different than how PHM affects the movement of interstitial fluid. For an example of a thorough analysis of injections within the CNS, see Guest et al., 2011, *Brain Res. Bulletin*. These issues should be addressed in the limitations of the study.
4. The use of a sensor to report the low-amplitude pressure waves within the interstitial fluid is not sufficiently described. How does the sensor operate, how were the results interpreted, and how does this relate to the movement of fluid? The brain is not a simple hollow organ, so measuring changes in pressure at one point may not clearly capture what is happening. Perhaps a supplemental note describing the use and interpretation of the sensor would be helpful.
5. The use of a PEG gel to restrict interstitial fluid flow is confusing and many of the potential off-target effects are not discussed. For example, PEG has been shown to improve outcomes following spinal cord injury via reduction of reactive oxygen species (Luo, Borgens, Shi 2002 *J. Neurochem*/Luo, Borgens, Shi 2004 *J. Neurotrauma*). Additionally, injecting a gel into the RVLM is likely to cause a disruption to the local architecture and disruption of cells. The authors themselves admit that injection of PEG gel is filled with challenges and “may not entirely prove the contribution of interstitial fluid movement”. The rationale for this experiment, and its interpretation, is not clear. Can an improved version of μ CT tracking of contrast agent and subsequent analysis be used to demonstrate that the interstitial flow is changed in response to gel formation?
6. The statistical tests used in this paper to support their conclusions are occasionally incorrect. Specifically, for the measurements of a variable over time within the same group (e.g., how does BP change over time with PHM), the authors should use a repeated measures ANOVA to account for the fact that the measurements are being repeatedly taken from the same animals. The statistical tests should be re-done and any resulting conclusions changed to highlight this.

MINOR COMMENTS

1. The authors should refrain from using the phrase “mechanical impact” unless referring to traumatic brain injury. In the case of this paper, “mechanical impact” grossly miscommunicates the current work. Perhaps

something like “passive motion” would be more appropriate. This applies to both the title and body of the paper.

2. It is possible that PHM has an additional effect via stimulating the muscles of the neck. How is PHM isolated from potential exercise of neck muscles?

3. Why did the authors use only male rats in this study? Sex is an important variable for nearly all diseases, including hypertension. The limitations of studying only male rats should be discussed. The human subjects were of mixed gender. Were there gender-specific effects in this data?

OTHER COMMENTS

1. The authors continually referred to use of PEG to “gel interstitial fluid”. This is an improper term. To gel means to move from the solution phase to the gel phase. The fluid is not going into a gel phase. The polymerization of the gel creates a region with much smaller pore size, which we reduce the flow in the system for a given pressure drop.

2. In line 161, the authors describe the results as approximate mirror images. This is imprecise, alternative wording should be used to more clearly communicate what the authors mean.

3. The authors describe the effect of fluid shear on cultured astrocytes in vitro as being cumulative. However, that does not clearly connect to the observed response of BP over time. According to figure 1B, the MAP of rats experiencing PHM continues to rise during the course of the study, just at a lower rate than the PHM-rats. This should be clarified.

4. On line 345, the authors mention that mechanical loading and/or FSS alleviates inflammation, which can affect AT1R expression. However, it is unclear how to connect inflammation within bone tissue (cited as one example) to inflammation within the CNS, which is immune-privileged. Additionally, reversible, oscillatory FSS is associated with increase inflammation in the vasculature. These comments need refined for accuracy.

5. Line 366/377 (“As the phrase...”) is too conversational and speculative and not of an appropriate tone for a journal article.

6. On line 389/390, what do the authors mean by mechanical factors being a target within the blood-brain barrier? It is unclear how a mechanical factor could be a specific target.

7. In Fig. 1g, 1h, 6e and Extended Data Fig. 1b, it is not clear that the GFAP-AGTRAP targeted astrocytes and not neurons. Given the close proximity between astrocytes and neurons, looking at neuronal nuclei and suggesting that the GFP staining is mutually exclusive is not clear from this picture. Likewise, in Extended Data 1D it is not clearly indicated that the GFP and GFAP labeling comes from different cells. Greater details regarding the determination of AT1R-positive neurons and AT1R-positive astrocytes should be provided? Was the overlap assessment done manually or was it automated? It is also not clear how the RVLM was histologically sectioned and whether all the cell counts came from roughly the same relative section for each rat – is AT1R expression uniform throughout the RVLM?

8. In line 677, the nomenclature for positive cells should be changed. The use of the “-“ symbol is confusing because this is commonly used to refer to negative or absent expression. Perhaps a different symbol, such as “/” could be used to compound proteins together?

Wed 18 May 2022

Decision on Article nBME-20-1499A

Dear Prof Sawada,

Thank you for your revised manuscript, "Antihypertensive effect of brain-targeted mechanical intervention with passive head motion", which has been seen by the original reviewers. In their reports, which you will find at the end of this message, you will see that the reviewers acknowledge the improvements to the work and raise a few additional technical criticisms that we hope you will be able to address. In particular, we would expect that the next version of the manuscript provides

*Clarification of whether the antihypertensive effect is mediated by reduced AT1R expression in astrocytes in the RVLM (with compelling arguments and/or additional evidence), or discussion of the limitations.

*Discussion on why the changes are not observed in stable hypertension and how A1TR be disconnected from change in blood pressure.

*Discussion of the limitations of the data that you can gather to support your claims on AT1R downregulation, in particular regarding estimation of changes in fluid shear stress.

As before, when you are ready to resubmit your manuscript, please upload the revised files, a point-by-point rebuttal to the comments from all reviewers, the reporting summary, and a cover letter that explains the main improvements included in the revision and responds to any points highlighted in this decision.

As a reminder, please follow the following recommendations:

* Clearly highlight any amendments to the text and figures to help the reviewers and editors find and understand the changes (yet keep in mind that excessive marking can hinder readability).

* If you and your co-authors disagree with a criticism, provide the arguments to the reviewer (optionally, indicate the relevant points in the cover letter).

* If a criticism or suggestion is not addressed, please indicate so in the rebuttal to the reviewer comments and explain the reason(s).

* Consider including responses to any criticisms raised by more than one reviewer at the beginning of the rebuttal, in a section addressed to all reviewers.

* The rebuttal should include the reviewer comments in point-by-point format (please note that we provide all reviewers will the reports as they appear at the end of this message).

* Provide the rebuttal to the reviewer comments and the cover letter as separate files.

We hope that you will be able to resubmit the manuscript within 12 weeks from the receipt of this message. If this is the case, you will be protected against potential scooping. Otherwise, we will be happy to consider a revised manuscript as long as the significance of the work is not compromised by work published elsewhere or accepted for publication at *Nature Biomedical Engineering*.

We look forward to receive a further revised version of the work. Please do not hesitate to contact me should you have any questions.

Best wishes,

Michelle

Dr Michelle Korda
Senior Editor, Nature Biomedical Engineering

Reviewer #1 (Report for the authors (Required)):

Overall, the paper remains compelling and the authors are to be lauded for attempting to address almost all of the concerns raised by the reviewers and editors. Accordingly, the authors have conducted a significant number of experiments at the in vitro and in vivo levels in both rodents and humans. While their data does not definitively prove their hypothesis beyond a reasonable doubt, they do certainly and reasonably point in that direction. Indeed, the experiments that are required to obtain truly definitive proof might not even be practically feasible, at least from this reviewer's perspective. For example, it is unclear what the shear stresses and fluid dynamics of the relevant microenvironment truly are but it is even more unclear how one could even truly obtain those measurements experimentally yet accurately and the authors have done their best to estimate those values. In addition, the authors have aptly expanded the discussion of the limitations of their methods and data, which increases the scientific rigor and transparency. Nevertheless, the overall concept of the revised manuscript will still have significant implications for the study and treatment of hypertension in the context of cardiovascular disease. The enthusiasm for this manuscript is dampened somewhat by the lack of placement of this work in the context of other researchers who have conducted similar experiments. While this paper is novel, other papers in this space should be more extensively cited and compared/contrasted to better orient the reader.

Reviewer #2 (Report for the authors (Required)):

This is a very thorough revision of the original manuscript. The authors present important additional experimental data which substantially strengthens their conclusion that mechanically mediated effects on the brain contribute significantly to the antihypertensive effect of physical exercise. They have also completely revised the calculation of the mechanical forces involved and now arrive at much more plausible values. Whether the antihypertensive effect is mediated by an attenuated expression and pharmacological responsiveness of the CNS AT1R signaling system, however, is in my view not yet sufficiently experimentally documented. Essentially, the argument entirely relies on the estimation of changes in fluid shear stress elicited by PHM in vivo, which is still arbitrary. The demonstration that in the presence of a blockade of the AT1R signaling system in the RVLM region (either pharmacologically or genetically), the antihypertensive effects of PHM are absent or at least greatly attenuated would make a causal role of this mechanism much more likely. The authors should at least explicitly address this limitation in the discussion.

Reviewer #3 (Report for the authors (Required)):

After reviewing the extensive rebuttal letter and highly revised manuscript, I have only a single major concern remaining. In Response (4) [Rebuttal pg 13-14], the authors respond to a point that Reviewer 2 made which also touched on a point I made (pts 18, 19, and 36), which is that the time evolution of the in vitro astrocytic response does not conceptually align with the time evolution of the data on PHM intervention in rats. They performed an experiment requested by Reviewer 2, and reported that "Four-week PHM did not significantly alter the blood pressure of SHRSPs when initiated in a phase of stable hypertension. However, PHM still decreased the expression of A1TR in the RVLM." First, this data does not appear to be in the manuscript. It should be added. Also, this result appears inconsistent with the authors' main point and consistent with Reviewer 2's alternative hypothesis. The authors should provide a full discussion regarding how Reviewer 2's alternative hypothesis can be excluded from potential interpretations. Key questions to be addressed are 1) why are changes not observed in stable hypertension? And 2) How can A1TR be disconnected from change in blood pressure if that is the core of their proposed mechanism? This is clearly an important result as it has implications both for the clinical relevance of their PHM intervention and the interpretation of their proposed mechanism. Thus, the authors should address this in more detail and clearly demonstrate its inclusion in the manuscript prior to publication.

Tue 19 Jul 2022

Decision on Article nBME-20-1499B

Dear Prof Sawada,

Thank you for your revised manuscript, "Antihypertensive effect of brain-targeted mechanical intervention with passive head motion". Having consulted with the original reviewers (whose comments you will find at the end of this message), I am pleased to write that we shall be happy to publish the manuscript in *Nature Biomedical Engineering*.

We are now performing detailed checks on your paper and will send you a checklist detailing our editorial and formatting requirements in due course. Please do not upload the final files until you receive this additional information from us.

Best wishes,

Michelle

Dr Michelle Korda
Senior Editor, Nature Biomedical Engineering

Reviewer #1 (Report for the authors (Required)):

The authors have adequately addressed the concerns of this reviewer and I believe the newest version of this manuscript is suitable for publication.

Reviewer #2 (Report for the authors (Required)):

The authors have added three paragraphs in the discussion section in which they address the concerns raised regarding the limitations of their study. In particular, they now convincingly explain why their transgenic model of astrocyte-specific AGTRAP overexpression cannot be used to experimentally test their mechanistic hypothesis for the observed antihypertensive effect of PHM.

I just do not understand the sentence "Whereas transient yet sustained AT1R activation can cause irreversible impairments of renal function and hypertension, severe kidney damages are observed in 20-week-old SHRSPs" (p.31, 582-583). Perhaps the authors can rephrase it somewhat more clearly.

Reviewer #3 (Report for the authors (Required)):

The authors have addressed my concerns.

Rebuttal 1

In particular, we would expect that a revised version of the manuscript provides:

Point 1

***Clarification of the causative mechanisms for the effects of the passive head movements with further experimental evidence, as suggested by all the Reviewers.**

Response 1

We have extensively conducted additional passive head motion (PHM) experiments with considerable variations, including different directions (Extended Data Fig. 2), frequencies, and peak magnitudes (Extended Data Fig. 3). Furthermore, we have analyzed the blood pressure and heart rate in the SHRSPs during the transition from before to after the initiation of PHM and found that both remained unaltered (Extended Data Fig. 14). These findings preclude the involvement of baroreceptor response, either carotid or aortic, in the effects of PHM. Consistently, the aortic depressor nerve activity remained unchanged from before to after the PHM initiation (Extended Data Fig. 14). We have described these new results in our revised manuscript to strengthen our claim regarding the mechanism underlying the antihypertensive effects of PHM.

Point 2

***Clarification of the calculation of shear forces during running, and further justification that fluid shear stress influences blood pressure with experimental evidence, as suggested by all the Reviewers.**

Response 2

To address the concerns raised by Reviewers #1 (Point 11), #2 (Point 24), and #3 (Points 25 and 26) regarding the fluid shear stress calculation, we have conducted additional experiments and analyzed the structure/orientation and dimension (cross-sectional area) of the interstitial space by combining fluorescent hydrogel introduction and multiphoton microscopic imaging. The interstitial space of the rat RVLM appeared moderately oriented approximately along the centroidal line of this part of the brain with minor lateral communications (Extended Data Fig. 6). Furthermore, the cross-sectional area of the RVLM interstitial space was estimated as 0.0083–0.18 μm^2 (Extended Data Fig. 7). Based on these findings together with the previous reports on the interstitial space/fluid /fluid flow in the brain, we have re-calculated the magnitude of fluid shear stress exerted on cells in the rat RVLM during PHM. The range of the fluid shear stress magnitude resulting from our revised calculation (0.076–0.53 Pa) was considerably smaller than that reportedly exerted on the arterial endothelium (1.5–2 Pa)¹.

As discussed later (Responses 11, 24, 25, 26, and 28), we acknowledge the limitations and approximate nature of our estimation of the fluid shear stress magnitude. Taking this into account, we have extensively conducted additional *in vitro* experiments using cultured astrocytes, in which we tested fluid shear stress of different magnitudes (smaller magnitudes; Fig. 5a), as well as hydrostatic pressure changes of various amplitudes (Fig. 5b).

We have described these results in the revised manuscript. Although it is impossible to completely recapitulate the *in vivo* conditions using 2-D cell culture *in vitro*, we believe our new

experimental results justify and support our claim that fluid shear stress on RVLM astrocytes mediates the antihypertensive effect of PHM.

Point 3

***Further time control data to align the in vitro and in vivo experiments as suggested by Rev #2.**

Response 3

Following the suggestion from Reviewer #2 (see Point 19), we conducted 2-day PHM experiments in which we analyzed the pressor and depressor responses by the local injection of angiotensin II and valsartan, respectively. Consistent with the time-course of the decrease in angiotensin II type 1 receptor (AT1R) expression after the application of fluid shear stress to cultured astrocytes, attenuation of the pressor and depressor responses was observed after 2-day PHM application (Extended Data Fig. 9).

We have extensively tested fluid shear stress and hydrostatic pressure changes of various magnitudes/amplitudes with appropriate time controls in the in vitro experiments (Fig. 5a–f and Extended Data Fig. 8b,c). Please see Response 11.

Point 4

***Clarification of the data pertaining to sympathetic tone, with experimental evidence as suggested by Reviewer #2.**

Response 4

To examine whether vertically oscillating chair riding (VOCR) affects the sympathetic nerve activity of hypertensive human subjects, we conducted continuous beat-by-beat blood pressure and inter-beat (R–R) interval (RRI) recording, and analyzed the systolic blood pressure and RRI variability. The low-frequency (LF) power of the systolic blood pressure variability and the ratio of LF/high frequency (HF) power (LF/HF ratio) in the RRI variability have been reported to relate to the vascular sympathetic nerve activity^{2,3} and the balance of cardiac sympathetic/parasympathetic nerve activity^{2,4}, respectively. VOCR, but not non-oscillating chair riding (NOCR), significantly decreased the LF power in the systolic blood pressure variability and elicited a decreasing tendency ($P = 0.063$) in the LF/HF ratio of the RRI variability (Fig. 8b,c of our revised manuscript). These results indicate a sympathoinhibitory effect of VOCR in hypertensive adult humans. The nerve conduction analysis requires close contact between the tester and testee (i.e., participant) in a shield room where adequate ventilation is not available. The 24-h urine analysis requires hospitalization of participants in Japan. Thus, the COVID-19 pandemic had some influence on our choice of analysis to examine the sympathoinhibitory effect of VOCR, although it was not the only reason for not choosing nerve conduction or 24-h urine analysis. Please also refer to Response 21.

Point 5

***Clarification of the statistics as suggested by Rev #3.**

Response 5

Following the comment from Reviewer #3, we have re-performed the statistical analyses for some of the data, as appropriate. Although the P values have changed after the re-analyses, the conclusions of those data were not altered. Please also see Response 30.

Please follow the following recommendations:

Point 6

*** Clearly highlight any amendments to the text and figures to help the reviewers and editors find and understand the changes (yet keep in mind that excessive marking can hinder readability).**

Response 6

We have clearly highlighted the amendments we have made in red characters. To avoid excessive marking, we have formulated our use of red characters as follows.

(1) Body text

Most of the amendments are highlighted in red. For the paragraphs in the Methods section where we have described the procedures for newly added experiments or analyses or added a considerably large amount of information regarding the detailed experimental protocols (e.g., cell culture), only the subheadings (i.e., the first sentences of the paragraphs) are highlighted in red.

(2) Figures, Extended Data Figures, and Supplementary Tables

- The entire figure or table is new or revised; only the labels at the top left corner (e.g., “Fig. 8” and “Extended Data Fig. 3”) are highlighted in red characters.
- The figure is partly revised (or added); the symbols (e.g., “a” and “b”) in the figures/tables containing amendments are highlighted.

(3) Legends to figures and tables

- The entire figure or table is new or revised; only the number and title of the figures/tables are highlighted.
- The figure is partly revised (or added); the symbols (e.g., “a” and “b”) for the figures/tables containing amendments are highlighted.

(4) There are additional highlights to address the reviewers’ comments (e.g., “repeated measures ANOVA”)

*** If you and your co-authors disagree with a criticism, provide the arguments to the reviewer (optionally, indicate the relevant points in the cover letter).**

*** If a criticism or suggestion is not addressed, please indicate so in the rebuttal to the reviewer comments and explain the reason(s).**

Point 7

*** Consider including responses to any criticisms raised by more than one reviewer at the beginning of the rebuttal, in a section addressed to all reviewers.**

Response 7

As suggested, we have described our responses to the criticisms and questions by more than one reviewer, gathered and summarized as (1) – (4) below, in the first section of our rebuttal to the individual reviewers' comments (i.e., just before the beginning of Reviewer #1's comments).

*** The rebuttal should include the reviewer comments in point-by-point format (please note that we provide all reviewers will the reports as they appear at the end of this message).**

*** Provide the rebuttal to the reviewer comments and the cover letter as separate files.**

COMMENTS AND CONCERNS RAISED BY MORE THAN ONE REVIEWER, AND OUR RESPONSES TO THEM

(1) Calculation of the magnitude of fluid shear stress exerted on rat RVLM astrocytes during PHM, and justification of the in vitro experiments of mechanical interventions using cultured cells (Points 11, 24, 25, and 26 from Reviewers #1, #2, and #3)

Point 11 (from Reviewer #1)

First, there is little evidence that astrocytes experience the changes in interstitial pressure as postulated in Figure 4. The contrast agent data demonstrates effects that occurs at a much higher size scale than at the cellular level. In fact, it is unclear that astrocytes within the complex anatomy of the medulla even are exposed to the contrast agent let alone the hydrodynamic forces thereof. In addition, the in vitro fluid stress experiments occur in a system that again is non-physiologic and do not recapitulate the in vivo conditions of the astrocytes' microenvironment.

Point 24 (from Reviewer #2)

In the discussion, the authors state that the magnitude of shear forces caused by interstitial fluid movement in the RVLM is as large as the shear forces acting on the vascular endothelium. In view of the strength of the blood flow in the circulation, this does not seem plausible to me.

Point 25 (from Reviewer #3)

To justify the parameters used in their in vitro experiments, the authors perform a simple calculation (which they confusingly refer to as a simulative calculation, despite no simulations conducted) based on data collected via uCT scans. These calculations are extremely simple and are very unlikely to be accurate. Critically, it is not clear whether this analysis is appropriate for oscillatory flow, as it does not seem possible to measure the dynamics of the flow from their uCT experiments. Additionally, the details of what was done are not presented clearly. For instance, it is unclear how x,y,z coordinates in the contrast agent data relate to a cell. In addition to addressing the above points, a diagram showing the anatomy of the interstitial space/fluid with astrocytes and neurons, how fluid flow would occur, and the orientation of the pertinent axes would be helpful.

Point 26 (from Reviewer #3)

The authors should investigate other, more accurate methods of modeling/simulating interstitial fluid flow in the brain in response to movement and provide extensive justification for the estimate of FSS experienced by individual cells. A nice review of modeling within the brain interstitial system can be found in Lei et al., 2017, Prog. Neurobiol. Additionally, see work of Jennifer Munson, Roger Kamm, and Melody Swartz.

Response (1)

We realize that the intramedullary pressure changes we described (~1.2 mm Hg) may not represent the magnitude of the pressure changes that the rat RVLM astrocytes experience during PHM.

Therefore, in our revised manuscript, the ~1.2-mm-Hg pressure waves are interpreted only as an indication of stress distribution changes or microdeformation in the rat medulla during PHM (page 12, line 220–222), and we have not referenced 1.19 mm Hg for our calculation of the fluid shear stress magnitude (Supplementary Table 1).

We agree that it is not reasonable to estimate the velocity of the interstitial fluid movement (flow) in the rat RVLM based on the quantification of contrast agent (Isovist) spreading detected by μ CT analysis. In revising the manuscript, we have not calculated the velocity of the interstitial fluid movement (flow) based on the μ CT data. However, the direction of interstitial fluid movement demonstrated in our μ CT experiments (Fig. 4g,h) appears to be consistent with the orientation of the interstitial space in the rat RVLM, as observed in our analysis using a multiphoton microscope (Extended Data Fig. 6a–c,e). Therefore, we have referred to the finding from our μ CT experiments just to approximately estimate the PHM-induced enhancement of the interstitial fluid movement in the rat RVLM as two- to three-fold.

We have analyzed the structure/orientation/dimension of the interstitial space using a multiphoton microscope through the process of revising the calculation of the fluid shear stress in the rat RVLM during PHM. We have accordingly drawn diagrams showing the anatomy of the interstitial space with astrocytes and neurons (Extended Data Fig. 6e and Extended Data Fig. 7d). We

also present a diagram illustrating the microdeformation that drives interstitial fluid flow in the rat RVLM (Extended Data Fig. 7e).

We do not have sufficient data available for accurate modeling/simulation to calculate the magnitude of PHM-induced fluid shear stress (see Response 26). Nonetheless, we believe that our revised estimation (Supplementary Table 1) is based on more reasonable input of parameters, which includes 0.2 $\mu\text{m/s}$ as the interstitial fluid flow velocity in the brain of sedentary rodents referenced from the previous publications^{5,6} suggested by Reviewer #3 (Point 26).

Our original calculation of the magnitude of the fluid shear stress exerted on the nervous cells in the rat RVLM (0.59–2.64 Pa) was partially based on our inappropriate citation of previous publications on the viscosity of interstitial fluid in the brain (1–20 mPa·s cited from references #57–#59 in our original manuscript). Although we are still unable to refer to a previous report definitely describing the value of the intracerebral interstitial fluid viscosity, it has been reported that the composition of interstitial fluid in the brain is similar to that of cerebrospinal fluid (CSF)⁷. Therefore, we have referenced the value reported as human cerebrospinal fluid viscosity (0.72 mPa·s)^{8,9} for the fluid shear stress calculation in our revised manuscript. Our revised calculation gave 0.076–0.53 Pa as the magnitude of fluid shear stress on the nervous cells in the rat RVLM during PHM. This value is considerably smaller than the magnitude of the shear force on the arterial endothelium reported previously¹.

We are aware of the limitations and approximate nature of our calculation of fluid shear stress magnitude. We also recognize the “non-physiologic” nature of the in vitro fluid shear stress

experiments using cultured cells, as pointed out by Reviewer #1 (Point 11). Taking these into account, we have extensively tested the mechanical interventions (application of fluid shear stress and hydrostatic pressure change) with various different magnitudes/amplitudes to cultured astrocytes (Fig. 5 and Extended Data Fig. 8b).

(2) Placement of our work in the context of others already published pertaining to the antihypertensive effects of exercise (Points 13, 17, and 23 from Reviewers #1, #2, and #3)

Point 13 (from Reviewer #1)

Importantly, the authors should better place their work in the context of others already published in the literature. In fact, if done correctly, this can better bolster their argument.

Point 17 (from Reviewer #2)

The relevance of direct mechanical effects on the RVLM may be better tested by investigating whether pre-treatment by PHM or prevention of interstitial fluid movements can significantly reduce the antihypertensive effects of running in SHRSP.

Point 23 (from Reviewer #3)

Other forms of exercise such as swimming, cycling or isometric exercise, which should have considerably less mechanical effects, produce equally large decreases in blood pressure as running. How do you explain this? Why, then, does running (because of the additional mechanically mediated effect) not lead to a significantly greater lowering of blood pressure than does isometric exercise, for example?

Response (2)

We realize that the novelty and significance of our study can be better argued by placing it in the context of exercise effects. We have accordingly revised the Abstract, Introduction, and Discussion sections, referring to physical exercise at the beginning of these sections. We have additionally cited others' publications on exercise effects. Furthermore, in the revised manuscript, we show that

treadmill running, which decreased the blood pressure in SHRSPs, was not significantly antihypertensive in the hydrogel-introduced SHRSPs (Extended Data Fig. 12), suggesting that the interstitial fluid movements are involved in the antihypertensive effects of treadmill running at a moderate velocity.

Given our new results from the PHM with different magnitudes and frequencies (Extended Data Fig. 3c,e), it is reasonable to refer to the possibility that the rostral-caudal head motion is involved in the antihypertensive effects of other forms of exercise, such as swimming and bicycle riding (page 25, line 471–474 of our revised manuscript). Although we have described the inability to explain the antihypertensive outcome of isometric exercise by direct mechanical effects on the brain as one of the limitations of our study (page 29, line 546–547), we believe that we have better placed our work in light of previous publications on the antihypertensive effects of exercise.

(3) “Mechanical Impact” as inappropriate wording (Points 14 and 31 from Reviewers #1 and #3)

Point 14 (from Reviewer #1)

The phrase “Mechanical Impact” in the title implies blunt head trauma, which is not the case and therefore should be removed or rephrased.

Point 31 (from Reviewer #3)

The authors should refrain from using the phrase “mechanical impact” unless referring to traumatic brain injury. In the case of this paper, “mechanical impact” grossly miscommunicates the current work. Perhaps something like “passive motion” would be more appropriate. This applies to both the title and body of the paper.

Response (3)

We have revised the title of the paper as follows.

Antihypertensive effect of brain-targeted mechanical intervention with passive head motion

We have also rephrased most of the phrase “mechanical impact” with alternative wording in our manuscript.

(4) Mechanisms that link angiotensin II receptor type 1 (AT1R) expression in RVLM astrocytes to blood pressure control (Points 18, 19, and 36 from Reviewers #2 and #3)

Point 18 (from Reviewer #2)

PHM appears to prevent a further increase in blood pressure rather than exerting a blood pressure lowering effect (see fig. 1B and fig. 6B). This raises the question of whether the effects of PHM observed in SHRSP may be more of a developmental nature. The antihypertensive effects of PHM should therefore also be shown in a phase of stable hypertension.

Point 19 (from Reviewer #2)

Even assuming that cumulative effects are necessary for the in vivo effect of increased fluid shear stress, a much earlier response should be expected if the in vitro model is valid. One way to test this could be to perform the experiment shown in fig. 2 after only two days of treatment with PHM.

Point 36 (from Reviewer #3)

The authors describe the effect of fluid shear on cultured astrocytes in vitro as being cumulative. However, that does not clearly connect to the observed response of BP over time. According to figure 1B, the MAP of rats experiencing PHM continues to rise during the course of the study, just at a lower rate than the PHM- rats. This should be clarified.

Response (4)

Four-week PHM did not significantly alter the blood pressure of SHRSPs when initiated in a phase of stable hypertension (21 weeks of age). However, PHM still decreased the expression of AT1R in the RVLM. Two-day PHM significantly reduced both pressor and depressor responses in SHRSPs (Extended Data Fig. 9), supporting the validity of the results from our in vitro fluid shear stress

experiments. We interpret >2 weeks (Fig. 1b) as the time required for attenuated AT1R expression in RVLM astrocytes to elicit its blood pressure-lowering outcome rather than the cumulative decrease in AT1R expression. We speculate that there might be a complex mechanism that links the AT1R expression in RVLM astrocytes with the blood pressure control, which can be irreversibly altered by some factor(s), such as vascular and renal (dys)function or severity of hypertension¹⁰⁻¹³. We have discussed this issue in our revised manuscript (page 15–16, line 285–290).

REBUTTAL TO INDIVIDUAL REVIEWERS' COMMENTS AND QUESTIONS.

Reviewer #1 (Report for the authors (Required)):

In this interesting paper, the authors claim that passive head motion of hypertensive rats generated at their heads during treadmill running at a moderate velocity decreases the expression of angiotensin II type 1 receptor (AT1R) in astrocytes in their rostral ventrolateral medulla, which in turn decreases blood pressure. Overall, the paper is compelling and the authors are to be commended for executing a large number of complementary, multidisciplinary experiments leading to significant amounts of data. If the authors' hypothesis is correct, this work will have profound implications for the study and treatment of hypertension in the context of cardiovascular disease. However, because the central ideal is indeed exciting, the burden of proof is high and as such, several parts of the paper have significant flaws that need to be addressed.

Point 8

1) First, while the overall scientific premise is interesting, could all of this be predominantly due to a

classic baroreceptor response? Indeed, when the head is being moved up and down repeatedly, carotid and aortic stretching can occur, which will lead to decrease in blood pressure. While the AT1R seems to be involved, could that signaling pathway be dwarfed by the parasympathetic baroreceptor response?

Response 8

Both the blood pressure and heart rate of hypertensive rats (SHRSPs) remained unchanged during the transition from before to after the initiation of PHM, precluding the baroreceptor response, either carotid or aortic. Consistently, the activity of the aortic depressor nerve, which transmits afferent signals from the baroreceptors and chemoreceptors located in the aortic arch, also remained unaltered during the transition from before to after PHM initiation. These results indicate that the baroreceptor response is not involved in the effects of PHM. We have described these new findings in Extended Data Fig. 14 of the revised manuscript. Please also see Response 1.

Point 9

2) The PHM needs to be better characterized in this paper. What happens when PHM directionality, frequency, and amplitude all are modulated systemically? How do these variables each affect the duration of the decrease in BP? Are there plateau effects with each variable? Rigorous characterization of these parameters could help make the authors' cases.

Response 9

To address the points raised by the reviewer, we have extensively tested various PHMs as follows.

1) Direction of PHM.

We have tested the PHM (2 Hz; 30 min daily; 28 days) that generated accelerations with a peak magnitude of $1.0 \times g$ in different directions. The forward-backward (i.e., rostral-caudal) PHM significantly decreased the blood pressure of SHRSPs, which was not significantly altered by the left-right PHM (Extended Data Fig. 2). This indicates the directional specificity of PHM in terms of its antihypertensive effect.

2) Frequency of PHM

We tested the PHM (peak acceleration magnitude of $1.0 \times g$; 30 min daily; 28 days) at different frequencies. The 0.5-Hz PHM significantly decreased the blood pressure of SHRSPs, whereas 0.2-Hz one did not. The antihypertensive effects were not significantly different between PHMs at 2 Hz and 0.5 Hz (Extended Data Fig. 3c). These results suggest that there are a threshold and a plateau phase of the frequency of PHM in light of its antihypertensive effect.

3) Magnitude (amplitude) of PHM.

The PHM (2 Hz, 30 min daily, 28 days) that generated vertical accelerations with a peak magnitude of $0.5 \times g$ was antihypertensive. However, the PHM with a peak magnitude of $0.2 \times g$ did not significantly alter the blood pressure of SHRSPs. The antihypertensive effects of PHM were not significantly different between peak magnitudes of $0.5 \times g$ and $1.0 \times g$ (Extended Data Fig. 3e). These results suggest that there are a threshold and a plateau phase of the magnitude of PHM regarding its antihypertensive effect.

Point 10

3) The human experiments comprising vertically oscillating chair riding utilize a different mode of force application, which could confound results. This needs to be addressed and discussed.

Response 10

The peak magnitude of vertical accelerations in the heads during human VOCR was $\sim 1 \times g$ (Extended Data Fig. 15a), which is approximately equivalent to that of 1.0- \times -g rat PHM. However, in the case of human VOCR, body parts other than the head were also subjected to the vertical movements. This was because it seemed difficult to apply vertical forces safely only to the heads of human subjects without causing considerable discomfort or distress to them. In our revised manuscript, we have described the reason for a different mode of force application in the human VOCR (page 18–19, line 345–346), and mentioned the possible influence of the vertical motion of body parts other than the brain (page 29, line 537–541).

Point 11

4) The experiments involving the conjecture that effects of interstitial fluid pressure on astrocytes is a major part of the underlying mechanism unfortunately are more correlative and demonstrate more of an association than causation as many confounders can be taking place. First, there is little evidence that astrocytes experience the changes in interstitial pressure as postulated in Figure 4. The contrast agent data demonstrates effects that occurs at a much higher size scale than at the cellular

level. In fact, it is unclear that astrocytes within the complex anatomy of the medulla even are exposed to the contrast agent let alone the hydrodynamic forces thereof. In addition, the in vitro fluid stress experiments occur in a system that again is non-physiologic and do not recapitulate the in vivo conditions of the astrocytes' microenvironment.

Response 11

With regard to the intramedullary pressure, our measurement (Fig. 4a–d) was at the mm level but not the μm level, because of the size of the sensor that we used (also see Response 28). As the reviewer points out, the value we described (1.19 mm Hg; Fig. 4c,d) may not represent the magnitude of the pressure changes that the RVLM astrocytes experience during PHM. Therefore, we have amended the description as to our interpretation of the intramedullary pressure changes we observed during the rat PHM. In our revised manuscript, the PHM-synchronized 1.19-mm-Hg pressure waves are interpreted only as an indication of the stress distribution changes or microdeformation in the rat medulla (page 12, line 220–222), and we have not referenced 1.19 mm Hg for our calculation of the fluid shear stress magnitude (Supplementary Table 1).

We appreciate and agree with the reviewer's comment on the issue of the size scale regarding our μCT analysis using a contrast agent. In the revised manuscript, we do not refer to the value of the velocity of interstitial fluid movement (flow) based on the quantitation of the μCT data. However, our analysis using a multiphoton microscope, which provides information at the μm level, demonstrates that the interstitial space of the rat RVLM is orientated in the rostral-caudal and dorsal-ventral directions rather than in the left-right direction (Extended Data Fig. 6c), consistent with our

observation in the μ CT experiments (Fig. 4g,h). Based on these findings, we have used our μ CT data to approximate the PHM-induced enhancement of the interstitial fluid flow in the rat RVLM as two- to three-fold (page 13, line 233–237). The issue regarding the range of fluid shear stress described in our original manuscript (0.59–2.64 Pa) is also addressed later (see Response 24).

We agree with the reviewer on the “non-physiologic” nature of our in vitro fluid shear stress experiments. However, 2-D cell culture experiments are non-physiologic particularly in light of the 3-D microenvironments in vivo, and do not entirely recapitulate the physiological conditions. Taking this into account, we have extensively conducted additional in vitro experiments, in which we tested fluid shear stress and hydrostatic pressure changes with different magnitudes and amplitudes (Fig. 5a,b). Our findings support the notion that the fluid shear stress, but not the pressure changes, is responsible for the PHM-induced decrease in the AT1R expression in the RVLM astrocytes. It is beyond the scope of this study to accurately determine the magnitude of interstitial fluid shear stress generated by the PHM. Whereas we referred to the non-physiologic nature of the in vitro experiments using cultured cells in both our original and revised manuscripts (page 23, line 434–437 of the revised manuscript), we have mentioned this issue as one of the limitations of our study (page 27, line 500–501 of the revised manuscript).

Point 12

Finally, the PEG injection experiments may be causing other effects aside from modulating AT1R expression and therefore obvious issues that need to be excluded are local inflammation or increased

intracranial pressure from the gel, which could affect blood pressure. All of these issues need to be definitively addressed and/or clarified.

Response 12

To address the issue of local inflammation, we conducted immunoblot analysis of the RVLM of SHRSPs. We did not observe significant changes in the expressions of pro-inflammatory proteins, TNF- α and IL- β in the hydrogel-introduced RVLM. Furthermore, the intramedullary pressure was not altered by the gel introduction. These findings suggest that the elimination of the antihypertensive effect of PHM by the gel introduction to the RVLM was not mediated by increased local inflammation or intramedullary pressure.

We have added these new results as figures (Extended Data Fig. 13h,i) with related statement in the text (page 17, line 323–325).

Point 13

5) Importantly, the authors should better place their work in the context of others already published in the literature. In fact, if done correctly, this can better bolster their argument.

Response 13

In our original manuscript, results from rat exercise experiments (i.e., treadmill running) were not adequately demonstrated. Therefore, we did not refer much to previous publications on the mechanism of the antihypertensive effects of exercise. However, we agree that the novelty and

significance of our study can be better argued by placing it in the context of exercise effects.

Following the reviewer's comment, we have revised the Abstract, Introduction, and Discussion sections, referring to physical exercise at the beginning of these sections. We have additionally cited others' publications on exercise effects. Furthermore, we have added new experimental results on the effects of exercise (Extended Data Fig. 12 of our revised manuscript). Please also see Response 17.

Minor points

Point 14

1) The phrase “Mechanical Impact” in the title implies blunt head trauma, which is not the case and therefore should be removed or rephrased.

Response 14

Considering this comment together with Reviewer #3's suggestion (see Point 31), we have revised the title of the paper as follows.

Antihypertensive effect of brain-targeted mechanical intervention with passive head motion

We have also rephrased most of the phrase “mechanical impact” with alternative wording in our manuscript.

Point 15

2) Limitations of the animal hypertension model should be mentioned and the model used should be better justified.

Response 15

SHR is the most commonly used animal model of essential hypertension, which does not require surgical procedures or pharmacological treatment, thereby allowing us to obtain a chronic stable hypertensive condition without difficult or life-threatening technical interventions. Inter-individual variation is nearly absent in SHR. Similar to humans, hypertension in SHR develops with age.

SHRSPs elicit higher BP than SHRs. This is advantageous to detect the antihypertensive effects of exercise and PHM, which were supposed to be moderate. However, SHRs or SHRSPs develop hypertension in young adulthood, but not in middle age as in humans. Furthermore, SHRs or SHRSPs cannot model environmental influences that trigger human hypertension, including increased salt intake, obesity, and physical inactivity. In our revised manuscript, we have described the aforementioned advantages to justify the use of SHRSPs as well as the limitations of this model (page 26, line 481–487).

Point 16

3) Is there attenuation of the deleterious effects of the mouse when PHM is applied? Is stroke incidence decreased?

Response 16

We have examined whether PHM attenuated the stroke incidence in SHRSP, and found that ≥ 4 -week PHM significantly decreased or delayed the incidence of stroke (Extended Data Fig. 1d).

Reviewer #2 (Report for the authors (Required)):

This is an ambitious study that proposes a novel mechanism to explain the antihypertensive effects of exercise in arterial hypertension. The authors conclude from a combination of in vivo and in vitro studies in stroke-prone spontaneously hypertensive rats (SHRSP), an animal model for primary hypertension, as well as an intervention study in hypertensive patients, that interstitial fluid movements triggered by physical movements in the rostral ventrolateral medulla (RVLM), a brain area important for the regulation of blood pressure, lower sympathetic tone via an AT-1 receptor (AT1R) dependent mechanism and thereby induce a reduction in blood pressure.

Although this new idea is appealing, I do not find that the data presented sufficiently support the overall hypothesis. I also have some conceptual difficulties in reconciling it with the existing basic scientific and clinical evidence.

Here is the list of my questions and concerns (all major), which I hope the authors will address.

Point 17

1. The experimental results presented by the authors show that passive head movements (PHM), which are calculated to replicate the acceleration effects of running, have quantitatively comparable effects on blood pressure to treadmill running in SHRSP. However, this is only correlative evidence

and does not indicate any causal relationships. The relevance of direct mechanical effects on the RVLM may be better tested by investigating whether pre-treatment by PHM or prevention of interstitial fluid movements can significantly reduce the antihypertensive effects of running in SHRSP.

Response 17

We appreciate the reviewer's comment. Following the reviewer's suggestion, we examined whether prevention of the interstitial fluid movements in the RVLM by hydrogel introduction reduced the antihypertensive effects of running in SHRSPs. As shown in Extended Data Fig. 12, treadmill running, which decreased the blood pressure in the control SHRSPs, was not significantly antihypertensive in the hydrogel-introduced SHRSPs. This suggests that the interstitial fluid movements are involved in the antihypertensive effects of treadmill running at a moderate velocity.

Point 18

2. All in vivo experiments in SHRSP were carried out when the blood pressure of the animals was still rising considerably. In fact, PHM appears to prevent a further increase in blood pressure rather than exerting a blood pressure lowering effect (see fig. 1B and fig. 6B). This raises the question of whether the effects of PHM observed in SHRSP may be more of a developmental nature. The antihypertensive effects of PHM should therefore also be shown in a phase of stable hypertension. If this is not possible in SHRSP, SHR could also be investigated.

Response 18

It has been reported that SHRSPs reach a phase of stable hypertension approximately at the age of 21 weeks. When initiated in this phase, the 4-week PHM did not significantly alter the blood pressure in SHRSPs. However, the PHM decreased the expression of AT1R in the RVLM. Together with the effect of 2-day PHM (see Response 19), we speculate that there might be complex mechanisms that link the AT1R expression in RVLM astrocytes to the blood pressure control, which can be irreversibly altered by some factor(s), such as vascular and renal (dys)function or severity of hypertension¹⁰⁻¹³. We have described these results (Extended Data Fig. 10) with relevant discussion (page 15–16, line 282–290) in our revised manuscript.

Point 19

3. There is a major discrepancy in the time course of in vivo responses to PHM (between day 8 and day 15) and in vitro responses that showed a rapid and strong inhibitory effect of pulsatile fluid shear stress on the expression of AT1R in cultured primary astrocytes, which was even more pronounced after 24 hours (Fig. 5). Even assuming that cumulative effects are necessary for the in vivo effect of increased fluid shear stress, a much earlier response should be expected if the in vitro model is valid. One way to test this could be to perform the experiment shown in fig. 2 after only two days of treatment with PHM. Please also provide time control data for 6 h and 24 h for the cell culture experiments.

Response 19

Following the reviewer's suggestion, we have examined whether 2-day PHM modulated the AT1R signaling in the RVLM of SHRSPs. As shown in Extended Data Fig. 9 of our revised manuscript, 2-

day PHM significantly reduced both pressor and depressor responses in SHRSPs, supporting the validity of the results from our in vitro fluid shear stress experiments.

We have also revised the figures for the in vitro fluid shear stress experiments using cultured cells, in which we provide control data for 6 h and 24 h (Fig. 5c–f and Extended Data Fig. 8b,c of our revised manuscript).

Point 20

4. Do SHRSP show unaltered responses to Ang II after the introduction of the hydrogel?

Response 20

As shown in Extended Data Fig. 11c–f, the hydrogel-introduced SHRSPs showed unaltered responses to the local injection of Ang II or valsartan into their RVLM. This indicates rapid diffusion of the solute in the hydrogels.

Point 21

5. The treatment with VOCR had a remarkably strong hypotensive effect (in one person the mean blood pressure dropped by almost 50 mmHg!). According to the authors' hypothesis, this reduction in blood pressure should be triggered mechanistically by a damping of sympathetic activity, however VOCR had practically no impact on all the indicators of sympathetic tone studied. To me, these results therefore seem to refute rather than support the authors' hypothesis. The authors should provide direct evidence of an inhibitory influence on sympathetic tone, e.g. by nerve conduction or

measurements of stable metabolites of the sympathetic neurotransmitters in urine. In addition, time control without VOCR is indispensable to allow a meaningful interpretation of the specific effects of VOCR.

Response 21

Following the reviewer's comment, we have examined whether non-oscillating chair riding (NOCR), where the chair was exactly the same as VOCR but the oscillation was left turned off, altered the blood pressure of hypertensive adult humans. We have also analyzed the influence of NOCR and VOCR on the vascular sympathetic nerve activity and the balance of cardiac sympathetic/parasympathetic nerve activity by conducting continuous beat-by-beat blood pressure/inter-beat (R-R) interval (RRI) recording at their beginning and ending periods.

As shown in Fig. 8a of our revised manuscript, the 4.5-week NOCR did not significantly alter the blood pressure of hypertensive adult humans. Furthermore, VOCR, but not NOCR, significantly decreased the LF power of systolic blood pressure variability (Fig. 8b) and showed a decreasing tendency ($P = 0.063$) in the LF/HF ratio of RRI variability (Fig. 8c). These results support our claim on the sympathoinhibitory effect of VOCR.

Point 22

6. AT1R antagonists are one of the most widely used drugs in the treatment of arterial hypertension. Since they can permeate the blood-brain barrier (PMID: 10882779), one would expect that exercise in hypertensive patients treated with sartans (or ACE inhibitors) should yield much weaker

antihypertensive effects. To my knowledge, this has not yet been described. Four of the 21 patients in this study were also treated with a sartan. Did these patients show a lower blood pressure response to VOCCR than the other subjects?

Response 22

In protocol 1, both of the two participants treated with sartans (ARB) showed decreases in the blood pressure by VOCCR; their MAP lowered by 6.7 mmHg and 4.0 mmHg (value of the day). In protocols 2 and 3, VOCCR significantly decreased the blood pressure (value of the week) in five hypertensives treated with ARB (Reviewer Fig. 1). The mean values of the blood pressure decrease in protocols 2 and 3 are slightly smaller in the ARB recipients (e.g., decrease in MAP: 4.5 mm Hg) as compared to that in the other (i.e., non-ARB recipient) hypertensive participants (e.g., decrease in MAP: 6.5 mm Hg) (Reviewer Fig. 2). However, the subject group (i.e., number of participants) was too small to draw a reasonable statistical conclusion as to whether ARB treatment attenuates blood pressure-lowering responses to VOCCR.

The literature that the reviewer refers to (PMID: 10882779; Nishimura et al. Brain Res 2000) demonstrates that chronic peripheral administration of candesartan, an ARB, blocks AT1R in the brain¹⁴. However, the BBB (blood-brain barrier) penetration of ARB appears to be complex and controversial, as it has been reported to depend on various factors, such as the chemical characteristics of particular ARB(s), dose, and length of treatment^{15,16}. Therefore, the antihypertensive effect of VOCCR observed in the ARB recipients in our study is not inconsistent with the VOCCR-induced attenuation of the AT1R signaling in the brain. None of the participants in our

study were treated with ACE inhibitor(s).

Reviewer Fig. 1

Reviewer Fig. 1 | VOCR has an antihypertensive effect on ARB recipients. BP and HR “value of the week”s immediately before and after the 4.5-week VOCR in protocols 2 and 3 (SBP: $P = 0.0247$. DBP: $P = 0.0362$. MAP: $P = 0.0273$. HR: $P = 0.3556$. $n = 5$). * $P < 0.05$; NS, not significant; paired two-tailed Student’s t -test.

Reviewer Fig. 2

Reviewer Fig. 2 | Comparison of VOCR effects between subjects with and without ARB treatment. Changes in the BP and HR “value of the week”s from immediately before to immediately after the 4.5-week VOCR in protocols 2 and 3 (SBP: $P = 0.8325$. DBP: $P = 0.8217$. MAP: $P = 0.6867$. HR: $P = 0.1292$. $n = 22$ for ARB -; $n = 5$ for ARB +). Data are presented as mean \pm s.e.m. NS, not significant; Mann-Whitney test.

Point 23

7. Other forms of exercise such as swimming, cycling or isometric exercise, which should have considerably less mechanical effects, produce equally large decreases in blood pressure as running. How do you explain this? Why, then, does running (because of the additional mechanically mediated effect) not lead to a significantly greater lowering of blood pressure than does isometric exercise, for example?

Response 23

As shown in Extended Data Fig. 3c,e of our revised manuscript, the PHM of SHRSPs with a peak vertical acceleration magnitude of $0.5 \times g$ or frequency of 0.5 Hz elicited an antihypertensive effect similar to that of the PHM with $1.0 \times g$ and 2 Hz (see Response 9). We observed that $\sim 0.5 \times g$ acceleration peaks can be generated at the human heads in the rostral-caudal direction during swimming and cycling of moderate intensities (Reviewer Fig. 3). Although we do not show this observation as a display item (i.e., figure) in our revised manuscript, it is reasonable to consider the possibility that the rostral-caudal head motion is involved in the antihypertensive effects of swimming and bicycle riding (page 25, line 471–474 of our revised manuscript). However, as we have stated in the “Limitation of study” section of the revised manuscript (page 29, line 546–547), the antihypertensive outcome of isometric exercise¹⁷ cannot be explained by direct mechanical effects on the brain; it is evidently mediated otherwise. For example, peripheral regulation of blood pressure-affecting factors such as vascular resistance may be responsible, or myokines that can modulate nervous cell function in the brain may play a major role.

Reviewer Fig. 3

Reviewer Fig. 3 | Trajectories of rostral-caudal accelerations in the human head during various types of exercise activities. The rostral-caudal acceleration was measured using an accelerometer attached to the forehead of a 59-year-old healthy man. In the top panel (swimming: crawl stroke), hollow arrows point to the acceleration peaks concomitant with the breaths, and solid arrows point to the peaks of acceleration waves synchronized with the arm strokes. Right-angled scale bars, 1 s / 0.5 × g.

Point 24

8. In the discussion, the authors state that the magnitude of shear forces caused by interstitial fluid movement in the RVLN is as large as the shear forces acting on the vascular endothelium. In view of the strength of the blood flow in the circulation, this does not seem plausible to me.

Response 24

With regard to our estimation of the magnitude of fluid shear stress exerted on the nervous cells in the rat RVLM, our calculation described in the original manuscript (0.59–2.64 Pa) was partially based on our inappropriate citation of previous publications on the viscosity of interstitial fluid in the brain (1–20 mPa·s cited from references #57–#59 in the original manuscript). We are still unable to refer to a previous report definitely describing the value of the intracerebral interstitial fluid viscosity. However, it has been reported that the composition of the interstitial fluid in the brain is similar to that of the cerebrospinal fluid (CSF)⁷. Therefore, we have referenced the value reported as the human cerebrospinal fluid viscosity (0.72 mPa·s) for the fluid shear stress calculation in our revised manuscript. The revised calculation gave 0.076–0.53 Pa as the magnitude of fluid shear stress on nervous cells in the rat RVLM during PHM. This value is considerably smaller than the magnitude of shear force on the arterial endothelium reported previously¹.

Regarding the concern raised by Reviewers #1 and #3 about our calculation being based on unreasonably simple modeling (see Points 11 and 25), please refer to Responses 11 and 25.

Reviewer #3 (Report for the authors (Required)):

OVERALL SUMMARY

In this study, the authors investigate the role of head movement as a byproduct of exercise in reducing essential hypertension. In order to isolate exercise-induced head movement (which they term passive head movement, or PHM) from the other effects of exercise, they implement a technique which they have previously described in order to oscillate the heads of rats in a

programmed fashion. In significant advances, they show that PHM reduces blood pressure in a rat model of hypertension whereas it has no effect on a WT control. They propose that this reduction in blood pressure is driven by PHM-induced fluid shear stress which decreases the expression of AT1R on astrocytes in the rostral ventrolateral medulla and affects blood pressure via the sympathetic nervous system. They present comprehensive evidence to support this claim. Finally, they investigate the effect of a similar approach, an oscillating chair, to reduce blood pressure in hypertensive humans. They find that it is effective and may act through a similar mechanism, providing a potential translational advancement. Overall, this study is interesting, comprehensive, and has significant ramifications relevant to human health, but key technical details, controls, and analyses are missing.

MAJOR COMMENTS

Point 25

1. The central premise of the proposed mechanism is that PHM causes changes in hydrostatic pressure which are sufficient to induce fluid shear stress (FSS) on the astrocytes within the RVLM. The authors present data which qualitatively indicates that PHM causes the induction of flow within the interstitial fluid (telemetry data from an implanted sensor, uCT of contrast agent). However, the effects of FSS on cells is dependent on the quantitative nature of the FSS (i.e., the magnitude of the shear matters as well as the dynamics of the flow). To validate their proposed mechanism, the authors performed an in vitro experiment wherein astrocytes and neurons were subjected to pulsatile fluid shear. To justify the parameters used in their in vitro experiments, the authors perform a simple calculation (which they confusingly refer to as a simulative calculation, despite no simulations conducted) based on data collected via uCT scans. These calculations are extremely simple and are very unlikely to be accurate. Critically, it is not clear whether this analysis is appropriate for oscillatory flow, as it does not seem possible to measure the dynamics of the flow from their uCT experiments. Additionally, the details of what was done are not presented clearly. For instance, it is

unclear how x,y,z coordinates in the contrast agent data relate to a cell. In addition to addressing the above points, a diagram showing the anatomy of the interstitial space/fluid with astrocytes and neurons, how fluid flow would occur, and the orientation of the pertinent axes would be helpful.

Response 25

We acknowledge the issue regarding the calculation of fluid shear stress exerted on nervous cells in the rat RVLM that we described in the original manuscript. While it was based on our intramedullary pressure measurement and μ CT analysis, we did not clearly justify the use of the values obtained from these experiments for the calculation of interstitial fluid shear stress in the rat RVLM.

Regarding the intramedullary pressure changes measured using a telemetry sensor (1.19 mm Hg), it may not accurately represent the pressure that the RVLM astrocytes experience during PHM. We also acknowledge that it is not reasonable to estimate the interstitial fluid flow velocity based solely on quantitative analysis of the contrast agent spreading observed in the μ CT images (Fig. 4g,h) (also see Response 11). In our revised manuscript, we have not referenced the value (1.19 mm Hg) obtained by our intramedullary pressure measurement.

However, the direction of interstitial fluid movement demonstrated in our μ CT experiments (Fig. 4g,h) appears to be consistent with the orientation of the interstitial space in the rat RVLM we observed in our analysis using a multiphoton microscope (Extended Data Fig. 6a–c,e); the *y*- and *z*-axes are dominant over the *x*-axis. Therefore, we have referred to the findings from our μ CT analysis to approximately estimate the PHM-induced enhancement of the interstitial fluid movement in the rat RVLM as two- to three-fold (also see Response 11). We have referenced 0.2 μ m/s as the velocity of

interstitial fluid movement in the brain of a sedentary rodent, citing previous publications^{5,6,18}.

Integrating these, we have entered 0.4–0.6 $\mu\text{m/s}$ (two- to three-times 0.2 $\mu\text{m/s}$) as the velocity of the interstitial fluid movement in the rat RVLM during PHM (Supplementary Table 1). As stated in Response 11, we are aware of the limitations and approximate nature of our calculation of the fluid shear stress magnitude. We also recognize the “non-physiologic” nature of the in vitro fluid shear stress experiments using cultured cells, as pointed out by Reviewer #1 (see Point 11). Taking these into account, we have extensively tested mechanical interventions (i.e., application of fluid shear stress or hydrostatic pressure change) with various different magnitudes/amplitudes to cultured astrocytes (Fig. 5 and Extended Data Fig. 8b).

Following the reviewer’s suggestion, we have described the diagrams illustrating the anatomical structure/orientation and dimension of the interstitial space in the rat RVLM based on our imaging experiments (Extended Data Fig. 6e and 7d,e) and relevant explanations of PHM-induced interstitial fluid movement (page 13, line 242–244 and the legend to Extended Data Fig. 7e).

Point 26

2. Given the data presented in the paper, there is no reason to assume that this simple model, even when explained in greater detail, accurately estimates the FSS experienced by astrocytes. This is highlighted by the fact that the high end for FSS values they estimate are on a similar order of magnitude to that experienced by endothelial cells within the vasculature (Resnick et al., 2003, Prog. Biophys Mol. Biol.). Additionally, if other velocity components were used these estimates would increase by a factor of 20-25. Furthermore, these estimates inform their choice of FSS magnitude for

use in cell culture. The authors should investigate other, more accurate methods of modeling/simulating interstitial fluid flow in the brain in response to movement and provide extensive justification for the estimate of FSS experienced by individual cells. A nice review of modeling within the brain interstitial system can be found in Lei et al., 2017, Prog. Neurobiol. Additionally, see work of Jennifer Munson, Roger Kamm, and Melody Swartz.

Response 26

We appreciate the reviewer's comment and suggestions. We acknowledge that our estimation of the interstitial fluid flow and shear stress that we described in the original manuscript was not accurate. Referring to the previous studies cited in the suggested reviews (Lei, Han et al. Prog. Neurobiol. 2017 and Chatterjee, Munson et al. J. Neurosci. Methods 2020)^{18,19}, we have analyzed the structure/orientation and dimension (cross-sectional area) of the interstitial space in the rat RVLM using multiphoton microscopy combined with fluorescent gel introduction. We have also revised the value of the intracerebral interstitial fluid viscosity, for which we erroneously cited previous publications in the original manuscript (see Response 24). In particular, the high end of the FSS value (2.64 Pa) we described in the original manuscript was based on our inappropriate citation of interstitial fluid viscosity (also see Response 24). Although we believe that the additional information and correction justify our calculation at least to some extent, we are still aware of the limitations of its accuracy in light of the in vivo situations, as stated above (Responses 11 and 25). Considering this, we have extensively conducted additional in vitro experiments in which we tested

fluid shear stress and hydrostatic pressure changes with various magnitudes/amplitudes (Fig. 5 and Extended Data Fig. 8b–e).

We have attempted more accurate methods of modeling/simulating interstitial fluid flow in the brain using the finite element analysis software, COMSOL Multiphysics (COSMOL, Burlington, MA). Based on the estimation by our imaging analysis (Extended Data Fig. 7c), we have simplified the interstitial spaces as voids formed by spherical particles with diameter of 0.10–0.48 μm that are arranged in a face-centered cubic lattice (Reviewer Fig. 4). Although the porosity in the brain has been reported to be ~ 0.2 ^{19,20}, we have entered 0.261, the available value closest to 0.2 for this lattice model in the software, as the porosity in our simulation (Reviewer Fig. 4). We have assumed Stokes flow with a fluid density and viscosity of 1 g/cm^3 and 0.72 $\text{mPa}\cdot\text{s}$ ⁷⁻⁹, respectively, that had an inflow velocity of 0.4–0.6 $\mu\text{m}/\text{s}$ at the inlet (i.e., the top of the model), zero-Pa pressure at the outlet, and symmetrical boundary conditions at the sides (Reviewer Fig. 5). The distribution of shear stress in the interstitial space was simulated, as shown in Reviewer Fig. 6, resulting in its average and maximum values on the spheres, as shown in Reviewer Table 1.

Reviewer Fig. 4

Reviewer Fig. 4 | Computational model of interstitial space in the brain arranged in a face-centered cubic lattice. The diameter of the sphere is assumed to be 0.10 μm in Model A and 0.48 μm in Model B.

Reviewer Fig. 5

Reviewer Fig. 5 | Simulation/modeling of Stokes flow of interstitial fluid in the brain.

Simulation with the sphere diameter of 0.10 μm (Model A) and the inflow velocity of 0.6 $\mu\text{m/s}$ is shown as an example. Cross-sectional heat maps represent the local distribution of the flow velocity.

Reviewer Fig. 6

Reviewer Fig. 6 | Simulated distribution of shear stress on the inner surface derived from interstitial fluid flow. Heat maps represent the local distribution of the shear stress intensity (magnitude).

Reviewer Table 1

Permeability [m^2]

Model	u_{in} [$\mu m/s$]	
	0.4	0.6
A	8.64×10^{-18}	1.30×10^{-17}
B	4.15×10^{-17}	6.23×10^{-17}

Wall shear stress (WSS) [m^2]

Model	WSS [Pa]	u_{in} [$\mu m/s$]	
		0.4	0.6
A	Average	0.135	0.203
	Maximum	0.311	0.466
B	Average	0.028	0.042
	Maximum	0.064	0.096

Reviewer Table 1 | Average and maximum values of simulated shear stress. The permeability was simulatively calculated with reference to the inflow velocity ($0.4 \mu m/s$ and $0.6 \mu m/s$), deriving the average and maximum values of wall shear stress on the inner surface of the region of interest (ROI) in Models A and B.

These computational simulations were based on the limited parameters obtained from previous publications in the literature and our imaging analyses. The geometry data currently available are insufficient for large-scale numerical simulations by the software; we are unable to accurately reproduce the complex structure of the interstitial space in the brain. Despite the input of a porosity of 0.261, which is larger than that previously reported (0.2)^{19,20}, the maximum value of the simulated shear stress in Model A (0.47 Pa; Reviewer Table 1) is within the range of magnitude of the fluid shear stress that significantly decreased the AT1R expression in cultured astrocytes (Fig. 5a). While it is beyond the scope of our current study to accurately determine the magnitude of interstitial fluid

shear stress in the brain, as stated in Response 11, our work may shed light on its importance in life science.

As an alternative to computer simulation, we have demonstrated a simple and analytical estimation of the fluid shear stress in the rat RVLM based on the parameters currently available (Supplementary Table 1). From our original manuscript, we have modified the process to estimate the Darcy permeability in the brain by employing the Kozeny-Carman equation, which allows us to calculate permeability given pore-size/sphericity data (see Methods).

Point 27

3. The use of uCT tracking of contrast agent may not capture the native fluid motion induced via PHM. Simply injecting a small volume of solution into the RVLM is likely to cause disruption of the tissue architecture surrounding the injection site. It is therefore likely that the effect PHM has on an injected bolus of fluid, in a region of the brain which may have been mechanically compromised, is different than how PHM affects the movement of interstitial fluid. For an example of a thorough analysis of injections within the CNS, see Guest et al., 2011, Brain Res. Bulletin. These issues should be addressed in the limitations of the study

Response 27

We appreciate the reviewer's comment. Although the infusion rate of the solution used for our μ CT analysis (0.2 μ l/min) was successfully employed in previous animal (rat) brain research by other groups²⁰⁻²², the spreading of the contrast agent (Isovist) may have been affected by the injection itself. Following the reviewer's suggestion, we tested the injection of the contrast agent at a rate of

44 nl/5 min, which has been reported to associate with excellent cerebral tissue preservation (Guest et al. 2011)²³. However, because of the insufficient contrast with this slow infusion rate, i.e., longer infusion time (114 min for 1- μ l infusion volume) or smaller infusion volume (0.1 μ l for 11.4-min infusion time), we were unable to obtain images by which we could reliably define the Isovist cluster (refer to the low-contrast X-ray opaque region in Reviewer Fig. 7a, as shown later). Whereas we have mentioned the issue of the possible compromise of brain tissue as one of the limitations of our study (page 28, line 529–533), we have referred to the findings from our μ CT experiments to approximately estimate the PHM-induced enhancement of interstitial fluid movement in the rat RVLM. Please also see Responses 11, 25, and 26.

Point 28

4. The use of a sensor to report the low-amplitude pressure waves within the interstitial fluid is not sufficiently described. How does the sensor operate, how were the results interpreted, and how does this relate to the movement of fluid? The brain is not a simple hollow organ, so measuring changes in pressure at one point may not clearly capture what is happening. Perhaps a supplemental note describing the use and interpretation of the sensor would be helpful.

Response 28

The MEMS-based sensor we used for the intramedullary pressure measurement was the same as that used to measure the blood pressure (intra-aortic pressure) of rats. It is 2.0 mm long, 0.47 mm wide, 0.60 mm thick (page 106, line 1408–1410), and detects the average pressure in the area in contact

with it. We agree with the reviewer that the brain is not a simple hollow organ and that the pressure (1.19 mm Hg) measured using a sensor of this size (a sub-mm- to mm-order) may not represent the magnitude of pressure changes at a cell-size level. In our revised manuscript, we have not referenced 1.19 mm Hg to calculate the fluid shear stress magnitude in the rat RVLM during PHM, but we interpreted it only as an indication of the stress distribution changes or microdeformation upon PHM (see Response 11).

Regarding the procedure of our pressure measurement in the brain, we have noticed that the method for the placement of the pressure sensor was not properly described in our original manuscript. In the Methods section of our revised manuscript, we have corrected and added the relevant description (page 111, line 1494–1504) and referred to this procedure as “intramedullary” pressure measurement in order to precisely represent what was actually done.

Despite the lack of strict confirmation of our sensor placement in the RVLM, the variation in the detected pressure values among the individual rats was fairly small (Extended Data Fig. 13i).

Point 29

5. The use of a PEG gel to restrict interstitial fluid flow is confusing and many of the potential off-target effects are not discussed. For example, PEG has been shown to improve outcomes following spinal cord injury via reduction of reactive oxygen species (Luo, Borgens, Shi 2002 J. Neurochem/Luo, Borgens, Shi 2004 J. Neurotrauma). Additionally, injecting a gel into the RVLM is likely to cause a disruption to the local architecture and disruption of cells. The authors themselves admit that injection of PEG gel is filled with challenges and “may not entirely prove the contribution

of interstitial fluid movement”. The rationale for this experiment, and its interpretation, is not clear. Can an improved version of μ CT tracking of contrast agent and subsequent analysis be used to demonstrate that the interstitial flow is changed in response to gel formation?

Response 29

It is technically difficult to interfere specifically with interstitial flow in the brain or other tissues/organs in living animals (page 27, line 512–514). We conducted PEG gel experiments to investigate the contribution of the interstitial flow, although we are aware of the possibility of off-target effects. In our revised manuscript, we have added new experimental results demonstrating that the expression of pro-inflammatory proteins (TNF- α and IL-1 β) and intramedullary pressure were not changed by hydrogel introduction in the rat RVLM (Extended Data Fig. 13h,i), excluding the involvement of massive detrimental processes. We have also cited one of the references mentioned by the reviewer (Luo, Borgens, Shi, J Neurotrauma 2004)²⁴ and discussed the possibility of the involvement of reactive oxygen species (page 28, line 523–526). Please also see Response 20.

As stated in Response 27, we were unable to obtain images by which we could reliably define the cluster of contrast agent (Isovist) injected at a rate of 44 nl/5 min. However, following the reviewer’s suggestion, we still attempted μ CT tracking of the contrast agent injected slowly (44 nl/5 min) in the gel-introduced rat RVLM. As shown in Reviewer Fig. 7, the contrast agent did not enter the parenchyma of the gel-introduced RVLM, demonstrating that the interstitial fluid flow is hindered in response to gel formation.

Reviewer Fig. 7

Reviewer Fig. 7 | Injected contrast agent does not enter the hydrogel-introduced rat RVLM.

Three hours after the injection of pre-gel PEG solution (b) or its ungelatable control (a) in the RVLM of WKY rats, 1 μ l of iodine-based contrast agent (Isovist) was injected at a rate of 44 nl/5 min into the RVLM. Following the 5-min syringe holding to avoid reflux after completing the injection, rats were subjected to brain μ CT scan. Images of the sagittal plane at 1.8 mm lateral to the midline are shown (caudal to the left and rostral to the right). Yellow “x”s mark 3.5 mm from the site of insertion of the injection needle at the dorsal brain surface, and indicate the location where the needle tip was placed during the injection. Magenta arrow in (a) points to the low-contrast X-ray-opaque region representing the Isovist spread around the injection site (i.e., “x”) in the control RVLM. White arrow in (b) points to the cluster of Isovist that presumably flowed back along the outer surface of the injection needle towards the dorsal surface of the medulla and partially entered the parenchyma outside the RVLM. Note the absence of Isovist spread surrounding “x” in the gel-introduced RVLM (b). Scale bars, 5 mm. The images are representative of at least three independent experiments with similar results.

Point 30

6. The statistical tests used in this paper to support their conclusions are occasionally incorrect. Specifically, for the measurements of a variable over time within the same group (e.g., how does BP change over time with PHM), the authors should use a repeated measures ANOVA to account for the fact that the measurements are being repeatedly taken from the same animals. The statistical tests

should be re-done and any resulting conclusions changed to highlight this.

Response 30

We appreciate the reviewer's comment. Following the reviewer's suggestion, we have re-performed the statistical analyses for the time-course type of data obtained from the same animals. We have also considered this issue in the additional experiments we conducted to revise the manuscript.

Specifically, we have used repeated measures ANOVA for the statistical analysis of the data presented in Fig. 1b, Fig. 3d, Fig. 6b, Extended Data Fig. 1a, Extended Data Fig. 2b, Extended Data Fig. 3b,d, Extended Data Fig. 5f, Extended Data Fig. 10b, Extended Data Fig. 12b, and Extended Data Fig. 14b.

The resultant alterations in *P* values from those described in our original manuscript did not change the conclusions related to them.

MINOR COMMENTS

Point 31

1. The authors should refrain from using the phrase "mechanical impact" unless referring to traumatic brain injury. In the case of this paper, "mechanical impact" grossly miscommunicates the current work. Perhaps something like "passive motion" would be more appropriate. This applies to both the title and body of the paper.

Response 31

Following the reviewers' comment, we have revised the title as follows.

Antihypertensive effect of brain-targeted mechanical intervention with passive head motion

We have also modified the text to avoid possible miscommunications associated with reference to traumatic brain injury (also see Response 14).

Point 32

2. It is possible that PHM has an additional effect via stimulating the muscles of the neck. How is PHM isolated from potential exercise of neck muscles?

Response 32

PHM may have some influence on neck muscles. However, hydrogel introduction in the RVLM almost completely eliminated the antihypertensive effect of PHM. Therefore, the neck muscles did not appear to play a significant role in the consequences of PHM analyzed in this study. Related to this, the carotid baroreceptor, which is also located in the neck, does not appear to be involved in the PHM effect (see Response 8).

Point 33

3. Why did the authors use only male rats in this study? Sex is an important variable for nearly all diseases, including hypertension. The limitations of studying only male rats should be discussed. The human subjects were of mixed gender. Were there gender-specific effects in this data?

Response 33

In many or even most animal experiments to investigate the pathogenesis of cardiovascular diseases, such as hypertension, males have been used unless there is a particular objective for assessing females (e.g., experiments to examine sex differences or actions of female sex hormones, estrogen, and progesterone). This is to preclude or minimize the potential influence of estrogen and progesterone, both of which basically act protectively on the cardiovascular system, including the heart and endothelium.

For similar reason, the antihypertensive effects of treadmill running in SHRs or SHRSPs have been demonstrated by experiments using males (e.g., Kishi et al., *Clin Exp Hypertens* 2012; Minami et al., *Am J Hypertens* 2003)^{25,26}. We intended to be consistent with these previous studies. However, we agree with the reviewer that sex is an important factor for hypertension. We have discussed this issue in our revised manuscript as one of the limitations of our animal study (page 26, line 487 – page 27, line 499). We have also demonstrated the lack of sex specificity for the antihypertensive effect of VOCCR in our human study (page 21, line 397–399 and Extended Data Fig. 17b).

OTHER COMMENTS

Point 34

1. The authors continually referred to use of PEG to “gel interstitial fluid”. This is an improper term. To gel means to move from the solution phase to the gel phase. The fluid is not going into a gel

phase. The polymerization of the gel creates a region with much smaller pore size, which we reduce the flow in the system for a given pressure drop.

Response 34

We appreciate the reviewer's comment. We realize that the phrase "gel interstitial fluid" is incorrect.

We have modified all the terms related to this mistake accordingly.

Point 35

2. In line 161, the authors describe the results as approximate mirror images. This is imprecise, alternative wording should be used to more clearly communicate what the authors mean.

Response 35

Following the reviewer's suggestion, we have revised the wording to indicate what we mean precisely and clearly (page 10, line 190).

Point 36

3. The authors describe the effect of fluid shear on cultured astrocytes in vitro as being cumulative. However, that does not clearly connect to the observed response of BP over time. According to figure 1B, the MAP of rats experiencing PHM continues to rise during the course of the study, just at a lower rate than the PHM- rats. This should be clarified.

Response 36

We appreciate the reviewer's comment. We realize that it is not reasonable to assume that the time required to detect a significant antihypertensive effect of PHM on SHRSPs (>2 weeks) indicates the accumulation of the fluid shear stress-induced decrease in the AT1R expression in RVLM astrocytes. Together with the results of our short-period (2-day) PHM experiments (Extended Data Fig. 9; also see Response 19), we interpret >2 weeks as the time required for attenuated AT1R expression in RVLM astrocytes to elicit its antihypertensive outcome rather than the cumulative decrease in AT1R expression. In the revised manuscript, we have modified the statement related to this issue (page 15, line 278–282).

Regarding the effect of PHM on the development and plateau phase of hypertension in SHRSPs, please refer to Response 18. We speculate that there might be complex mechanisms that link AT1R expression in RVLM astrocytes to blood pressure control, which is irreversibly altered in SHRSPs during the plateau phase of their hypertension development (page 15–16, line 282–290 of our revised manuscript).

Point 37

4. On line 345, the authors mention that mechanical loading and/or FSS alleviates inflammation, which can affect AT1R expression. However, it is unclear how to connect inflammation within bone tissue (cited as one example) to inflammation within the CNS, which is immune-privileged. Additionally, reversible, oscillatory FSS is associated with increase inflammation in the vasculature. These comments need refined for accuracy.

Response 37

The AT1R expression has been reported to be transcriptionally enhanced by pro-inflammatory proteins, including TNF- α and IL-1 β ²⁷. In our original manuscript, we intended to refer to the FSS-induced decrease in AT1R expression in cultured astrocytes as the anti-inflammatory effect of FSS^{28,29}, rather than the connection of the brain to the bone or vasculature. However, we acknowledge that our statement was too speculative and unclear. To avoid misleading descriptions, we have deleted this sentence from our manuscript.

Point 38

5. Line 366/377 (“As the phrase...”) is too conversational and speculative and not of an appropriate tone for a journal article.

Response 38

We agree that the paragraph starting with “As the phrase...” is too conversational and speculative. Because it does not convey the significance of this study, we have deleted this paragraph from our manuscript.

Point 39

6. On line 389/390, what do the authors mean by mechanical factors being a target within the blood-brain barrier? It is unclear how a mechanical factor could be a specific target.

Response 39

We intended to refer to the possibility that physical conditions or matters in the brain, such as the (micro)structure of the RVLM, are involved in the pathogenesis of hypertension or other brain-related disorders. However, we acknowledge that our statement is unclear and too speculative.

Therefore, we have deleted this sentence from our manuscript.

Point 40

7. In Fig. 1g, 1h, 6e and Extended Data Fig. 1b, it is not clear that the GFAP-AGTRAP targeted astrocytes and not neurons. Given the close proximity between astrocytes and neurons, looking at neuronal nuclei and suggesting that the GFP staining is mutually exclusive is not clear from this picture. Likewise, in Extended Data 1D it is not clearly indicated that the GFP and GFAP labeling comes from different cells. Greater details regarding the determination of AT1R-positive neurons and AT1R-positive astrocytes should be provided? Was the overlap assessment done manually or was it automated? It is also not clear how the RVLM was histologically sectioned and whether all the cell counts came from roughly the same relative section for each rat – is AT1R expression uniform throughout the RVLM?

Response 40

In Fig. 1g, 1h, 6e and Extended Data Fig. 1b in the original manuscript (Fig. 1g, 1h, 6e and Extended Data Fig. 4a in our revised manuscript), samples were prepared from SHRSPs that were not

subjected to adeno-associated virus (AAV) injection. The green fluorescence in these figures represents signals emitted from the secondary antibody bound to the primary antibody against GFAP, but not GFP from AAV-infected cells (AAV was not used in the experiments for Figs. 1 and 6). The AAV was used only in the experiments shown in Fig. 3 and Extended Data Fig. 5 (Fig. 3 and Extended Data Fig. 2, respectively, in our original manuscript). The mutual exclusiveness between GFAP-AGTRAP and NSE-AGTRAP is demonstrated in Extended Data Fig. 5b–e of our revised manuscript (Extended Data Fig. 2b–e of our original manuscript).

The assessment of the overlap of anti-GFAP and anti-NeuN staining with anti-AT1R staining was done manually. Anti-AT1R immunosignals whose overlap with anti-GFAP or anti-NeuN immunosignals was not clearly determined were excluded from the quantification of AT1R⁺ astrocytes and AT1R⁺ neurons.

Since the AT1R expression did not appear to be completely uniform, we counted cells in the approximately same relative section for each rat (11.3–12.3 mm caudal to the bregma, 1.5–2.5 mm lateral to the midline, and 0–1 mm dorsal to the ventral surface of the medulla). We have described this information in the Methods section (page 108, line 1439–1442).

Point 41

8. In line 677, the nomenclature for positive cells should be changed. The use of the “-“ symbol is confusing because this is commonly used to refer to negative or absent expression. Perhaps a different symbol, such as “/” could be used to compound proteins together?

Response 41

As suggested by the reviewer, we have used “/” instead of “-” to compound proteins together in the legends to Fig. 5d (page 69, line 1109), Extended Data Fig. 5b (page 55, line 980–981), and Extended Data Fig. 8e (page 67, line 1091).

References

1. Malek, A.M., Alper, S.L. & Izumo, S. Hemodynamic shear stress and its role in atherosclerosis. *JAMA* **282**, 2035-2042 (1999).
2. Furlan, R., *et al.* Oscillatory patterns in sympathetic neural discharge and cardiovascular variables during orthostatic stimulus. *Circulation* **101**, 886-892 (2000).
3. Milutinovic, S., Murphy, D. & Japundzic-Zigon, N. Central cholinergic modulation of blood pressure short-term variability. *Neuropharmacology* **50**, 874-883 (2006).
4. Pagani, M., *et al.* Power spectral analysis of heart rate and arterial pressure variabilities as a marker of sympatho-vagal interaction in man and conscious dog. *Circ Res* **59**, 178-193 (1986).
5. Geer, C.P. & Grossman, S.A. Interstitial fluid flow along white matter tracts: a potentially important mechanism for the dissemination of primary brain tumors. *J Neurooncol* **32**, 193-201 (1997).
6. Kingsmore, K.M., *et al.* MRI analysis to map interstitial flow in the brain tumor microenvironment. *APL Bioeng* **2**(2018).
7. Cserr, H.F. & Patlak, C.S. Secretion and bulk flow of interstitial fluid. in *Physiology and Pharmacology of the Blood-Brain Barrier*. ed. Bradbury, M.W.B. 245-261. Springer Berlin Heidelberg, Berlin, Heidelberg. (1992).
8. Yetkin, F., *et al.* Cerebrospinal fluid viscosity: a novel diagnostic measure for acute meningitis. *South Med J* **103**, 892-895 (2010).
9. Bloomfield, I.G., Johnston, I.H. & Bilston, L.E. Effects of proteins, blood cells and glucose on the viscosity of cerebrospinal fluid. *Pediatr Neurosurg* **28**, 246-251 (1998).
10. Husken, B.C., Hendriks, M.G., Pfaffendorf, M. & Van Zwieten, P.A. Effects of aging and hypertension on the reactivity of isolated conduit and resistance vessels. *Microvasc Res* **48**, 303-315 (1994).
11. Sepehrdad, R., *et al.* Amiloride reduces stroke and renal injury in stroke-prone hypertensive rats. *Am J Hypertens* **16**, 312-318 (2003).

12. Pirici, D., *et al.* Common impact of chronic kidney disease and brain microhemorrhages on cerebral A β pathology in SHRSP. *Brain Pathol* **27**, 169-180 (2017).
13. Smeda, J.S. Renal function in stroke-prone rats fed a high-K⁺ diet. *Can J Physiol Pharmacol* **75**, 796-806 (1997).
14. Nishimura, Y., Ito, T., Hoe, K. & Saavedra, J.M. Chronic peripheral administration of the angiotensin II AT(1) receptor antagonist candesartan blocks brain AT(1) receptors. *Brain Res* **871**, 29-38 (2000).
15. Kishi, T., Hirooka, Y. & Sunagawa, K. Sympathoinhibition caused by orally administered telmisartan through inhibition of the AT(1) receptor in the rostral ventrolateral medulla of hypertensive rats. *Hypertens Res* **35**, 940-946 (2012).
16. Michel, M.C., Foster, C., Brunner, H.R. & Liu, L. A systematic comparison of the properties of clinically used angiotensin II type 1 receptor antagonists. *Pharmacol Rev* **65**, 809-848 (2013).
17. Carlson, D.J., Dieberg, G., Hess, N.C., Millar, P.J. & Smart, N.A. Isometric exercise training for blood pressure management: a systematic review and meta-analysis. *Mayo Clin Proc* **89**, 327-334 (2014).
18. Chatterjee, K., Carman-Esparza, C.M. & Munson, J.M. Methods to measure, model and manipulate fluid flow in brain. *J Neurosci Methods* **333**, 108541 (2020).
19. Lei, Y., Han, H., Yuan, F., Javeed, A. & Zhao, Y. The brain interstitial system: Anatomy, modeling, in vivo measurement, and applications. *Prog Neurobiol* **157**, 230-246 (2017).
20. Shi, C., *et al.* Transportation in the interstitial space of the brain can be regulated by neuronal excitation. *Sci Rep* **5**, 17673 (2015).
21. Faingold, C.L., *et al.* GABA in the inferior colliculus plays a critical role in control of audiogenic seizures. *Brain Res* **640**, 40-47 (1994).
22. Li, G.Q., *et al.* Role of cannabinoid receptor type 1 in rostral ventrolateral medulla in high-fat diet-induced hypertension in rats. *J Hypertens* **36**, 801-808 (2018).
23. Guest, J., Benavides, F., Padgett, K., Mendez, E. & Tovar, D. Technical aspects of spinal cord injections for cell transplantation. Clinical and translational considerations. *Brain Res Bull* **84**, 267-279 (2011).
24. Luo, J., Borgens, R. & Shi, R. Polyethylene glycol improves function and reduces oxidative stress in synaptosomal preparations following spinal cord injury. *J Neurotrauma* **21**, 994-1007 (2004).
25. Kishi, T., *et al.* Exercise training causes sympathoinhibition through antioxidant effect in the rostral ventrolateral medulla of hypertensive rats. *Clin Exp Hypertens* **34**, 278-283 (2012).
26. Minami, N., *et al.* Effects of exercise and β -blocker on blood pressure and baroreflexes in spontaneously hypertensive rats. *Am J Hypertens* **16**, 966-972 (2003).
27. Elton, T.S. & Martin, M.M. Angiotensin II type 1 receptor gene regulation: transcriptional and posttranscriptional mechanisms. *Hypertension* **49**, 953-961 (2007).
28. Hahn, C. & Schwartz, M.A. Mechanotransduction in vascular physiology and atherogenesis. *Nat Rev Mol Cell Biol* **10**, 53-62 (2009).

29. Miyazaki, T., *et al.* Mechanical regulation of bone homeostasis through p130Cas-mediated alleviation of NF- κ B activity. *Sci Adv* **5**, eaau7802 (2019).

Rebuttal 2

Point 1

*Clarification of whether the antihypertensive effect is mediated by reduced AT1R expression in astrocytes in the RVLM (with compelling arguments and/or additional evidence), or discussion of the limitations.

Response 1

As shown in Fig. 3d-f of our manuscript, the adeno-associated virus (AAV)-mediated expression of angiotensin II (Ang II) type 1 receptor (AT1R)-associated protein (AGTRAP), which interacts with AT1R and tempers the Ang II-mediated signals by promoting AT1R internalization¹, in astrocytes but not in neurons of the bilateral RVLMs in SHRSPs significantly lowered the blood pressure (BP) as compared with the control. This indicates the causal or hierarchical relationship between the AT1R signaling in RVLM astrocytes and the BP. However, the BP-lowering effect of exogenous expression of AGTRAP in RVLM astrocytes was not long-lasting and became non-significant three weeks after the AAV injection (Extended Data Fig. 5g in our revised manuscript). Considering the persistent AAV-mediated protein expression in brain cells for a few months or longer², it is unlikely that the short-lasting BP lowering with the exogenous expression of AGTRAP in RVLM astrocytes results from short duration of AGTRAP overexpression.

Related to this, the transgenic mice in which AGTRAP was specifically overexpressed in the renal tubules did not present a significant decrease in the basal BP although they did manifest a reduced BP-increasing response to pressor-dose Ang II infusion³. This raises the possibilities as follows: (1) AGTRAP overexpression persistently attenuates both steady-state and responsive AT1R signaling in the renal tubules, the former of which is somehow disconnected from the regulation of BP; (2) attenuation of steady-state AT1R signaling by AGTRAP overexpression is not permanent or long-lasting. At the moment, we cannot preclude the possibility (1). However, the deletion of endogenous AT1R (type A) in the renal proximal tubules decreased the basal BP in mice^{4,5}, indicating the persistent connection between attenuated steady-state AT1R signaling in the renal tubules and basal BP lowering. Considering these previous reports together with the short duration of the basal BP lowering consequent upon the exogenous expression of AGTRAP in RVLM astrocytes (Extended Data Fig. 5g), we speculate that the decreasing effect of AGTRAP overexpression on steady-state AT1R signaling is

relatively short-lasting because of a compensatory or neutralizing mechanism, which is yet to be defined.

Our approach using the AAV system enabled us to modulate the local AT1R signaling in a cell-type specific manner, thereby differentiating between the roles of RVLM astrocytes and neurons (Fig. 3), both of which have been reported to be involved in BP regulation (page 22, line 413-414 in our revised manuscript) ⁶⁻¹¹. In addition, the rat genome harbors two distinct genes for AT1R, *Agtr1a* and *Agtr1b*, which would make effective, local, and cell-type specific genetical silencing of AT1R expression more uncertain and complicated. The use of AAV-mediated AGTRAP overexpression seemed to be advantageous in this regard as well. However, the aforementioned issue concerning the duration of the basal BP-lowering effect of AGTRAP overexpression in RVLM astrocytes gives a limitation to explicitly demonstrating the apparently complex connection between the decrease in AT1R signaling in RVLM astrocytes and the basal BP lowering consequent on daily PHM or treadmill running, which becomes significant in three weeks or later (Fig. 1b and Extended Data Fig. 12b).

In our revised manuscript, we state that the data shown in Fig. 3 supports the notion that the reduced AT1R expression in RVLM astrocytes mediates the BP-decreasing effects of PHM and treadmill running (page 28, line 528-532). On the other hand, we have explicitly described the limitation of experiments using exogenous AGTRAP expression (page 12, 213-216 and page 28-29, line 533-551). Related to this issue, we have used the term, “basal BP” in some particular contexts where we refer distinctly to the steady-state resting BP, but not casual or responsive BP like the pressor/depressor response, in rats.

Point 2

*Discussion on why the changes are not observed in stable hypertension and how A1TR be disconnected from change in blood pressure.

Response 2

Given the time lag from the PHM-induced decrease in AT1R signaling in the RVLM astrocytes (within two days; Extended Data Fig. 9) to the decrease in basal BP in SHRSPs (three weeks or longer; Fig. 1b), there is unlikely to be a direct connection between them. Alternatively, the link from the AT1R signaling in RVLM astrocytes to the basal BP is presumably comprised of slow or chronic multifactorial processes.

It has been reported that the hypothalamic paraventricular nucleus (PVN) integrates the signals concerning the factors that affect sympathetic activity and BP, such as Ang II, proinflammatory cytokines, and reactive oxygen species (ROS), from the basal forebrain, and sends the information to the RVLM¹²⁻¹⁵. AT1R signaling induces proinflammatory processes and ROS production in astrocytes¹⁶. Collectively, we assume that the persistent increase of AT1R expression in the RVLM astrocytes in SHRSPs (e.g., compare columns 1 and 3 in Fig. 1j) gives rise to sustained or chronic inflammation and oxidative stress in the aforementioned brain regions involved in the sympathetic activity and BP regulation. Because of the chronic nature of the increased AT1R signaling in the RVLM astrocytes in SHRSPs, the consequential inflammation and oxidative stress are also chronic, involving multiple factors but lacking in a potential of quick responsive changes. As a result, the PHM-induced decrease in AT1R signaling in the RVLM astrocytes may not promptly lead to a decrease in the sympathetic outflow. Such a notion conforms to a previous report describing the time course of exercise-induced attenuation of inflammation and oxidative stress in the PVN followed by the decrease in the basal BP in SHR¹⁷. It also agrees with the involvement of multiple pro-/anti-inflammatory and pro-/anti-hypertensive factors in the PVN and lamina terminalis (LT) in the effect of exercise in hypertension-induced rats¹⁸.

Furthermore, unlike an acute change in sympathetic activity, the consequence of a slow and moderate decrease in steady-state sympathetic outflow may involve relatively slow or time-consuming vascular responses and processes such as remodeling¹⁹, which has been reported to be positively modulated by exercise^{20,21}.

Regarding the disconnection of AT1R signaling in the RVLM astrocytes from the basal BP control in SHRSPs, we speculate that it derives from irreversible or refractory organic changes in one or more element of the aforementioned link between them and/or other factors affecting the BP. Chronic inflammation and oxidative stress are implicated in severe degenerative organic damages in various tissue and organs, including the blood vessel²², kidney²³, and brain²⁴. In particular, it has been reported that hypertension is prone to become irreversible when the renal function is severely impaired²⁵. From the apparently limited contribution of the systemic renin-angiotensin system (RAS) to the antihypertensive effects of exercise revealed by a meta-analysis²⁶, impairments in renal function may negatively affect the BP regulation fairly independently of PHM and override its antihypertensive effect in SHRSPs. Whereas transient yet sustained AT1R activation can cause irreversible impairments of the renal

function and hypertension²⁷, severe kidney damages are observed in 20-week-old SHRSPs²⁸.

From the emergence of apparent vascular remodeling and cell senescence at 24 weeks and calcification at 48 weeks of age in SHR²⁹, it is possible or even likely that the pathological changes in the vessels in SHRSPs^{30,31} confer severe irreversible or refractory functional defects in their vascular system at ≥ 21 weeks of age. SHRSPs share several more characteristics commonly with clinical cases of refractory hypertension, including glucose intolerance^{32,33}, left ventricular hypertrophy^{34,35}, stroke proneness^{32,35}, and sympathetic hyperactivity^{36,37}. Taken together, the hypertension in SHRSPs is possibly prone to become more refractory with the advance of age, involving severe organic damages in multiple tissues and organs.

In the revised manuscript, we have specifically discussed the connection (page 25-26, line 461-484 and page 28-29, line 528-546) and disconnection (page 15-16, line 288-293 and page 30-31, line 569-583) between the attenuated AT1R signaling in RVLM astrocytes and the decrease in basal BP adequately citing previous literatures.

Point 3

*Discussion of the limitations of the data that you can gather to support your claims on AT1R downregulation, in particular regarding estimation of changes in fluid shear stress.

Response 3

We estimated the magnitude of fluid shear stress exerted on the cells in the rat RVLM as 0.076-0.53 Pa (Supplementary Table 1). However, the fluid shear stress of 0.1 Pa, which is within this range, did not significantly alter the AT1R expression in cultured astrocytes (Fig. 5a). We speculate that this discrepancy derives from the approximate nature of our estimation of fluid shear stress in vivo and/or non-physiologic nature of the in vitro fluid shear stress experiments using cultured cells. Although we referred to this issue in the “Limitation of study” section in our manuscript of the previous version (line 500-503 in the previous version), we acknowledge that our statement was unclear, lacking in specific reference to the aforementioned discrepancy. In our revised manuscript, we have discussed the limitation of our study pertaining to the magnitude of fluid shear stress that decreases the AT1R expression in the RVLM astrocytes in vivo, specifically referring to the discrepancy between our estimation in vivo and experimental results in vitro (page 29-30, line 552-565 in our revised manuscript).

As before, when you are ready to resubmit your manuscript, please upload the revised files, a point-by-point rebuttal to the comments from all reviewers, the reporting summary, and a cover letter that explains the main improvements included in the revision and responds to any points highlighted in this decision.

Point 4

* Clearly highlight any amendments to the text and figures to help the reviewers and editors find and understand the changes (yet keep in mind that excessive marking can hinder readability).

Response 4

We have clearly highlighted the amendments we have made, including those related to Extended Data Fig. 5g, in red characters.

Point 5

* Consider including responses to any criticisms raised by more than one reviewer at the beginning of the rebuttal, in a section addressed to all reviewers.

Response 5

Reviewers #2 and #3 raised a concern about the discussion of the connection/disconnection between the AT1R signaling in RVLM astrocytes and the BP regulation in SHRSPs. As we have addressed this issue in Response 2, we state so just before the beginning of Reviewer #1's comments.

A CONCERN RAISED BY MORE THAN ONE REVIEWER, AND OUR RESPONSE TO IT

Reviewers #2 and #3 raised a concern about the connection/disconnection between the AT1R signaling in RVLM astrocytes and the BP regulation in SHRSPs. Please refer to Response 2, where we have addressed this issue.

Reviewer #1 (Report for the authors (Required)):

Overall, the paper remains compelling and the authors are to be lauded for attempting to address almost all of the concerns raised by the reviewers and editors. Accordingly, the authors have conducted a significant number of experiments at the in vitro and in vivo levels in both rodents and humans. While their data does not definitively prove their hypothesis beyond a reasonable doubt, they do certainly and reasonably point in that direction. Indeed, the experiments that are required to obtain truly definitive proof might not even be practically feasible, at least from this reviewer's perspective. For example, it is unclear what the shear stresses and fluid dynamics of the relevant microenvironment truly are but it is even more unclear how one could even truly obtain those measurements experimentally yet accurately and the authors have done their best to estimate those values. In addition, the authors have aptly expanded the discussion of the limitations of their methods and data, which increases the scientific rigor and transparency. Nevertheless, the overall concept of the revised manuscript will still have significant implications for the study and treatment of hypertension in the context of cardiovascular disease.

Point 6

The enthusiasm for this manuscript is dampened somewhat by the lack of placement of this work in the context of other researchers who have conducted similar experiments. While this paper is novel, other papers in this space should be more extensively cited and compared/contrasted to better orient the reader.

Response 6

The previous revision of our manuscript included referring to the antihypertensive effects of exercise at the beginning of Abstract, Introduction, and Discussion sections. However,

we acknowledge that we did not sufficiently reference preceding studies of the mechanisms of how exercise lowers the BP in hypertensive humans and animals. In our revised manuscript, we have cited the literatures describing the positive effects of exercise on vascular structure, such as increases in luminal diameter of arteries^{20,21} and in capillary density in skeletal muscle³⁸, endothelial function³⁹, redox homeostasis, and inflammation^{17,40} in light of BP control (page 22, line 407-409). Although direct mechanical effects on the brain do not seem to be considered in these previous studies of exercise as an antihypertensive measure, our findings do not conflict with them but rather provide the mechanism underlying them. Therefore, we believe that we have now better placed our work in the context of others' research in the relevant field (page 22, line 13 409-412).

Among the references we have newly cited regarding the relevant subject, the papers demonstrating the effects of exercise on the brain in the context of hypertension^{17,18} are helpful for discussing the chronic and multifactorial nature of the mechanism that links AT1R expression in the RVLM astrocytes to the BP control in hypertensive rats (page 25-26, line 477-481 in our revised manuscript; see Response 2). Furthermore, the report of a meta-analysis that does not support a clear role for the systemic RAS in the antihypertensive effect of exercise²⁶ agrees with the importance of the brain RAS, and helps us discuss the disconnection between the AT1R signaling in the RVLM astrocytes and the basal BP in SHRSPs (page 31, line 578-581 in our revised manuscript; also see Response 2).

Thus, we appreciate the reviewer's comment.

Reviewer #2 (Report for the authors (Required)):

This is a very thorough revision of the original manuscript. The authors present important additional experimental data which substantially strengthens their conclusion that mechanically mediated effects on the brain contribute significantly to the antihypertensive effect of physical exercise. They have also completely revised the calculation of the mechanical forces involved and now arrive at much more plausible values.

Point 7

Whether the antihypertensive effect is mediated by an attenuated expression and pharmacological responsiveness of the CNS AT1R signaling system, however, is in my

view not yet sufficiently experimentally documented. Essentially, the argument entirely relies on the estimation of changes in fluid shear stress elicited by PHM in vivo, which is still arbitrary. The demonstration that in the presence of a blockade of the AT1R signaling system in the RVLM region (either pharmacologically or genetically), the antihypertensive effects of PHM are absent or at least greatly attenuated would make a causal role of this mechanism much more likely. The authors should at least explicitly address this limitation in the discussion.

Response 7

We acknowledge the reviewer's comment about the demonstration of the PHM effect in the presence of a blockade of the AT1R signaling in the RVLM astrocytes. As stated in Response 1, the exogenous AGTRAP expression in the RVLM astrocytes elicited a relatively short-lasting basal BP-lowering effect; therefore, we realize that this approach does not fit with the experiments to examine the antihypertensive outcome of PHM or treadmill running, which becomes significant in ≥ 3 weeks. In the revised manuscript, we have explicitly addressed this issue as one of the limitations of our current study (page 29, line 546-551). With regard to the detail of the problem concerning AGTRAP overexpression, please refer to Response 1.

Reviewer #3 (Report for the authors (Required)):

After reviewing the extensive rebuttal letter and highly revised manuscript, I have only a single major concern remaining. In Response (4) [Rebuttal pg 13-14], the authors respond to a point that Reviewer 2 made which also touched on a point I made (pts 18, 19, and 36), which is that the time evolution of the in vitro astrocytic response does not conceptually align with the time evolution of the data on PHM intervention in rats.

Point 8

They performed an experiment requested by Reviewer 2, and reported that "Four-week PHM did not significantly alter the blood pressure of SHRSPs when initiated in a phase of stable hypertension. However, PHM still decreased the expression of A1TR in the RVLM." First, this data does not appear to be in the manuscript. It should be added.

Response 8

Please refer to Extended Data Fig. 10 in which we show the data that PHM does not lower the BP in SHRSPs, but decrease the AT1R expression in their RVLM during the plateau phase of hypertension development.

Point 9

Also, this result appears inconsistent with the authors' main point and consistent with Reviewer 2's alternative hypothesis. The authors should provide a full discussion regarding how Reviewer 2's alternative hypothesis can be excluded from potential interpretations. Key questions to be addressed are 1) why are changes not observed in stable hypertension? And 2) How can A1TR be disconnected from change in blood pressure if that is the core of their proposed mechanism? This is clearly an important result as it has implications both for the clinical relevance of their PHM intervention and the interpretation of their proposed mechanism. Thus, the authors should address this in more detail and clearly demonstrate its inclusion in the manuscript prior to publication.

Response 9

As stated in Response 2, we assume that a complex multifactorial mechanism, rather than a quick direct link, connects the attenuated AT1R expression in the RVLM astrocytes to the decrease in the basal BP in SHRSPs. In concrete, PHM or treadmill running attenuates the AT1R signaling in RVLM astrocytes, thereby alleviating the inflammation and oxidative stress in the brain region involved in the regulation of sympathetic activity and BP, including the PVN17 and RVLM itself⁴¹. Because of the chronic nature of the increased AT1R signaling in the RVLM astrocytes in SHRSPs, the consequent inflammation and oxidative stress are chronic, multifactorial, and devoid of a potential of quick responsive changes in sympathetic outflow. Therefore, there may reasonably be a considerable delay in the attenuation of sympathetic outflow consequent on the PHM induced decrease in the AT1R expression in the RVLM astrocytes. Furthermore, the attenuation of steady-state sympathetic activity may involve relatively slow vascular events, such as remodeling¹⁹, to manifest its antihypertensive consequence. Such proposed mechanisms are responsible or account for the time lag between PHM-induced decrease in the AT1R expression in the RVLM astrocytes and the basal BP lowering in SHRSPs. We have described these discussions regarding the connection between the AT1R signaling in RVLM astrocytes and the basal BP in SHRSPs in our revised manuscript (page 25–26, line 461–484).

Concerning the disconnection between them, we speculate that it originates from irreversible or refractory organic changes in the aforementioned intervening mechanisms and/or other elements/factors affecting the BP in SHRSPs. As in other cases of chronic inflammation and oxidative stress²²⁻²⁴, the hypertension-relevant pathogenic changes in the brain of SHRs¹⁷ or SHRSPs⁴¹ may become irreversible or more refractory with the advance of age, involving extended duration and/or aggravated severity of inflammation and adverse stress. Furthermore, the impairments of kidney²⁸ and vascular function²⁹⁻³¹ possibly become so severe as to potentiate the irreversibility of hypertension²⁵ in SHRSPs during the plateau phase of the BP development, i.e., at the age of ³²¹ weeks. As also stated in Response 2, the hypertension in SHRSPs may be prone to become more refractory with age. In our revised manuscript (page 15-16, line 288-293 and page 30–31, line 569–583), we have discussed the issue related to the disconnection between the AT1R signaling in RVLM astrocytes and the BP in SHRSPs.

References

1. Tamura, K., *et al.* The physiology and pathophysiology of a novel angiotensin receptor-binding protein ATRAP/Agtrap. *Curr Pharm Des* **19**, 3043-3048 (2013).
2. Lo, W.D., *et al.* Adeno-associated virus-mediated gene transfer to the brain: duration and modulation of expression. *Hum Gene Ther* **10**, 201-213 (1999).
3. Wakui, H., *et al.* Enhanced angiotensin receptor-associated protein in renal tubule suppresses angiotensin-dependent hypertension. *Hypertension* **61**, 1203-1210 (2013).
4. Li, H., *et al.* Renal proximal tubule angiotensin AT1A receptors regulate blood pressure. *Am J Physiol Regul Integr Comp Physiol* **301**, R1067-1077 (2011).
5. Gurley, S.B., *et al.* AT1A angiotensin receptors in the renal proximal tubule regulate blood pressure. *Cell Metab* **13**, 469-475 (2011).
6. Jancovski, N., *et al.* Angiotensin type 1A receptor expression in C1 neurons of the rostral ventrolateral medulla contributes to the development of angiotensin-dependent hypertension. *Exp Physiol* **99**, 1597-1610 (2014).
7. Guo, F., *et al.* Astroglia are a possible cellular substrate of angiotensin(1-7) effects in the rostral ventrolateral medulla. *Cardiovasc Res* **87**, 578-584 (2010).
8. Chen, D., *et al.* Angiotensin type 1A receptors in C1 neurons of the rostral ventrolateral medulla modulate the pressor response to aversive stress. *J Neurosci* **32**, 2051-2061 (2012).

9. Chen, D., Bassi, J.K., Walther, T., Thomas, W.G. & Allen, A.M. Expression of angiotensin type 1A receptors in C1 neurons restores the sympathoexcitation to angiotensin in the rostral ventrolateral medulla of angiotensin type 1A knockout mice. *Hypertension* **56**, 143-150 (2010).
10. Allen, A.M., *et al.* Expression of constitutively active angiotensin receptors in the rostral ventrolateral medulla increases blood pressure. *Hypertension* **47**, 1054-1061 (2006).
11. Kishi, T. Regulation of the sympathetic nervous system by nitric oxide and oxidative stress in the rostral ventrolateral medulla: 2012 Academic Conference Award from the Japanese Society of Hypertension. *Hypertens Res* **36**, 845-851 (2013).
12. Johnson, A.K. & Gross, P.M. Sensory circumventricular organs and brain homeostatic pathways. *FASEB J* **7**, 678-686 (1993).
13. Mimeo, A., Smith, P.M. & Ferguson, A.V. Circumventricular organs: targets for integration of circulating fluid and energy balance signals? *Physiol Behav* **121**, 96-102 (2013).
14. Dampney, R.A. Central neural control of the cardiovascular system: current perspectives. *Adv Physiol Educ* **40**, 283-296 (2016).
15. Sharma, N.M., *et al.* Central Ang II (angiotensin II)-mediated sympathoexcitation: role for HIF-1a (hypoxia-inducible factor-1a) facilitated glutamatergic tone in the paraventricular nucleus of the hypothalamus. *Hypertension* **77**, 147-157 (2021).
16. Gowrisankar, Y.V. & Clark, M.A. Angiotensin II induces interleukin-6 expression in astrocytes: Role of reactive oxygen species and NF-kB. *Mol Cell Endocrinol* **437**, 130-141 (2016).
17. Masson, G.S., *et al.* Time-dependent effects of training on cardiovascular control in spontaneously hypertensive rats: role for brain oxidative stress and inflammation and baroreflex sensitivity. *PLoS One* **9**, e94927 (2014).
18. Xue, B., *et al.* Voluntary exercise prevents hypertensive response sensitization induced by angiotensin II. *Front Neurosci* **16**, 848079 (2022).
19. Wu, L.L., *et al.* Impact of selective renal afferent denervation on oxidative stress and vascular remodeling in spontaneously hypertensive rats. *Antioxidants (Basel)* **11**(2022).
20. Dinunno, F.A., *et al.* Regular endurance exercise induces expansive arterial remodelling in the trained limbs of healthy men. *J Physiol* **534**, 287-295 (2001).
21. Thijssen, D.H., de Groot, P.C., Smits, P. & Hopman, M.T. Vascular adaptations to 8-week cycling training in older men. *Acta Physiol (Oxf)* **190**, 221-228 (2007).

22. Mas-Bargues, C., Borrás, C. & Alique, M. The Contribution of extracellular vesicles from senescent endothelial and vascular smooth muscle cells to vascular calcification. *Front Cardiovasc Med* **9**, 854726 (2022).
23. Shabaka, A., Cases-Corona, C. & Fernandez-Juarez, G. Therapeutic insights in chronic kidney disease progression. *Front Med (Lausanne)* **8**, 645187 (2021).
24. Saavedra, J.M. Evidence to consider angiotensin II receptor blockers for the treatment of early Alzheimer's disease. *Cell Mol Neurobiol* **36**, 259-279 (2016).
25. Beilin, L.J. & Ziakas, G. Vascular reactivity in post-deoxycorticosterone hypertension in rats and its relation to 'irreversible' hypertension in man. *Clin Sci* **42**, 579-590 (1972).
26. Goessler, K., Polito, M. & Cornelissen, V.A. Effect of exercise training on the reninangiotensin-aldosterone system in healthy individuals: a systematic review and metaanalysis. *Hypertens Res* **39**, 119-126 (2016).
27. Heijnen, B.F., *et al.* Irreversible renal damage after transient renin-angiotensin system stimulation: involvement of an AT1-receptor mediated immune response. *PLoS One* **8**, e57815 (2013).
28. Kato, T., Mizuguchi, N. & Ito, A. Blood pressure, renal biochemical parameters and histopathology in an original rat model of essential hypertension (SHRSP/Kpo strain). *Biomed Res* **36**, 169-177 (2015).
29. Shi, X., *et al.* Ageing-related aorta remodelling and calcification occur earlier and progress more severely in rats with spontaneous hypertension. *Histol Histopathol* **33**, 727-736 (2018).
30. Husken, B.C., Hendriks, M.G., Pfaffendorf, M. & Van Zwieten, P.A. Effects of aging and hypertension on the reactivity of isolated conduit and resistance vessels. *Microvasc Res* **48**, 303-315 (1994).
31. Harvey, A.P., *et al.* Vascular dysfunction and fibrosis in stroke-prone spontaneously hypertensive rats: The aldosterone-mineralocorticoid receptor-Nox1 axis. *Life Sci* **179**, 110-119 (2017).
32. Yamori, Y., *et al.* Glucose tolerance in spontaneously hypertensive rats. *Jpn Circ J* **42**, 841-847 (1978).
33. Armario, P., *et al.* Prevalence and clinical characteristics of refractory hypertension. *J Am Heart Assoc* **6**(2017).
34. Inoue, T., *et al.* Heart-bound adiponectin, not serum adiponectin, inversely correlates with cardiac hypertrophy in stroke-prone spontaneously hypertensive rats. *Exp Physiol*

102, 1435-1447 (2017).

35. Acelajado, M.C., *et al.* Refractory hypertension: definition, prevalence, and patient characteristics. *J Clin Hypertens (Greenwich)* **14**, 7-12 (2012).
36. Cui, Z.H., *et al.* Exaggerated response to cold stress in a congenic strain for the quantitative trait locus for blood pressure. *J Hypertens* **22**, 2103-2109 (2004).
37. Modolo, R., de Faria, A.P., Almeida, A. & Moreno, H. Resistant or refractory hypertension: are they different? *Curr Hypertens Rep* **16**, 485 (2014).
38. Gliemann, L., *et al.* Capillary growth, ultrastructure remodelling and exercise training in skeletal muscle of essential hypertensive patients. *Acta Physiol (Oxf)* **214**, 210-220 (2015).
39. Pedralli, M.L., *et al.* Different exercise training modalities produce similar endothelial function improvements in individuals with prehypertension or hypertension: a randomized clinical trial exercise, endothelium and blood pressure. *Sci Rep* **10**, 7628 (2020).
40. Agarwal, D., *et al.* Role of proinflammatory cytokines and redox homeostasis in exercise-induced delayed progression of hypertension in spontaneously hypertensive rats. *Hypertension* **54**, 1393-1400 (2009).
41. Kishi, T., *et al.* Exercise training causes sympathoinhibition through antioxidant effect in the rostral ventrolateral medulla of hypertensive rats. *Clin Exp Hypertens* **34**, 278-283 (2012).